# Meta-analysis of randomised controlled trials testing behavioural interventions to promote household action on climate change

Claudia F. Nisa[1]*, Jocelyn J. Bélanger[1], Birga M. Schumpe[1] & Daiane G. Faller[2]

No consensus exists regarding which are the most effective mechanisms to promote household action on climate change. We present a meta-analysis of randomised controlled trials comprising 3,092,678 observations, which estimates the effects of behavioural interventions holding other factors constant. Here we show that behavioural interventions promote climate change mitigation to a very small degree while the intervention lasts ($d = -0.093$ 95% CI $-0.160$, $-0.055$), with no evidence of sustained positive effects once the intervention ends. With the exception of recycling, most household mitigation behaviours show a low behavioural plasticity. The intervention with the highest average effect size is choice architecture (nudges) but this strategy has been tested in a limited number of behaviours. Our results do not imply behavioural interventions are less effective than alternative strategies such as financial incentives or regulations, nor exclude the possibility that behavioural interventions could have stronger effects when used in combination with alternative strategies.

[1] Psychology Division, New York University Abu Dhabi, PO BOX 129188 Abu Dhabi, UAE. [2] Center for Global Sea Level Change, Research Institute, New York University Abu Dhabi, PO BOX 129188 Abu Dhabi, UAE. *email: cfn1@nyu.edu

There is unequivocal consensus that we must reduce emissions to net-zero in the coming decades. Regarding lifestyle behaviour changes, pathways to curb global warming require adopting energy-efficient appliances, reducing food waste, reducing meat and dairy intake, and promoting energy-saving behaviours, such as decreasing the use of air conditioning and increasing the use of public transportation[1]. However, while there is little controversy about the need to embrace these mitigation strategies, no scientific agreement exists regarding which are the most effective mechanisms to achieve it.

This paper examines which interventions are effective in promoting climate change mitigation by individuals and households, and how effective efforts have been to date. We specifically examine the impact of behavioural interventions—interventions that do not involve economic (dis)incentives and regulations—holding all other factors constant. This is the specific focus of this paper because there have been growing hopes for behavioural interventions[2,3], such as consumption feedback, social comparison messages or tailored environmental appeals[4–6], as potential (cost)effective alternatives to traditional market tools and regulations. There are several areas of consumer behaviour relevant to climate change that are not typically targeted by governmental legislation, e.g., meat and dairy eating habits, or food waste. Moreover, market tools, such as price variations or subsidies, have been shown less effective than expected to reduce electricity demand[7] or to promote the consumption of renewable energies[8]. In other areas, prices are not even used as an instrument to influence consumers' behaviour: the price of water at the consumer touchpoints is typically low, and penalties for not recycling are largely inexistent. What is more, in some areas such as air travel or consumption of animal products, market mechanisms work against climate change mitigation, with prices decreasing over time to incentivise demand[9]. Resulting from all the above, behavioural interventions are gaining momentum as non-trivial policy tools and represent an increasingly larger portion of the action on climate change[2–4].

The relative effectiveness of such interventions, nevertheless, has not been carefully evaluated. There have been several attempts to summarise evidence on this topic, including quantitative meta-analyses[10–14], but this previous work suffers from substantive limitations. These limitations include estimating behavioural effects from self-reported beliefs, attitudes or intentions, as well as combining estimates from observational and experimental study designs, in addition to mixing artificial lab evidence with real world behaviour. Such limitations pose severe problems for impact evaluation. On the one hand, estimates from quasi or non-experimental studies tend to be larger and are more likely to be significant by comparison to estimates based on more stringent experimental designs[15]. On the other hand, the gap between attitudes and intentions reported in the lab, and real-world behaviour is well-documented[16], and similarly, the impact of interventions on self-reported measures tends to be higher and more likely to reach statistical significance compared to effects on tangible behaviour[17]. Such biasing effects should not be underestimated. These may induce the perception that most common interventions are effective, erroneously leading scholars and policy makers to use the available (possibly inflated) estimates as the basis for important predictions for climate change mitigation[18].

We address this gap by performing a large-scale meta-analysis strictly based on randomised controlled field trials, measuring factual changes in behaviour. The behaviours targeted are essential to mitigate climate change and include household and individual behaviours directly relevant to the production of greenhouse gases emissions. The six included behaviours are as follows. First, energy consumption in the home: within this category we examined interventions that targeted electricity or gas demand, purchase of energy-efficient appliances or consumption of renewable energy. Second, transportation: this category includes the analysis of air travel, private car use, or alternative options such as walking, cycling, or use of public transportation. Changes in both energy and transportation behaviours are deemed critical to achieve reductions in $CO_2$ emissions[18]. Third, consumption of animal products: interventions targeting reduced consumption or avoidance of meat and dairy products. Recent estimates[19] highlight that, particularly in developed countries, eating fewer animal products has a high potential for carbon emissions reduction. Fourth, food waste: interventions aiming to reduce the frequency or amount of food wasted. Food waste is a vastly overlooked driver of climate change. The Food and Agriculture Organisation (FAO) quantified the food wastage in terms of carbon footprint and concluded that the contribution of food wastage emissions to global warming is almost equivalent (87%) to global road transport emissions[20]. Fifth, water consumption: interventions targeting the quantity of water consumed including, for instance, time in shower or laundry frequency. Household actions that consume the largest amounts of water (showers and dishes/clothes washing) are also actions that consume large amounts of energy, and thus saving water saves energy (US Department of Energy)[21,22]. Sixth, recycling: includes both the reuse of natural resources, recycling of plastic, paper or cans. Recycling contributes to reducing emissions from resource extraction, landfill methane and waste incineration (US Environmental Protection Agency [EPA])[23].

The content of the interventions, that is, what strategy was implemented to try to change behaviour, was not defined a priori but was identified bottom-up from the literature. This allowed us to understand what the predominant strategies are, opening a window into the conceptual assumptions shared by practitioners about what is expected to work.

Our work suggests that behavioural interventions, acting alone, will provide little benefit to mitigate climate change. Results show that most household behaviours relevant to mitigate climate change exhibit resistance when targeted only by behavioural interventions. In particular, critical actions such as the purchase of energy-efficient appliances and private car use were barely affected by the behavioural interventions examined. The most promising types of interventions in terms of effectiveness are choice architecture (nudges) and social comparison messages. Information-based interventions, although the most common strategies used to date, have very limited impact. Our results do not prioritise regulatory and financial incentive-based policy prescriptions above behavioural interventions, but suggest the need to examine interactions between behavioural and non-behavioural strategies.

## Results

**Descriptive statistics.** The final sample according to our inclusion criteria comprised 83 studies, providing a total of 144 estimates. These estimates are unique, meaning that these originated from independent datasets and controlled for same data/author bias[24]. Included papers were published between 1976 and 2017, and are fully described in Supplementary Data 1 and Supplementary References.

Table 1 summarises the main descriptive statistics and the effect sizes estimated per key moderators. About 40% of the papers targeted families, comprising a total of 724,792 households. The remaining interventions targeted discrete participants, adding to a total of 2,367,886 individuals. Sample sizes were small (≤100 per experimental arm) in more than half the papers. Only about 13% of the papers reported a large, robust number of

**Table 1 Effect sizes per key moderators**

| Moderator | | k | N | Effect size d (CI) | $I^2$ (%) | POB (%) |
|---|---|---|---|---|---|---|
| Overall effect size | | 144 | 3,092,678 | −0.093 (−0.160, −0.055) | 64.6** | 6.6 |
| *Sensitivity analysis* | | | | | | |
| Sample type | Households | 66 | 724,792 | −0.112 (−0.221, −0.057) | 73.1** | |
| | Individuals | 78 | 2,367,886 | −0.118 (−0.221, −0.060) | 51.9** | |
| Sample size per condition | ≤100 | 82 | 5709 | −0.335 (−−0.555, −0.190) | 49.9** | |
| | ]100, 500[ | 45 | 22,840 | −0.141 (−0.280, −0.063) | 51.4** | |
| | ≥500 | 17 | 3,074,121 | −0.028 (−0.106, −0.006) | 25.6 | |
| Self-selection | Self-selected | 79 | 12,550 | −0.279 (−0.465, −0.161) | 60.3** | |
| | Naïve | 65 | 3,080,128 | −0.040 (−0.103, −0.016) | 53.6** | |
| Region | Europe | 43 | 2,333,441 | −0.210 (−0.446, −0.093) | 58.6** | |
| | US/Canada | 78 | 750,854 | −0.108 (−0.208, −0.054) | 72.7** | |
| | Rest World | 23 | 8383 | −0.059 (−0.407, −0.013) | 0 | |
| *Behaviour*[a] | | | | | | |
| Energy | | 47 | 719,059 | −0.094 (−0.133, −0.055) | 67.7** | 6.6 |
| | Appliances | 12 | 108,077 | −0.036 (−0.129, 0.058) | 22.6 | 2.5 |
| Transportation | | 29 | 2,245,972 | −0.136 (−0.183, −0.089) | 98.4** | 9.6 |
| | Car use | 21 | 2,242,781 | −0.036 (−0.039, −0.034) | 0 | 2.5 |
| Water | | 42 | 124,082 | −0.052 (−0.079, −0.025) | 40.1** | 3.7 |
| | Towel | 18 | 8909 | −0.168 (−0.271, −0.064) | 47.8** | 11.9 |
| Food waste | | 4 | 218 | −0.231 (−0.518, 0.056) | 21.6 | 16.3 |
| Meat | | 7 | 666 | −0.239 (−2.81, 0.008) | 36.8 | 16.9 |
| Recycling | | 23 | 2766 | −0.457 (−0.595, −0.319) | 69.9** | 32.3 |
| *Intervention* | | | | | | |
| Information | | 53 | 2,354,243 | −0.048 (−0.075, −0.021) | 34.7** | 3.4 |
| Social comparison | | 32 | 719,756 | −0.077 (−0.108, −0.046) | 72.2** | 5.4 |
| Engagement | | 38 | 10,486 | −0.253 (−0.336, −0.170) | 71.8** | 17.9 |
| | Commitment | 10 | 1446 | −0.480 (−0.704, −0.255) | 75.8** | 33.9 |
| Appeals | | 10 | 5952 | −0.266 (−0.445, −0.086) | 70.5** | 18.8 |
| Nudges | | 11 | 795 | −0.352 (−0.492, −0.212) | 0 | 24.9 |

Note: k = #estimates; N = sample size; $I^2$ = Heterogeneity; POB = probability of benefit (effect size $d/\sqrt{2}$)
**$p < 0.05$
[a]The total aggregate sample size per analysis of behaviour is 3,092,763—an additional 85 individuals than the overall 3,092,678. This difference is due to a single study (Kurz et al. 2005 in Supplementary References) testing the effect of an intervention in both water and energy and, thus, its sample (N = 85) was accounted in both behaviours

observations (≥500 per experimental arm) but this small group of papers correspond to over 90% of the total number of available observations. Around 60% of the papers were based on self-selected samples, that is, studies in which participants opted-in or explicitly consented to participate.

Regarding the duration of interventions, these ranged from a single day (one-shot intervention) to 730 days (2 years). Most interventions were short-lived, with mean duration 64 days (SD = 140 days) and median 7 days. Half of the interventions lasted up to 1 week, two-thirds lasted up to a month (67.4%) and 83.4% up to 3 months (Fig. 1).

In terms of geographical origin, roughly half of all studies were conducted in the US/Canada, and about 30% were conducted in European countries. With respect to the behaviour targeted, most available estimates were related to energy consumption (36%; N = 30, 47 estimates), followed by transportation choices (23%; N = 20, 29 estimates), water use (23%; N = 19, 42 estimates), and recycling (16%; N = 13, 23 estimates). There is a very limited number of interventions targeting food waste and demand for livestock products. For some behaviours, such as demand for air travel or renewable energies, no randomised controlled trials (RCTs) using behavioural interventions were identified.

Content-wise, there was a wide range of strategies used and, in many cases, interventions implemented a bundle of experimental stimuli, such as combining tips on how to save energy, making appeals and reporting on neighbourhood behaviour. Based on inter-judge agreement (see the "Methods" section), we propose a categorisation that maximised the singularity of different interventions, focusing on the key content of the materials delivered to participants. Interventions used as main strategy one

of the following several possibilities. First, information: in this category, interventions range from simple messages conveying tips on how to save energy, in-home displays, energy labels or statistics about climate change (N = 41, 53 estimates). Second, appeals: here experimental stimuli are delivered under the form of requests, pleas and appeals to change behaviour based on values of humanity, cooperation and social responsibility (N = 9, 10 estimates). Third, engagement: interventions in this category try to change psychological processes, such as promoting goal-setting, implementation intentions, commitment or engagement, and mindfulness towards climate change mitigation (N = 33, 38 estimates). Fourth, social comparison: these interventions tend to provide a comparative reference with respect to the mitigation behaviours of close others, such as neighbours, colleagues/friends or fellow citizens, based on principles of social influence and social comparison (N = 22, 32 estimates). Lastly, choice architecture: usually designated as nudges[25], these interventions influence behaviour by removing external barriers, expediting access or facilitating climate change mitigation behaviours by altering the structure of the environment in which people make choices. Examples are putting recycling bins closer to regular waste, reducing plate/glass size or setting air conditioning by default to higher temperatures (N = 7, 11 estimates).

**Behavioural interventions have small effects ceteris paribus.** Based on the criterion of statistical significance, behavioural interventions promote climate change mitigation while the intervention is in place (d = −0.093 95% CI −0.160, −0.055). However, the aggregate magnitude of behavioural interventions is very small. In reference to Cohen's $d$[26], values below 0.1 tends to

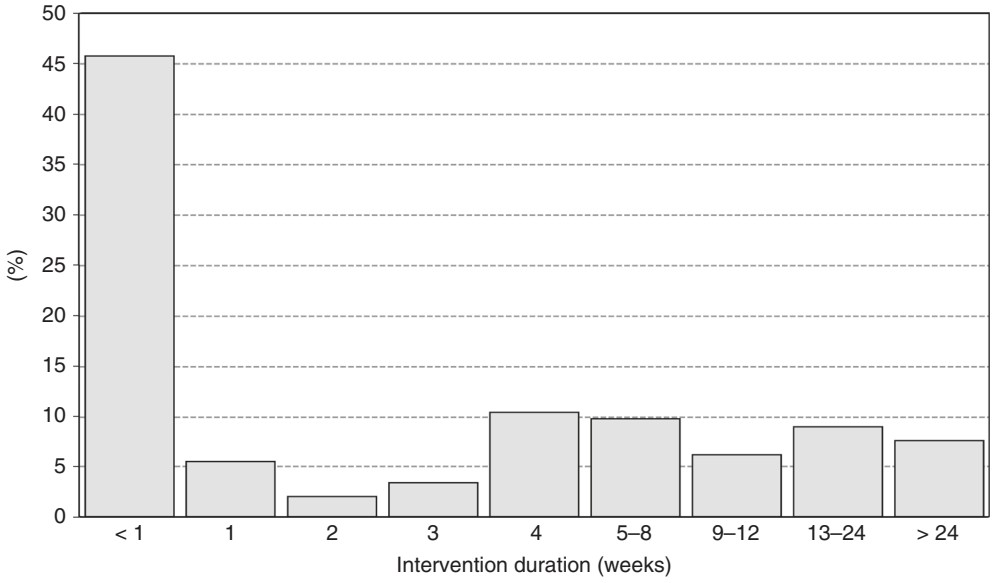

**Fig. 1** Duration of the behavioural interventions in weeks. Shows the distribution of intervention duration according to the number of weeks from the beginning to the conclusion of the experimental period

be interpreted as a very small effect, around 0.2 a small effect, 0.3 −0.4 a medium-small effect, 0.5–0.6 a medium effect, and 0.7 and above a large effect.

We present all effect sizes $d$ in combination with an additional estimate of behavioural plasticity: probability of benefit (POB)[27]. POB is the probability that the intervention will promote climate change mitigation behaviours in the experimental group, by comparison to the control no-intervention group. This estimate is based on the magnitude of the effect size $d$ ($d/\sqrt{2}$)[27] and provides a more intuitive measure of impact. Overall, the expected POB from behavioural interventions is 6.6%. This means that the probability that a randomly selected individual in the experimental group shows an improvement in mitigating climate change exceeding that of a randomly selected individual in the control group is 6.6%.

However, there is high between-study heterogeneity [$I^2 = 64.6\%$ $p < 0.001$[28]], suggesting large variations in the effectiveness of interventions according to moderator variables. Using meta-regression models[29], we examined whether the duration of the intervention could be a possible moderator of the overall effect but results showed no significant effect ($B = -0.001$, $p = 0.435$). We explored alternative sources of this heterogeneity, testing the effect of sample type (households vs. individuals), sample size, self-selection and geographical region (USA, Europe, and rest of the world) (Table 1).

Both sample size and self-selection significantly explain the variability found in the overall estimate. Restricting the calculation of the effect size to the more precise, robust estimates based on studies with no self-selection bias or studies with large samples, results show smaller average impacts (respectively, $d = -0.040$ 95% CI $-0.103$, $-0.016$; and $d = -0.028$ 95% CI $-0.106$, $-0.006$ $I^2$ ns). This adjustment estimates the aggregate POB between 2% and 3%.

The fact that studies with small samples reported substantially higher average effect sizes than papers with larger samples ($d = -0.335$ vs. $d \le -0.141$, respectively) suggests small-study bias[30]. Figure 2 visually displays the relationship between estimate precision (based on standard errors) and the magnitude of the effect reported. Estimates with lower standard errors tend to cluster around a zero effect. This visual examination is statistically confirmed by the Egger's test[31], with a significant negative slope ($B = -0.0321$, $p < 0.001$) indicating that the smaller the sample,

the stronger the effect size reported. This result would not pose a problem if small sample studies were reporting low between-study variance ($I^2 < 40\%$). However, only the large sample studies ($N \ge 500$ per group) are estimating precise effect sizes for their population ($I^2 = 25.6\%$ ns).

Moreover, cumulative effect sizes over time (cumulative meta-analysis) show a temporal profile of decreasing impact, from high to medium effect sizes up to the early 1990s, to then medium-small effect sizes for the next two decades, and small to very small effect sizes in approximately the last 10 years (Fig. 3). This analysis suggests that the larger effects from smaller studies were more characteristic several decades ago, whereas more recent studies provide more conservative estimates.

With respect to sustained effects after the intervention is concluded, 15 studies reported 21 estimates which measured if any lasting changes could be identified. There is no evidence of a durable positive impact ($d = -0.13$ 95% CI $-0.42$, 0.28). The follow-up period was on average 21 weeks (SD = 23.2 weeks) with a median of 6.5 weeks. Meta-regression results show no relationship between follow-up duration and the significance of interventions in the longer term. Given that no sustained effects were identified, the following analyses of impact per type of behaviour and type of intervention are based on estimates while the intervention is in place.

**Most mitigation actions have low behavioural plasticity**. Estimates for all examined behaviours tend to be low, with the exception of recycling, which reaches a medium effect size ($d = -0.457$ 95% CI $-0.595$, $-0.319$ POB 32.3%). Interventions promoting a reduction in energy consumption also have an expected POB of 6.6% ($d = -0.094$ 95% CI $-0.133$, $-0.055$). Within the category of energy studies, we included studies targeting the purchase of energy-efficient appliances. The impact of interventions to specifically influence these purchases is estimated with precision ($I^2 = 22.6\%$ ns) as only marginally significant ($d = -0.036$ 95% CI $-0.129$, 0.058), with an estimated POB of 2.5%.

Interventions targeting transportation, including less private car use and more public transportation, bicycle, and walking, have a combined POB of 9.6% ($d = -0.136$ 95% CI $-0.183$, $-0.089$). However, combining all interventions targeting

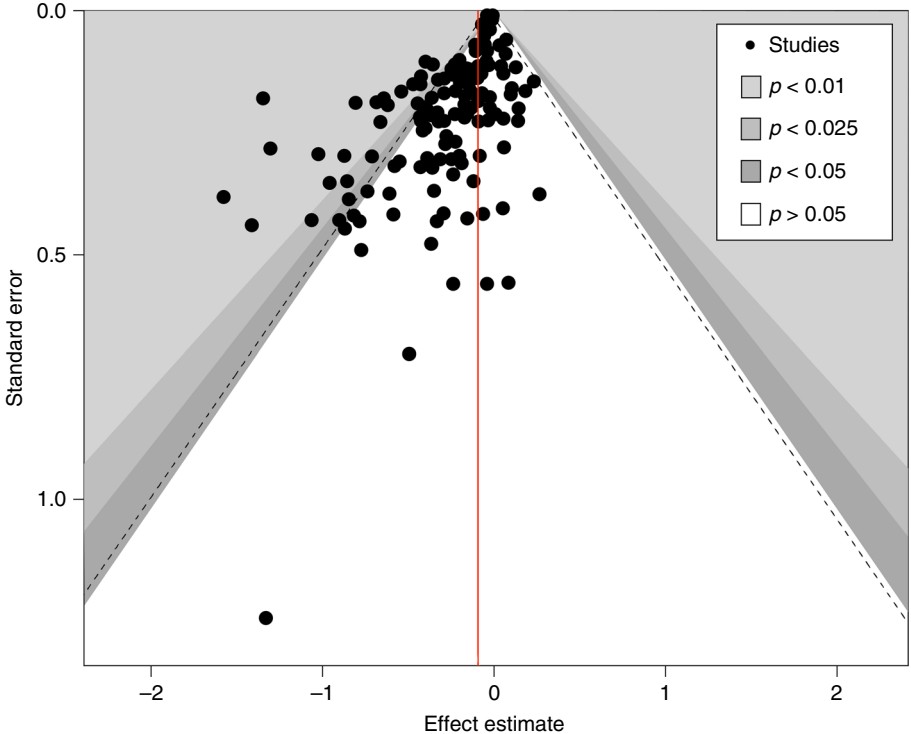

**Fig. 2** Funnel plot displaying the relationship between estimate quality and effect size. Each dot represents a study (e.g. measuring the effect of a certain behavioural intervention); the *y*-axis represents study precision (standard error) and the *x*-axis shows the study's result (effect estimate). This scatterplot is used for the visual detection of systematic heterogeneity between studies. It assumes that studies with high precision will be plotted near the average (red line), and studies with low precision will be spread evenly on both sides of the average, creating a roughly funnel-shaped distribution. Deviation from this shape suggest small-study bias, which is the case here, with lower precision studies reporting stronger effects

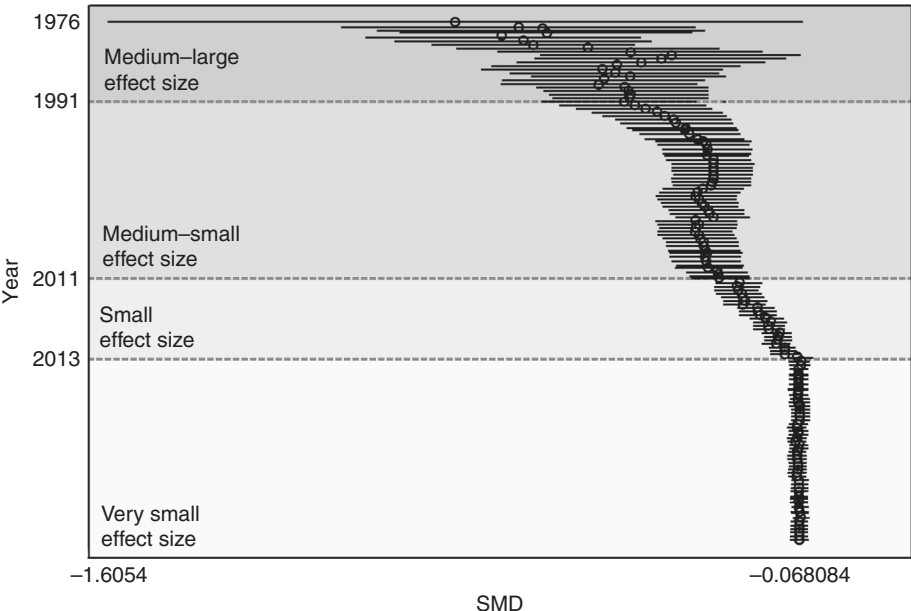

**Fig. 3** Cumulative meta-analysis. In cumulative meta-analysis, the pooled estimate of the treatment effect is updated each time the result from a new study is included. The circles in the plot represent the cumulative effect size at any given moment, that is, the average effect size resulting from the inclusion of studies up to a certain point. This allows tracking the accumulation of evidence on the effect of behavioural interventions over time

transportation reveals highly heterogeneous estimates ($I^2 = 98.4\%$) implying that very different effects are obtained for different types of transportation modes. Isolating the impact for decreasing private car use substantially reduces the POB to 2.5%

with an effect size close to zero ($d = -0.036$ 95% CI $-0.039$, $-0.034$). This is a statistically precise estimate ($I^2 = 0\%$).

Attempts to reduce water consumption also show small effects ($d = -0.052$ 95% CI $-0.079$, $-0.025$). This category includes a

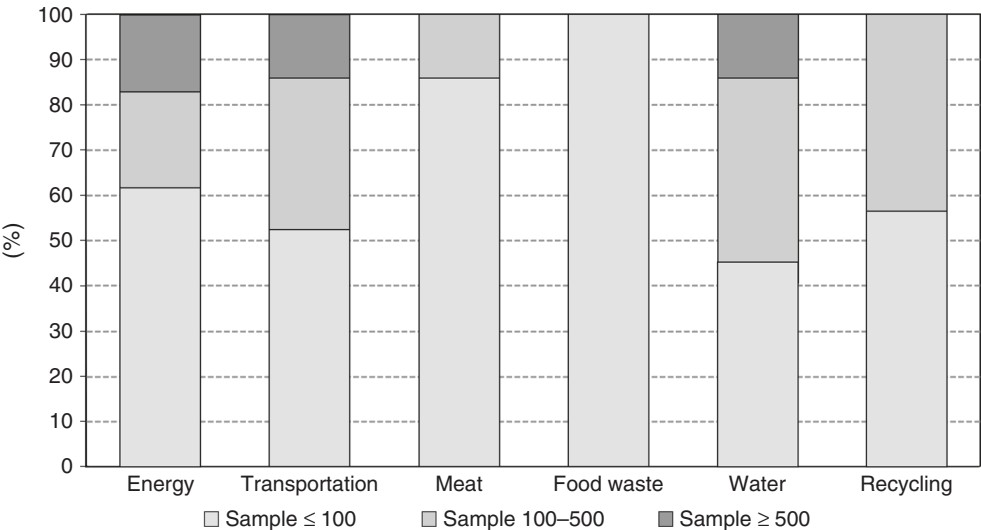

**Fig. 4** Sample size per experimental group in behavioural interventions targeting different types of behaviour. Shows how different household mitigation behaviours have been tested with interventions with a distinct number of observations

subset of papers examining how to reduce the use of multiple towels by hotel guests. This subset of papers reports the highest effect sizes ($d = -0.168$ 95% CI $-0.271$, $-0.064$) within this category with a POB of 11.9%. Removing these papers and analysing the impact of interventions specifically to reduce daily water use in households decreases the effect size to $d = -0.028$ (95% CI $-0.046$, $-0.010$) statistically homogeneous ($I^2 = 19.5\%$ ns). This corresponds to a POB of 2%.

Interventions to reduce food waste and animal products consumption have similar characteristics: limited number of studies with low between-study heterogeneity ($I^2 = 21.6\%$ and $I^2 = 36.8\%$ both ns), meaning that interventions targeting these behaviours tend to report statistically similar effects. Although the average magnitude of these effect sizes is higher compared to energy, transportation or water, these are marginally significant (food waste $d = -0.231$ 95% CI $-0.518$, 0.056 POB 16.3%, and animal products consumption $d = -0.239$ 95% CI $-2.81$, $-0.008$ POB 16.9%).

Lastly, recycling seems to be the behaviour more susceptible to change. Interventions aiming to increase recycling have, on average, the strongest effect sizes ($d = -0.457$) although highly heterogeneous ($I^2 = 69.9\%$). Individuals targeted with behavioural interventions in this context are expected, on average, to be about a third more likely to recycle than the individuals in the control group.

**Nudges and social comparison report the highest effects**. Information and social comparison are the most common strategies used in field experiments, both in terms of the number of studies published as well as the quantity of people reached. However, both show a very small impact. Information-based strategies (e.g., statistics, in-home displays, factual feedback, energy labels) has an expected POB of 3.4% ($d = -0.048$ 95% CI $-0.075$, $-0.021$). Strategies of social comparison, such as providing social references and peer comparisons, are slightly more effective ($d = -0.077$ 95% CI $-0.108$, $-0.046$; POB 5.4%). Nonetheless, both estimates are based on the largest RCTs included in our sample, suggesting that consistent (small) effects can be expected using these strategies, albeit seemingly more consistent when using information than social comparison ($I^2 = 34.7\%$ vs. $I^2 = 72.2\%$, respectively).

Appealing to the need for climate change mitigation or trying to increase people's engagement (either by goal-setting or

commitment to climate change mitigation) seem to produce higher average effect sizes ($d = -0.253$ to $d = -0.266$). Some caution is required, however, when interpreting these effects. Studies employing these strategies tend to be composed mostly of self-selected samples, which greatly inflates results. In particular, commitment strategies require, by definition, that people accept to commit to a goal, otherwise they are excluded from the study. Therefore, such average effect sizes are much higher ($d = -0.480$; POB 33.9%) than strategies that target the general population, with various levels of commitment or interest in climate change mitigation. Moreover, between-study heterogeneity is high both in interventions using appeals and engagement ($I^2 = 70.5\%$ and $I^2 = 71.8\%$ respectively).

The strongest effect is estimated for nudges or choice architecture, which is not only the highest effect size ($d = -0.352$; POB 25%) but also a robust calculation ($I^2 = 0\%$), and estimated from studies that tend to not rely on self-selected participants. Being exposed to interventions based on choice architecture increases the probability of mitigating climate change in about a quarter.

Up to this point we have examined average effect sizes per different moderator but many features co-occur between studies, which increases uncertainty about the main influences of the effect. While subgroup analysis does not establish causal effects, we attempted to disentangle the potential role of different moderators using a meta-regression model jointly testing all covariates, including sample size, self-selection, type of behaviour and type of intervention. The model [$F(11,131) = 6.70$ $p < 0.01$] showed that sample size, the interventions social comparison and nudges, and the behaviour recycling were the covariates explaining significant variations in the effect sizes. In Figs. 4 and 5 we explore sample size per behaviour and intervention. Energy, transportation and water were the only behaviours examined in very large studies which may explain the lowest effect sizes found (Fig. 4). Per intervention, nudges were a strategy only tested in small sample studies, which in turn may be related to its high effect size (Fig. 5).

The combined analysis of intervention per type of behaviour (Fig. 6) suggests an alternative explanation for the low effect sizes reported in energy and transportation, which could be attributed to the dominant use of information-based interventions. Other types of behaviours were also dominated by specific interventions, such as nudges for reduced meat consumption, social comparison for water saving, and psychological engagement, such as goal-

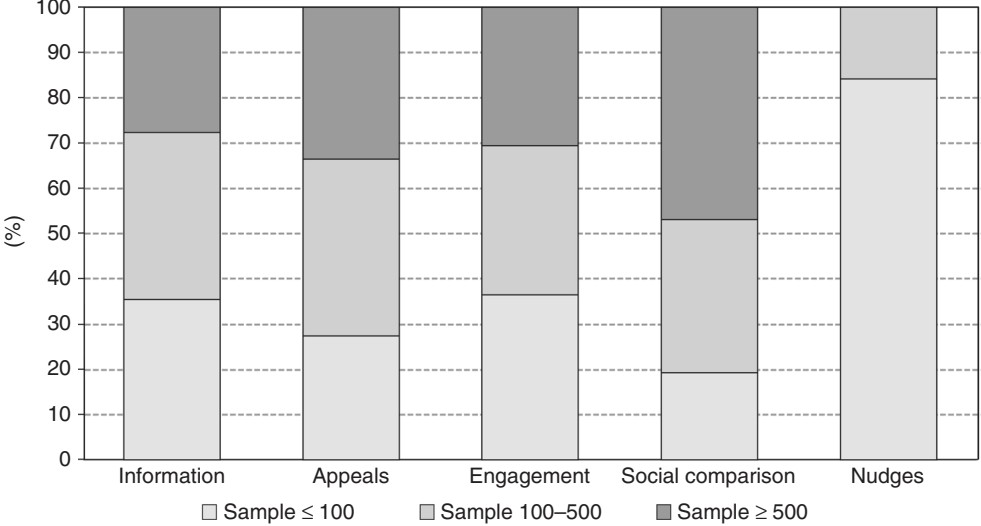

**Fig. 5** Sample size per experimental group in behavioural interventions using different strategies. Shows how different behavioural stimuli have been tested in interventions with a distinct number of observations

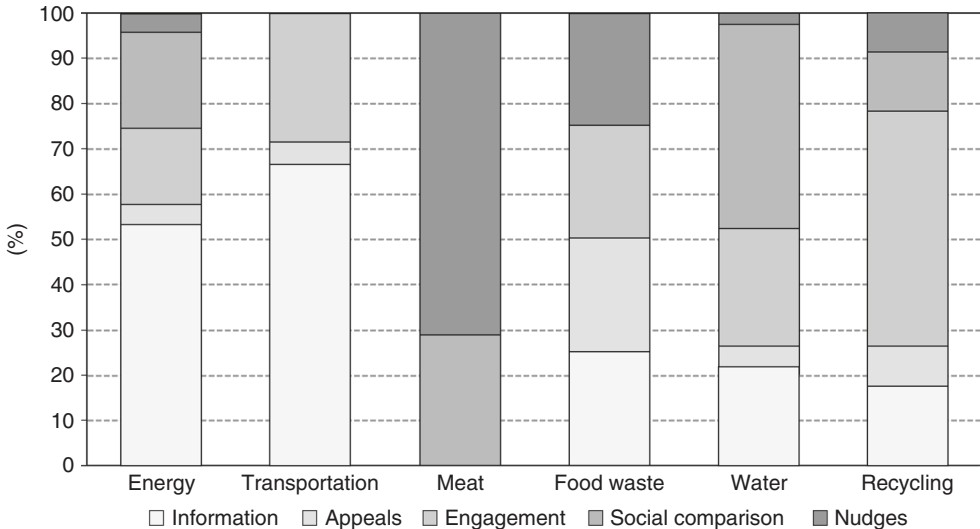

**Fig. 6** Type of intervention tested per mitigation behaviour. Shows how different mitigation behaviours have been targeted by distinct behavioural stimuli

setting, for recycling. The direction of causality is unclear, but these features tend to be associated and future research should unravel more precise sources of impact.

## Discussion

Taken in isolation, behavioural interventions have a very small positive effect on climate change mitigation behaviours while the intervention is in place. Once the intervention stops, there is no evidence that such interventions produce lasting positive changes. While the intervention lasts, the small overall effect size corresponds to the probability of such interventions promoting mitigation behaviours in 6.6%. This general estimate is unadjusted to several biases such as small samples and participant self-selection. These two forms of bias were shown instrumental to inflate average effect sizes. Restricting the analysis to more conservative and precise estimates from large samples and naïve subjects, the expected probability of change from behavioural interventions is reduced to 2–3%.

Our work indicates alarmingly low levels of behavioural plasticity: most household behaviours relevant to climate change exhibit resistance when targeted only by behavioural interventions. In particular, critical actions such as purchase of energy-efficient appliances and private car use were barely affected by the behavioural interventions examined. We compare selected estimates with Dietz et al.[18], who also estimated the behavioural plasticity of several similar behaviours. For instance, Dietz et al. estimated that, under the most effective interventions, an 80% behavioural plasticity could be achieved for the purchase of energy-efficient appliances. We report an average null effect size. For choices related to the private use of cars, Dietz et al.'s estimates are more conservative (15–25%) but still substantially higher than ours (2.5%).

There are some key differences between our work and Dietz et al.[18], which could account for these discrepancies. Firstly, Dietz et al. included interventions also using financial incentives. The discrepancies found may be due to a higher effectiveness of financial incentives. However, there is no clear way to establish

this hypothesis because Dietz et al. did not specify which interventions were used, under which circumstances and for which specific behaviours. Secondly, Dietz et al. based their estimates on proxies and, in some cases, less stringent research designs. We consider this an important difference compared to our methodological approach, which allowed us to estimate the behavioural plasticity of tangible behaviour in natural experimental settings. It is well-established that interventions based on non-experimental studies tend to produce higher effect sizes[15,17]. Thirdly, Dietz et al. based a significant part of their analysis on several high-impact one-time actions (e.g., home weatherisation, purchase of energy-efficient home appliances and fuel-efficient vehicles), whereas our work mostly examined frequently occurring behaviours (e.g., energy and water saving at home, recycling, food waste) because the available experimental field evidence mostly targeted frequently occurring behaviours. The possibility exists that single action behaviours may be more effectively influenced by behavioural interventions compared to recurring behaviours, which may be more resistant to change if automatised in habitual routines[32]. Although the roots for these discrepancies cannot be conclusively uncovered, our results raise the possibility that these past forecasts about how much can be achieved to reduce carbon emissions by household action could apply only to some specific best cases. This is a critical issue because Dietz et al. used their results to estimate reductions in carbon emissions over 10 years from household action. Our estimates about behavioural plasticity are more conservative than Dietz et al.'s. Future research should further address these inconsistencies because, even after a decade, Dietz et al. represent the benchmark estimates for the reduction in carbon emissions that could be expected from household action[1,3]—and these estimates are based on more optimistic levels of behavioural plasticity and specific mitigation behaviours compared to our work.

Thus, the available field experiments do not give rise to hopeful predictions about relying on behavioural interventions alone to tackle climate change. Our results show that it is unlikely that behavioural interventions, in isolation, will help us achieve large and pressing reductions in carbon emissions. There is, however, the possibility that behavioural effects could vary depending on the policy or economic forces concurrently in place, or that specific combinations of behavioural interventions with legislation and/or financial incentives produce better results[33–35]. Indeed, there is evidence strongly suggesting an interactive effect. For instance, financial incentive programmes for home insulation that required prior energy audits were less effective in promoting conservation actions compared to programmes that offered similar monetary incentives, but that did not involve requesting an audit[36,37]. Another example provided by Stern and colleagues[38] reported that the same programme to install energy-saving equipment in households produced very different uptake rates for energy audits (up to a 20% difference), depending on the source of the information letter received. These same authors reviewed multiple conservation programmes in the U.S. combining both financial and non-financial interventions and concluded that, controlling for type and size of incentives, participation in conservation programmes could vary by a factor of 10, depending on concomitant behavioural strategies, such as perceived social norms or message framing. This evidence suggests that financial incentives capture consumers' attention, after which behavioural interventions can have larger effects than if used in isolation. Our findings, in combination with this seminal evidence, suggest that researchers should focus more on interaction effects. An important issue to be further addressed is whether, and under what conditions, specific behavioural interventions can make a substantial difference when integrated with other types of interventions to increase adoption of actions with a high potential to

reduce carbon emissions. This is an important issue because behavioural interventions, like financial incentives and regulation, typically fall short of expectations derived from untested assumptions about behavioural plasticity. Interactions may be where the greatest potential lies.

The need for more investment in rigorous impact evaluation is pressing because the last decades have witnessed an unprecedented interest in climate change and thousands of papers have been published on this topic. The abundance of research may lead us to infer that much is already known about what promotes behavioural changes towards climate change mitigation. Yet, the reality is that high-quality field experiments, from which precise effects can be estimated, are scarce. Not only are high-quality field experiments rare but few set the specific goal to test interventions to mitigate climate change and fight global warming. It is noteworthy that most studies from which estimates about transportation, food waste and meat consumption could be extracted were not conducted to tackle climate change, but were borrowed from trials on health promotion, e.g., reduce meat consumption to increase vegetable intake; walk or cycle to work instead of using the car to reduce obesity; reduced food waste as a proxy for lower calories intake.

The fact that some estimates were borrowed from different academic fields suggests the benefits that could be achieved in future interdisciplinary collaboration. Conducting behavioural interventions from an interdisciplinary perspective is crucial particularly when implementing interventions that may combine different stimuli, e.g., public service announcements, social comparison and government action. Complex, multi-layered interventions from the standpoint of their content and implementation may require the joint work of multiple experts. Also, using advanced technological solutions for data collection may also be more effectively implemented when combining behavioural scientists with technical expertise such as engineers and computer scientists.

We propose six main recommendations for future (interdisciplinary) research examining complex issues such as global warming. First, more research is needed to tackle behaviours with high technical potential, that is, behaviours that greatly contribute to reducing carbon emissions. Comparative estimates[19] showed that, for the average household in North America and Europe, the actions with the largest potential for carbon emissions reduction are eating fewer animal products, reducing driving and flying, and saving household energy, particularly from heating and cooling. Programmes and initiatives targeting, for instance, recycling, turning off the lights after leaving a room, or unplugging battery chargers after use, also contribute to mitigate climate change, but have substantially lower marginal effects—even if the interventions could achieve 100% behavioural plasticity. Ideally, interventions worthier of implementation should both achieve high behavioural plasticity and target behaviours with high technical potential.

Second, future research should implement research designs that allow for causal inferences. Whereas all research designs are valid tools for knowledge production, establishing causality requires specific research designs. Not only RCTs can be used to achieve this, but they are the gold-standard to test causal effects. Future research should focus on establishing to what extent a particular intervention—and that intervention alone—contributes to behaviour change. This is established by a valid comparison group which has the same characteristics, on average, as the treatment group in the absence of the intervention, and remains unaffected by the intervention. Randomised assignment of participants is the most recommended procedure to achieve a valid comparison group. However, many real-world circumstances prevent this randomisation, but researchers can turn to

alternative impact evaluation methods[39], such as interrupted time series, instrumental variables (i.e., variables outside the control of individuals that influence participation in the intervention but are otherwise irrelevant), regression discontinuity design, differences-in-differences or matching (for when intervention assignment rules are not clear). These more pragmatic, quasi-experimental methods may be faster, cheaper and logistically more feasible than RCTs, and could provide valuable information[40,41]. Results should, nevertheless, be presented with due caution. Simple before-and-after analyses, or descriptive comparisons of programmes with different characteristics provide the least trustworthy estimates.

Third, future work should try to isolate the main effects of different behavioural stimuli (e.g., information, social comparison, environmental appeals), as well as the combined effect of several behavioural stimuli, or their combined use with incentives and regulations. The content of interventions needs to be tested more precisely because a large proportion of interventions to date implements bundles of stimuli from which the identification of the key driver of effectiveness is difficult to grasp. The implicit assumption behind combining stimuli is that the intervention is more likely to be effective, the larger the number of stimuli involved. But we should not automatically assume positive additive or multiplicative effects just because a variety of actions is taken in combination. It is crucial to estimate the additional marginal contribution of multiple actions to mitigate climate change. Conclusively establishing main versus interactive effects requires comparative analyses, specifically designed for this purpose. As this work attests, ample research has been conducted in the field randomly assigning individuals or households to different treatments. When such randomisation is possible, factorial designs are an adequate tool to test main versus interaction effects[42]. Testing these main versus interaction effects requires only minor changes to the study design and potentially an increase in sample size, mostly for discrete dependent variables[43]. However, when individual or household-level randomisation is not possible, cluster randomisation or quasi-experimental designs can still be conducted using a factorial design[44]. In more restrictive research contexts, researchers may use counterbalanced designs[45,46], in which different treatments are implemented stepwise. These treatments may include single stimulus and bundles of stimuli, introduced and/or removed sequentially[47]. Although the estimation of these effects may be more challenging and less precise than with a true experimental design, these quasi-experimental alternatives may still shed some light about the differential effects of single versus combined stimuli.

Fourth, researchers should continue to invest efforts in interventions targeting direct triggers of behaviour change, such as social comparison messages and choice architecture (nudges). Although such interventions have their own set of challenges (e.g., unknown collateral effects of downward social comparisons, or ethical concerns with consent and autonomy in nudge interventions), our results indicate these are the most effective interventions. For instance, choice architecture relies on people preferring the path of least resistance, such as having to walk less to the recycling bin or not giving a second look at the air conditioning temperature. This approach could be implemented in less explored areas. For instance, utilities companies could implement a default (opt-out) payment for renewable energy (e.g. refs. [48,49]). Moreover, the layout of cafeterias, restaurants and supermarkets could be rearranged to display animal products in less accessible areas. This is a strategy commonly used in health promotion to reduce consumption of sugary foods, but seldom used to fight climate change. Given the pressing need to change the consumption patterns of millions of individuals and households, interventions should target proximal drivers of behaviour

change— over and above the longstanding goal of changing hearts and minds. This is recommended because, conversely, traditional policy instruments such as public service announcements, statistics, consumption feedback, and energy labels, were found to be the least effective ways to produce behavioural change. The assumption about the positive role of information seems to die hard, despite consistent evidence of its low impact. Whereas this is an unsurprising result to behavioural scientists[25] and some results have pointed in this direction[7], there is a considerable gap between what is known in behavioural science and what is proposed and implemented by policy-makers. These strategies continue to dominate the proposed solutions in the latest Intergovernmental Panel on Climate Change (IPCC) Special Report with respect to demand-side behaviour changes[1], and the recently published randomised field trials persist on the idea of testing information-based strategies to promote mitigation behaviours (Supplementary References[3,18,48]). Information-based strategies are employed under the assumption that, when provided with facts and figures, people will adjust their behaviour and monitor their actions that produce carbon emissions[50,51]. Such assumptions are challenged by our results, estimating the probability that information can produce positive changes in 3.4%. This is a result of paramount significance considering the central role these strategies are expected to play in the fight against global warming[52,53]. Nonetheless, there is still the need to evaluate the conditions under which the effect of information strategies may be boosted, for instance when combined with financial incentives, or when delivered by trustworthy sources and easily available at the point of decision[34,36–38]. Whereas information has little impact to motivate unmotivated individuals, information may be crucial to guide behaviour once people are fuelled by complementary, more motivating strategies.

Fifth, more evidence is needed to identify which interventions can stand the test of time. Regarding whether behavioural interventions have persistent effects once the intervention is concluded, the available data showed an overall average null effect. We could not provide a definitive answer on persistent effects per specific type of intervention due to the small number of papers that reported follow-up effects. Our current recommendations for more investment in interventions with the highest behavioural plasticity potential (social comparison and nudges) are based on effect sizes while the intervention is in place. Future research may adjust these recommendations if evidence establishes that interventions with a more modest short-term impact maintain their effect in follow-up periods, whereas more impactful interventions may see their effect fade sharply after the intervention is concluded. Habit is one of the persistence pathways proposed to explain sustained effects[54]. Research on habit formation has shown that action repetition, particularly under the same environmental cues, is critical to establish new habits[55,56]. Therefore, the expectation could be that, the longer the intervention, the more likely it is to create a positive habit in recurring contexts. Yet, holding all factors constant, the intervention duration did not have an effect. This is puzzling, to some extent, because most household actions examined in our work are frequent and recurrent in stable settings (e.g., turning on the air conditioning at home, driving to work, selecting meal size and type). Our work included very heterogeneous research contexts and behaviours, possibly with different timescales required for change to happen, which may have masked the effect of intervention duration. Other persistence pathways include changing how people think (e.g., beliefs), changing future costs of mitigation behaviours (e.g., energy-efficient appliances), and external reinforcement (either social or financial). There is no conclusive answer on how to succeed in these pathways, or which one is more effective. Nonetheless, isolated examples show that evidence

for changing future costs is not compelling[57] but some interventions using social comparison (external reinforcement), and nudges (automatic habit reshape) have reported persistent effects. Ferraro and colleagues[5] report that the impact from a single social comparison treatment could be detected more than 2 years after this one-shot intervention (1.3% less water consumption in the treatment group). Allcott and Rogers[58] report that, after a 2-year intervention using social comparisons to reduce energy consumption, effects persisted when the intervention was discontinued, decaying at 10–20% per year. Regarding nudges, Kurz[59] showed that a 3-month layout change in a restaurant produced a persistent reduction in meat consumption by 4% in the following 3 months after the intervention ended. These examples suggest social comparison and nudges may be good options to transform transient behaviour changes into sustained behavioural patterns. Nonetheless, future research should establish more precisely when and why intervention effects persist.

Given that most behavioural interventions seem to be short-lived, incorporating multiple, sequential intervention components could be a promising solution. Relevant to this debate are the concepts of complex and adaptive interventions[60,61]—interventions composed of various interconnecting parts that emphasise the timing and sequence for different treatment(s)—as a way to adjust treatment response to circumvent saturation effects during implementation, and to promote sustained effects after implementation is concluded[60]. Applied to climate change research, an adaptive complex intervention could start with a social comparison message, after which it could shift to (or add) an information-based strategy, and conclude with a change to (or an extra added) environmental appeal to save the planet. However, a challenge for these sequential designs is what theoretical basis justifies the order selected and tipping point for adjustment[61]. In line with our previous arguments, a sequence of interventions should probably start with motivating, eye-catching strategies (e.g., social comparison, financial incentives), to subsequently move to (or add-on) more information-based interventions, once attention for climate change has been grabbed. The analysis of temporal dynamics in such cumulative designs could contribute both to providing evidence on whether interventions should be adaptive over time, along with contributing to our understanding of how stimuli bundle works (third recommendation).

Sixth, future research should enrol, to the extent possible, naïve participants. Our results showed a sevenfold difference between studies that required participants' explicit consent versus studies that had no such requirement. One explanation for this effect is that people that accept to enrol in environmental studies are more interested in climate change mitigation than people that decline such invitations. Moreover, participants that provide consent are aware that their behaviour is monitored, thus leading to greater desire to conform to what is normative[62]. Researchers may correct this bias by using publicly available data, requesting waived informed consent due to compelling validity concerns[63], monitoring non-participants, at least at the aggregate level, and/or correcting for the estimates collected with non-naïve participants. This is an important question because scaling-up and implementing these interventions in a larger scale, possibly designed as policy-level action, necessarily targets a much wider variety of individuals with varying motivation to act pro-environmentally. Using the estimates from self-selected participants to inform policy makers creates an expectation that may not be matched when targeting larger populational segments.

In summary, our work speaks to the preferred options within the universe of behavioural interventions. Based on the best evidence currently available, our recommendation is to increase the use of nudges and social comparison interventions, given that both types of interventions are independent of individuals'

motivation to mitigate climate change. Furthermore, we recommend, firstly, targeting the most impactful behaviours to achieve carbon reductions. Secondly, using RCTs or other research designs that allow to establish causality. Thirdly, measuring tangible behaviour changes, both in while the intervention lasts and after the intervention has been concluded (follow-up period). Lastly, testing the interventions in naïve participants. Taken together, our results should not be used to argue for a disinvestment in behavioural interventions because this work did not compare behavioural interventions with alternative approaches, nor did it examine their interactive effect with other strategies. Ultimately, our work shows that mitigating climate change via voluntary behavioural changes in households and individuals is a real challenge—not that behavioural interventions are less effective than alternative approaches.

Some limitations of our analysis should be discussed. Intrinsic to the nature of meta-analyses, results depend on the scope and quality of the available empirical evidence. Therefore, our results are as good as the raw materials available to inform our analysis. An additional limitation is related to the outcome standardisation as Cohen's $d$. This enabled us to compare the effect sizes of different interventions in different contexts but disallowed the translation of effect sizes into meaningful savings, such as kWh, liters, or kilograms. Although we cannot precise the carbon emission reductions derived from the effect sizes reported here, these are indubitably small, particularly in light of the ambitious goals set for carbon emissions reduction. Another limitation worth mentioning is the small number of contexts in which nudges have been tested, mostly targeting reduced meat consumption and food waste. Although we recommend this type of intervention due to the higher effect sizes reported, this strategy has seldom been tested in behaviours with high technical impact, such as energy consumption or private car use—where the greatest gains for carbon reductions can be achieved. A final limitation relates to our strict inclusion criteria may have excluded important behaviours. For instance, there were no RCTs testing behavioural interventions to reduce air travel, increase the use of renewable energy or purchase of electric cars. There are, nonetheless, multiple studies addressing these topics using non-experimental designs, self-reported attitudes or simulating the effects of implementing government subsidies. Such papers were excluded due to our methodological choices. While our inclusion criteria may have restricted or limited the behaviours examined, it nevertheless increased the reliability and ecological validity of the estimates that we were able to calculate—which were sizeable in terms of sample size, behaviours and interventions examined, and total number of papers.

## Materials and methods

**Literature search and data extraction**. This meta-analysis followed the procedures and guidelines established by the Cochrane Group[64] and the PRISMA Statement[65] (Supplementary Table 1). We conducted electronic and manual searches from inception to 30 June 2018 in the main international databases (EBSCO Business Source Complete, EconLit, PsycNET, JSTOR) as well as unpublished grey literature and Google Scholar. References from previous reviews were hand-searched. The full search strategy is presented here: [((Energy OR electricity OR gas OR towel OR water OR heating OR cooling OR drying OR Food OR meal OR Meat OR beef OR dairy OR (air travel) OR fly OR car OR vehicle OR airline) AND ((((reduc* OR decreas* OR low* OR mitigat* OR limit* OR control) AND (wast* OR excess OR discart* OR surplus OR leftover OR consum* OR use OR eat OR purchas* OR demand OR driv*)) OR ((Recycl* OR reuse OR ((waste OR garbage OR trash OR plastic OR cart OR can OR glass OR paper) AND sort*)) AND ((public transport*) OR (bike OR bicycle OR bus OR subway OR (car pool*) OR (trip chain) OR (car shar*) OR ((fuel efficien* OR electric) AND (car OR vehicle))) AND (Efficien* OR (energy saving)) AND (appliance OR fridge OR dryer OR (air conditioning)) AND (Vegetarian OR vegan) AND ((increas* OR improv* OR promot* OR foster) AND (efficien* OR sav*)))] AND (experiment OR (field stud*) OR intervention OR (random AND (select OR allocat*)) OR (control group) OR (comparison group)))]

The flow diagram (Supplementary Fig. 1) depicts the flow of information through the different phases of our systematic review. Initial searches held 29,820 electronic records. After duplicates were removed ($N = 13,562$), at a first stage of screening, the criterion of experimental design eliminated 6376 records, whereas the criterion of observed behavioural outcomes removed an additional 8593 records. A more extensive screening was performed on a working sample of 1289 papers. A final selection at this stage was based on removing quasi-experimental or pretest–posttest studies without a control group, papers without proper statistics reported and studies reporting group-level changes (e.g., per buildings, student halls, employee departments) without any link to individual-level behaviour changes.

The final sample according to the inclusion criteria covered 83 studies, providing a total of 157 estimates. Not all estimates were unique, meaning several originated from the same dataset which violates independence of observations and induces same data/author bias. These 157 estimates were used only when subgroup comparisons were made. For instance, if the same study reported estimates on the effect of an intervention in walking, cycling and car use, only a randomly selected estimate was used to estimate the global effect for transportation but they were all used if the goal was to estimate the specific effect in walking or cycling). Only 144 estimates are unique, that is, from independent datasets. These final 144 estimates are publicly available[66] and the papers from which these estimates were extracted are listed in Supplementary References.

Three reviewers (lead author and two research assistants) independently assessed initial titles and abstracts, which were excluded only if all reviewers agreed that the trial did not meet eligibility criteria. Full-text articles of remaining citations were retrieved and assessed for inclusion by all reviewers. Disagreements were resolved by consensus. Results published only in abstract form were not included unless adequate details were available for quality assessment. We designed and used a structured data abstraction form. The reviewers abstracted data from each study and evaluated the completeness of the data for extraction. Differences in quality ratings were resolved by discussion or by involving an additional reviewer. The following data were extracted: study, country of origin, setting, type of intervention, target population, sample size and participant self-selection (Supplementary Data 1).

**Data standardisation.** The dependent variables under analysis were objective behavioural changes measured as factual variations, for instance, in energy usage (kilowatts [kW]), water usage (litres), quantity of recycled materials or food wasted (in kgs). (This criterion was weakened in the context of transportation trials. Several transportation trials did not directly measure behaviour but have higher-quality proxies than other contexts, such as diaries with exact number of kilometres driven or number of days per week that car rides were replaced by public transportation.) Given the large variation in outcomes reported between studies (e.g., frequencies, means and standard deviations, regression coefficients) we converted all measurements to standardise mean differences (SMD) or Cohen's $d$. SMD converts the results of multiple studies into a uniform scale before these can be combined, expressing the size of the intervention effect in each study relative to its observed variability. The average estimates presented are negative because we estimated how much an intervention reduces the use of natural resources (e.g., reduced energy and water consumption or food waste). We multiplied by $-1$ the estimates in the few cases where an increase is beneficial (e.g., recycling or purchase or efficient appliances) to ensure that all measurements are in the same direction[31]. Cluster randomised studies, i.e., with randomisation performed at the group-level, were adjusted for an 'inflation factor' (IF)[67]. For trials randomising clusters of average size $m$ to two or more intervention groups, this factor can be estimated approximately by $IF = 1 + (m-1)\rho$, where $\rho$ is the sample estimate of the intracluster correlation. The effective sample size for the trial is then given approximately by $N/IF$, where $N$ is the total number of subjects enroled.

We used the Effect Size Calculator from the Campbell Collaboration for the conversion and standardisation of the dependent variables. Only the POB was calculated manually, estimated from $SMD/\sqrt{2}$[22]. The Campbell Collaboration https://www.campbellcollaboration.org/ is a leading organisation for evidence synthesis and meta-analysis production. This organisation developed a tool to support scholars and practitioners that run complex meta-analysis, which requires outcome conversions and standardisation. This tool is the Effect Size Calculator https://www.campbellcollaboration.org/research-resources/research-for-resources/effect-size-calculator.html.

The procedure was as follows: for each individual paper in our sample, we extracted their reported outcomes—typically reported as $t$-test, regression coefficients + SE, means + SDs, percentages, etc. We inputted these estimates plus the sample size for each individual study in the calculator. The calculator then returns an estimate for Cohen's $d$, its variance and the 95% CI. These are then the estimates transferred to STATA to perform the meta-analysis. We present the sample sizes per condition in the columns N_EG (sample size for the experimental group) and N_CG (sample size for the control group) in the STATA database (Supplementary Data 1).

**Data analysis.** We conducted a random-effects meta-analysis, pooling study-level data using STATA IC14 (*metaan command*)[68] (These analyses can be replicated in R using metagen or rma (see "Code availability" statement below). We used the most common approach to modelling between-study variance, the DerSimonian–Laird (DL) model. *Metaan dl* assumes that the true effect can be different for each study and that the individual-study effects are distributed with a variance $T^2$ around an overall true effect. Summary estimates are reported with 95% CIs and stratified by different interventions. Statistical heterogeneity was assessed with the $I^2$ statistic[28], which describes the percentage of variation across studies that is due to heterogeneity rather than chance. It is an intuitive and simple expression of the inconsistency of studies' results. A confidence interval for $I^2$ is constructed by *metaan* using the iterative non-central chi-squared distribution method of Hedges and Piggott (2001) described in[64].

Meta-regression models[29] were estimated to examine the effect of key moderator variables (*metareg command*). Publication bias was visually explored using funnel plots[69] (*confunnel command*) and the Egger´s test (*metabias command*)[31]. A funnel plot is a scatterplot used for detecting systematic heterogeneity by presenting the distribution of treatment effect against a precision measure (standard error). To statistically assess the role of small sample studies in potentially biasing overall estimates of effectiveness, we conducted the Egger´s test which regresses the intervention effect estimates on their standard errors.

Cumulative meta-analysis was performed using the *metacum command*[70]. In cumulative meta-analysis, the pooled estimate of the treatment effect is updated each time the results of a new study are included. This makes it possible to track the accumulation of evidence on the effect of interventions over time. The circles in the plot represent the cumulative effect size at any given moment, i.e., the average effect size resulting from the inclusion of studies up to a certain point. There is not an 'average' effect size per year, but each individual study is ordered and added by year of publication. Cumulative meta-analysis is not additive, i.e., it is not the sum of individual studies. Meta-analytic procedures are weighted averages and thus cumulative meta-analysis is an updated average effect size over time.

## Data availability

The authors declare that the data supporting the findings of this study are available within the paper and its supplementary information files. We provide a Source Data File that is publicly available at Figshare [https://doi.org/10.6084/m9.figshare.9641999]. This Source Data provides all underlying data for Table 1 and Figs. 1–6.

## Code availability

All commands used in STATA are publicly available:
   metaan https://www.stata-journal.com/article.html?article = st0201
   metareg https://www.stata-journal.com/article.html?article = sbe23_1
   metabias https://www.stata-journal.com/sjpdf.html?articlenum = sbe19_6
   confunnel https://www.stata-journal.com/sjpdf.html?articlenum = gr0033
   metacum http://fmwww.bc.edu/RePEc/bocode/m/metacum.html
   These analyses can be replicated in R using metagen or rma, and these R codes are publicly available at Figshare [https://doi.org/10.6084/m9.figshare.9641999].

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

## Acknowledgements

This work was funded by New York University Internal Funding—University Research Challenge Fund (URCF) (7271260ADHPGRA300). The first author acknowledges the

funding and extensive support of ETH-Zurich and Future Resilient Systems (FRS) at the Singapore-ETH Center (SEC) in early stages of this work (2015–2016), namely Professors Renate Schubert and Hans Heinimann, Director Remo Burkhard, Martina Cecchini, Jiang Zhengyi, Xing Zhang, Yunfeng Lu and Aaron Ang. We acknowledge the support and collaboration from Dr. David Holland and the Center for Global Sea Level Change.

## Author contributions

C.F.N. was involved in all stages of this work, from inception, data search and extraction, data analysis and manuscript preparation. J.J.B. and B.M.S. contributed to manuscript preparation and submission. D.G.F. contributed to data analysis, namely the identification and analysis of the R codes equivalent to the STATA commands used, and figure editing.

## Competing interests

The authors declare no competing interests.
