## [Peer Review File · Nature Communications]

Reviewers' comments:

Reviewer #1 (Remarks to the Author):

This paper stands out positively for its apparent methodological rigor, but also negatively for its conceptual shortcomings. I first discuss methodology, but only briefly. I hope other reviewers will have more to say in detail about the statistical techniques used for data analysis.

Regarding methods, the focus only on controlled trials with behavioral outcomes measured is understandable for increasing rigor of analysis. It does provide rigor, presuming the analysis has been done well, but the inferences to be drawn from the analysis need to be thought through very carefully. My comments focus mainly on issues of conceptualization of the variables used in the analysis and on the relevance of the findings to efforts to intervene to make significant contributions to mitigating climate change or other undesirable environmental outcomes.

Regarding conceptualization, the paper is not clear or consistent about the domain of the review in terms of the outcome variables covered. The title indicates that the outcome variable of concern is human impacts on "climate change" but the abstract says it is about "sustainability" and "sustainable behaviours". These are different things. For example, reduced water consumption promotes environmental quality and sustainability in some respects, but its impact on climate change is unclear, and probably minimal, particularly in relation to other individual and household behaviors, such as the purchase of energy-efficient appliances or automobiles to the extent that this equipment is powered by fossil fuels. I would say that the review's focus, in terms of outcome variables, is on environmentally significant behaviors at the household level, and within that, almost entirely on behaviors that involve the use of existing household equipment (what the abstract calls "lifestyle interventions"), rather than, for instance, on behaviors with longer-lasting effects, such as purchases of appliances or automobiles, choice of home size and location in relation to public transportation, childbearing choices, or individual influences on public policy or the choices of organizations in which individuals may have influence (see, for example, Table 1 in ref. 3 of the submission). Thus, the paper's conclusions do not apply to all "sustainable behaviors" but rather to a subset for which available studies meet the authors' methodological criteria. If this is not made clearer, readers are likely to misinterpret the importance of the results. This is particularly the case because the longer-lasting kinds of behavior, for which evidence is barely reviewed here, have greater potential impact on the environment than the kinds of behavior that have produced studies meeting the paper's methodological criteria. The "effectiveness" of interventions can be assessed in terms of behavioral change or in terms of environmental impact, which can be expressed as the product of technical potential (TP, or the impact that would be achieved if the behavior were universally adopted) and behavioral plasticity (BP, or the proportion of universal adoption that can be achieved). This paper focuses only on BP. It does a rigorous analysis of this, for the behaviors on

which studies available studies meet its methodological criteria and for the efforts used in that work to change behavior.

Regarding the conceptualization of “behavioural interventions”, it seems clear from the categorization of interventions, that among the interventions that are excluded from this review are regulations, government policies, financial incentives, energy or equipment prices, and the like, all of which affect behavior. This is an understandable choice, but by making this choice, the author(s) make it impossible to consider the likelihood that the interventions studied may have greatly different effects depending on the policy or economic forces also affecting decisions. The study investigates the roles of several “moderator variables” but not of policy or economic factors (except for the comparison across geographical regions, which surely includes policy and economic influences but does not disaggregate them). Past research indicates that the effects of particular kinds of “behavioural” interventions can be strongly influenced by other factors in place when the interventions area introduced. Moreover, the most effective interventions, in terms of TP x BP, are those that integrate interventions of different types. For instance, to quote Dietz et al. (2009), “the most effective documented weatherization programs... have combined financial incentives (grants or rebates covering most of the retrofit cost), convenience features (e.g., one-stop shopping), quality assurance (e.g., certification for contractors, inspection of work), and strong social marketing. The highest recorded plasticity is 85% over 27 months,” which was recorded in the Hood River (Oregon, USA) Conservation Project in the early 1980s. And weatherization has much higher TP x BP than almost all the behaviors for which studies met this paper’s methodological criteria.

This paper says that Dietz et al (2009) used their estimates of behavioral plasticity to predict reductions in carbon emissions over 10 years and that the 2018 IPCC Special Report used those numbers to estimate emissions reductions from “behavior change interventions.” In fact, Dietz et al used their BP estimates to point to “Reasonably Achievable Emissions Reductions”, that is, reductions that could be achieved if the best documented and feasible known interventions were used. In other words, this was a best-case estimate, not a prediction. If the IPCC report took those estimates as predictions, it misread the Dietz et al. paper. The BP estimates in Dietz et al. should not be taken as predictions. I see this paper as offering documentation that the studies meeting this paper’s methodological criteria do not by and large represent best practice. The great variation in effect size reported in this paper indicates that much remains unknown about how to achieve higher levels of BP, including levels that Dietz et al. considered reasonably achievable. This is an area where more research is much needed, especially if it is focused on behaviors with high TP.

I recognize that it is difficult to conduct randomized trials of interventions to affect high-TP household behaviors, especially because most of these behaviors are infrequent. It is especially difficult if the interventions most worthy of examination integrate financial and regulatory incentives with “behavioural” influences, as was the case in the Hood River project. This likely explains the relative absence of such studies from those reviewed here. It is also difficult to do rigorous evaluation when so many interacting variables may be in play. These difficulties do not invalidate the

analyses done here, but they do affect the conclusions that can be drawn from them. This review appears to indicate that, measured in terms of degree of BP, the interventions studied, taken in isolation from other variables with which they are likely to interact, have very limited effects that are not long-lasting.

What does this mean in terms of publishing the paper? Presuming that the statistical analyses hold up to scrutiny from reviewers who know these methods better than I do, it means that the analyses deserve publication (although I'm not sure whether a communications journal is the right venue) but that before any publication, the authors' claims for their practical implications need to be considerably revised and toned down to be defensible. In my view, the data do not indicate that "predictions about how much carbon emissions can be reduced by targeting demand-side behaviour changes" are necessarily inflated, because the work meeting the criteria for inclusion in this review represent a biased sample of efforts to change household demand behavior—especially considering that such efforts include regulatory and financial incentive-based interventions, as well as combinations of these with what this paper refers to as "behavioural" ones. The paper indicates that efforts to target demand via the intervention types reviewed, alone and not in combination with other intervention strategies, have small effects, so should not be expected, on their own, to have major effects in the near term on the behaviors examined in this review. One last point: the paper doesn't really speak in any direct way to "carbon emissions." Many of the outcome variables in the studies reviewed, such as water consumption and food waste disposal, have little or nothing to do with carbon emissions. The paper would be more on target (if perhaps less interesting to IPCC) if it identified the outcome variables in terms of direct household demands on environmental resources rather than on carbon emissions.

Reviewer #2 (Remarks to the Author):

The paper clearly outlines and summarizes the limitations of existing behavioral interventions targeting at improving sustainable/pro-environmental behaviors. These are major pitfalls encountered across multiple research disciplines (e.g., ranging from psychology, behavioral science to preventive medicine). The paper is methodologically sound, and takes the "intervention content" into account (i.e., strategies employed to change people's behaviors), which tends to be a missing piece in most meta-analysis papers.

The paper also provides substantial and novel insights regarding the missing pieces and challenges of behavioral interventions for sustainability. These include but are not limited to the following: (1) the limited number of interventions targeting at food waste & demand for livestock products tend to be

health promotion studies instead of aiming at improving sustainability. (2) Most sustainable behaviors display low behavioral plasticity, except recycling behaviors. (3) Information-based and social influence strategies are dominant, however, they both show a small impact. (4) Most interventions use multiple stimuli in one study, making it difficult to identify the main driver of behavior change.

Assuming the goal of this paper is to shorten the gap between empirical evidence and existing policy instruments (the so-called translational research), the paper can be a lot stronger if the policy (and/or theoretical) implications are provided. The Discussions section clearly outlines research challenges and gaps, however, discussion regarding what we should do next as researchers and policy-makers is missing in the paper. For instance, knowing that studies composed mostly of self-selected samples tend to greatly inflate the intervention results, how should we better approach study recruitment? Some of these issues are study limitations while some are not (meaning can be improved). Teasing these out is important to advance behavioral intervention research.

Additionally, I think the author(s) can touch base a bit on the instruments used in the intervention to measure behavioral changes – how these contribute to or weaken the intervention effectiveness. While looking through the table of included papers (SI.2), it appears none of the interventions adopts technology that captures objective behavioral data, such as wearable trackers. I wonder if this type of study tends to be ruled out during the eligibility assessment process. If yes, then it's hard to say that "previous work has overestimated the impact of lifestyle interventions that promote sustainable behaviors", and that it can be mentioned in the study limitations. In fact, it's also important to note that research employing objective behavioral tracking/monitoring technology could be less methodologically rigorous (relatively speaking), yet whether and to what extent its behavioral outcomes are sustaining compared to information and social influence strategies needs to be investigated. The paper can also benefit the audience by discussing the future directions of conducting behavioral interventions from an interdisciplinary angle.

Reviewer #3 (Remarks to the Author):

This paper addresses a potentially important point. There is a lot of discussion about how we might attempt to 'nudge' individuals and households (or firms even) into following pro-environmental behaviors. However, my own take on the overall corpus of research in this area is that there is increasingly stronger evidence that it is 'working'. You present a contrasting point of view, which is worthy of recognition, so long as the analysis holds up to scrutiny. However, in this matter I have several reservations so far and I require some issues on the analysis to be addressed.

In short, in my attempts to replicate your main results (using R, not STATA) I have encountered some differences. The first, and probably most important is that the sample sizes for the 'individuals' sub-sample are not a perfect match. Even making this comparison was time-consuming, since you in fact did not include N as a variable in your .dta file. I therefore had to manually enter, check and re-check these from the table in the paper. Given your discussions around the importance of sample size I was surprised about this (and even a little apprehensive as to why you had removed it from the STATA .dta file.) Please be sure to provide ALL variables in a future version of the data.

From my estimates $N = 2,376,336$ but from your paper you claim that $N = 2,364,924$, giving a difference of 11,412, that I have so far been unable to reconcile through a number of attempts. For the households there are 725,904 cases (we arrive at the same numbers here), but then in row 1 of Table 1 the paper refers to a total $N = 2,984,159$, the difference being explainable by using the value of household- $N = 619,235$, yet without clarity over where the remaining 106,669 households went to (comparing again to the number in the abstract of 725,904). This leaves open two possibilities either (i) that I am making some trivial mistakes in counting the cases which I would presume to be due to some steps in the process you take needing to be made a little clearer, or (ii) that there are errors. What is required of me at this stage is to verify that (ii) is not the case.

I therefore, at this stage of review process present a direct request that you politely double check the numbers, but to also remove the possibility that I am taking different steps than you are, to provide the replication file itself (a .do file I assume) so that I can verify the steps taken.

I would further add that I attempted to further replicate all remaining analysis in the paper and obtained some different conclusions. Some of the differences are immaterial. For example, my own funnel plots place fewer observations outside of the confidence interval boundaries, but otherwise the core conclusion holds. However other differences are more material. For example, I cannot get close the main Cohen's d estimates discussed from the final paragraph of page 3 onwards. Where you claim small effects, I am finding it difficult to obtain anything other than medium or large effects, unless I take steps which don't follow the applied method, as I understand it from your provided description. I do not provide my analysis here for the simple reason that, as described above, I cannot yet even reconcile the descriptive statistics from the analysis and so subsequent analysis would of course differ. Maybe in solving that primary concern, the secondary concern over the estimates can also be alleviated.

Regarding the subject matter, and phrasing of the results and contribution, I think it would be wise to contrast against the growing literature on 'nudges'. This would seem necessary, since you need in my opinion to offer more direct qualification that your results challenge or contest our 'known' understandings, once we condition on all the things we 'know'. I wonder for example if research on the use of in-home smart-meters or peer-benchmarked utility bills are reflected in your analysis or final conclusions. I provide a link to an industry report that points more towards the sort of areas in

my mind:

<https://www.mckinsey.com/~media/mckinsey/industries/electric%20power%20and%20natural%20gas/our%20insights/giving%20us%20energy%20efficiency%20a%20jolt/sizing%20the%20potential%20of%20behavioral%20energy%20efficiency%20initiatives%20in%20the%20us%20residential%20market.ashx>

General searching around goal-setting and social-comparisons might be worth considering to uncover relevant academic literature.

In the paper you refer to the idea of short-term effects, yet do not in my mind clearly qualify what you mean by short term. After several re-reads I fell on the conclusion that by short term you mean while an intervention is in force, and that no long term effect refers to the idea that behavior reverses once the intervention is removed. But if an intervention is in force for a decade – could we still fairly deem this as short term? Is my inference right about how you define short-term? (I am fine if this is the case, but in my own primary discipline we often take a different view of what long-term and short-term mean).

As a note, in the introduction you imply that higher prices for water and energy are suitable. While this would effect change at the margins, there must be some concern that these effects would be most heavily felt among the lower income component of society and that they might be contradictory to economic growth, or rather development, aspirations.

There is a paper identified in the dataset, that is not included in the list of papers contained in the manuscript. Why? You have no available N for the paper by Hongback et al (2016). Note that this would only increase the sample sizes further, and would therefore be unable to explain why my calculated N is greater than that reported in the paper.

The formatting on Fig. 2 seems to need attention. First, the bottom part of 2017 is cut-off on the y-axis. Since this is a time-series, why not reverse the axis, as is a more typical convention when plotting time series data. And what do the circles on the plot represent? Intuitively in my mind there would be one circle per year reflecting the 10 year 'cumulative' effect but there are many more circles than this. Lastly, are these really cumulative plots? If you sum up the numbers in any given year they end up much larger than the numbers reported here. Can you please help clarify? My mean/median calculations are closer to your graph (though still not the same, with less volatility at the early part of the sample, and a larger sustained mean towards the end).

I cannot say that the above will be all the comments I would raise on the paper, but given my current concerns regarding replicability, they are I think sufficient enough at this stage. I hope that you are able to attend to these points.

Response to Reviewers

Reviewer #1:

Comment #1 (Reviewer #1): *Regarding conceptualization, the paper is not clear or consistent about the domain of the review in terms of the outcome variables covered. The title indicates that the outcome variable of concern is human impacts on “climate change” but the abstract says it is about “sustainability” and “sustainable behaviours”. These are different things. For example, reduced water consumption promotes environmental quality and sustainability in some respects, but its impact on climate change is unclear, and probably minimal, particularly in relation to other individual and household behaviors, such as the purchase of energy-efficient appliances or automobiles to the extent that this equipment is powered by fossil fuels. I would say that the review’s focus, in terms of outcome variables, is on environmentally significant behaviors at the household level, and within that, almost entirely on behaviors that involve the use of existing household equipment (what the abstract calls “lifestyle interventions”), rather than, for instance, on behaviors with longer-lasting effects, such as purchases of appliances or automobiles, choice of home size and location in relation to public transportation, childbearing choices, or individual influences on public policy or the choices of organizations in which individuals may have influence (see, for example, Table 1 in ref. 3 of the submission). Thus, the paper’s conclusions do not apply to all “sustainable behaviors” but rather to a subset for which available studies meet the authors’ methodological criteria. If this is not made clearer, readers are likely to misinterpret the importance of the results. This is particularly the case because the longer-lasting kinds of behavior, for which evidence is barely reviewed here, have greater potential impact on the environment than the kinds of behavior that have produced studies meeting the paper’s methodological criteria.*

Reply to Comment #1 (Reviewer #1): This is an important comment about the conceptual scope of our paper and we sincerely thank the reviewer for the possibility to clarify this issue. The reviewer makes the point that there is a difference between climate change mitigation (which the reviewer accepts to include energy demand and transportation) and broader sustainable behaviours, which would include water demand and recycling.

A response to this comment requires some discussion about definition and conceptual boundaries. Sustainability conceptualizations vary substantially but these typically encompass some notion of human survival on a global scale requiring certain basic support systems, which can be maintained only with a healthy environment and a stable human population (Brown et al 1987; Santillo 2007; Morelli 2011). Climate change definitions have evolved from an initial scope bordering on sustainability¹, to a later focus on mitigating greenhouse gases emissions², and recently towards an

¹ Early seminal definitions of climate change were not markedly different from sustainability. The first World Climate Conference in 1979, proposed a definition, unanimously adopted by the participants, that framed human induced climate change as a broad and major environmental problem involving various dimensions of conservation <https://unesdoc.unesco.org/ark:/48223/pf0000037648>: “The long term survival of mankind depends on achieving a harmony between society and nature. The climate is but one characteristic of our environment that needs to be wisely utilized. Degradation of the environment in any national or geographical area must be a major concern of society because it may influence climate elsewhere. The nations of the world must work together to preserve the fertility of the soils; to avoid misuse of the world's water resources, forests and rangelands; to arrest desertification; and to lessen pollution at the atmosphere and the oceans. These actions by nations will require great determination and adequate material resources, and they will be meaningful only in a world at peace”. This definition would not exclude water saving (“avoid misuse of the world's water resources”) and recycling (e.g., “lessen pollution at (...) the oceans”) from climate change-relevant behaviours.

² Later definitions, such as from the United Nations Framework Convention on Climate Change (UNFCCC) on 1992, describe climate change as an environmental degradation problem and focuses on measures to mitigate greenhouse gas (GHG) emissions: [Article 2 “The ultimate objective of this Convention and any related legal instruments that the Conference of the Parties may adopt is to achieve (...) stabilization of greenhouse gas concentrations in the atmosphere at a level that would prevent dangerous anthropogenic interference with the climate system”]. The same rationale for mitigation of carbon emissions is respected by the Kyoto Protocol. This standpoint on climate change implies the behaviours more directly related

integration of both mitigation and adaptation strategies³. Please note full discussion of this evolution in the footnotes.

We acknowledge we have used climate change and sustainability interchangeably given some overlap in their conceptual boundaries. Nonetheless, we concur these are different concepts and we appreciate the opportunity for clarification. Our conceptual scope is restricted within the boundaries of climate change mitigation i.e., behaviours that help reducing carbon emissions. Not sustainable behaviours more broadly defined, and not behaviours related to adaptation strategies. In the revised version of our paper, we have now more clearly stated our conceptual scope: the effects of behavioural interventions to promote household action on climate change mitigation. We have replaced all passages using terms related to sustainability with terms related to climate change mitigation (e.g., Title, abstract; Introduction; Research questions; Results and Discussion sections) – please note track changes throughout the paper.

We reiterate our defence for including water consumption and recycling as two behaviours relevant to mitigation strategies. Household actions that consume the largest amounts of water (showers and dishes/ clothes washing) are also actions that consume large amounts of energy, and thus saving water saves energy (US Department of Energy 2006, 2016). Recycling also contributes to reducing emissions from resources extraction, landfill methane and waste incineration (US Environmental Protection Agency [EPA] 2016). We present this rationale now at the end of page 2 and page 3 – please note track changes detailing an improved rationale for the behaviours selected.

The reviewer, citing Stern et al. (2016), mentions that actions such as “*purchases of appliances of automobiles, choice of home size and location in relation to public transportation, childbearing choices, or individual influences on public policy or the choices of organizations in which individuals may have influence*” have more ‘*longer-lasting effects*’ and are potentially more relevant. However, recent estimates (Lacroix 2018) actually showed that, for the average households in North America and Europe, the actions with the largest potential for carbon emissions reduction are eating fewer animal products and having more fuel saving driving habits. All other household actions that also contribute to climate change have substantially lower marginal effects. Furthermore, much of the focus in Stern et al. (2016) was in the actions taken by organizations, communities, or government institutions, while our work is strictly centred on individuals and household actions.

Lastly, the reviewer considers that we only examined behaviours “*for which available studies meet the authors’ methodological criteria*”. This is correct. For instance, there were no randomised controlled trials testing behavioural interventions to reduce air travel, increase the use of renewable energy or purchase of electric cars. There are, nonetheless, multiple studies addressing these topics using non-experimental designs, self-reported attitudes or simulating the effects of implementing government subsidies. Such papers were excluded due to our methodological choices. While our methodological approach may have restricted or limited the behaviours examined, it nevertheless increased the internal and external validity of the estimates that we were able to calculate – which were sizeable in terms of sample size, behaviours and interventions examined, and total number of papers.

We consider our response in this last paragraph to be crucial to improve the strength of our results and conclusions. Therefore, we now include this rationale in our Discussion section (track changes in page 11; second paragraph from the bottom of the page).

Brown, B. J., Hanson, M. E., Liverman, D. M., & Merideth, R. W. (1987). Global sustainability: Toward definition. *Environmental Management*, 11(6), 713-719.

Gupta, J. (2010). A history of international climate change policy. *Wiley Interdisciplinary Reviews: Climate Change*, 1(5), 636-653.

to carbon emissions production are the behaviours more relevant to tackle climate change. This would include all behaviours requiring burning coal, oil or gas (producing carbon dioxide and nitrous oxide) or consumption of livestock farming products (producing methane).

³ More recent understandings of climate change include both mitigation and adaptation strategies (Shipper 2006), noticeable in the Paris Agreement https://unfccc.int/sites/default/files/english_paris_agreement.pdf. Thus, according to this definition, behaviours such as building flood defences, installing water-permeable pavements, cultivating resilient crop varieties or planting trees could also be defended as closely related to climate change, and as such, relevant to our paper.

- Lacroix, K. (2018). Comparing the relative mitigation potential of individual pro-environmental behaviors. *Journal of Cleaner Production*, 195(10), 1398-1407.
- Matos, C., Briga-Sá, A., Bentes, I., Faria, D., & Pereira, S. (2017). In situ evaluation of water and energy consumptions at the end use level: The influence of flow reducers and temperature in baths. *Science of the Total Environment*, 586, 536-541.
- Morelli, J. (2011). Environmental sustainability: A definition for environmental professionals. *Journal of Environmental Sustainability*, 1(1), 1-9.
- Santillo, D. (2007). Reclaiming the Definition of Sustainability. *Environmental Science and Pollution Research-International*, 14(1), 60-66.
- Schipper, E. L. F. (2006). Conceptual history of adaptation in the UNFCCC process. *Review of European Community & International Environmental Law*, 15(1), 82-92.
- Stern, P. C., Janda, K. B., Brown, M. A., Steg, L., Vine, E. L., & Lutzenhiser, L. (2016). Opportunities and insights for reducing fossil fuel consumption by households and organizations. *Nature Energy*, 1(5), 16043.
- US Department of Energy (2006). *Saving Water Saves Energy*. Technical Report <https://www.osti.gov/servlets/purl/927020>
- US Department of Energy (2016). *Exploring the energy benefits of advanced water metering* (authors) Berger, Michael A., Hans, Liesel, Piscopo, Kate, & Sohn, Michael D. Technical Report <https://www.osti.gov/biblio/1345196-exploring-energy-benefits-advanced-water-metering>
- US Environmental Protection Agency [EPA] 2016. *Advancing Sustainable Materials Management: 2014 Fact Sheet*. https://www.epa.gov/sites/production/files/2016-11/documents/2014_smmfactsheet_508.pdf
- Vlassopoulos, A. C. (2012). Competing definition of climate change and the post-Kyoto negotiations. *International Journal of Climate Change Strategies and Management*, 4(1), 104-118.

Comment #2 (Reviewer #1): *The “effectiveness” of interventions can be assessed in terms of behavioral change or in terms of environmental impact, which can be expressed as the product of technical potential (TP, or the impact that would be achieved if the behavior were universally adopted) and behavioral plasticity (BP, or the proportion of universal adoption that can be achieved). This paper focuses only on BP. It does a rigorous analysis of this, for the behaviors on which studies available studies meet its methodological criteria and for the efforts used in that work to change behavior.*

Reply to Comment #2 (Reviewer #1): We fully endorse the reviewer’s perspective about the focus of our work – we only examine behavioural change effects (or behavioural plasticity - BP). We also agree that the technical potential (TP) of different interventions is an important point and worthy of attention. It was simply not our goal to perform this analysis. We are behavioural scientists and it is in that capacity that we are focused on behavioural effects – not on technical effects, even though we do acknowledge their importance. Furthermore, we do not consider an analysis strictly focused on behavioural effects less relevant or a smaller contribution to the literature. Our aim was to conduct a rigorous and well-grounded analysis of the most effective ways to promote behaviour change, which can provide a reliable basis for further BPxTP analyses for scholars whose interests and expertise are more aligned with such a research goal. Ideally, interventions worthier of implementation should have a high BPxTP impact but such a claim does not preclude the need for a precise estimation of BP effects.

We also include this response in our Discussion section (please note track changes in page 11; third paragraph from the bottom of the page).

Comment #3 (Reviewer #1): *Regarding the conceptualization of “behavioural interventions”, it seems clear from the categorization of interventions, that among the interventions that are excluded from this review are regulations, government policies, financial incentives, energy or equipment prices, and the like, all of which affect behavior. This is an understandable choice, but by making*

this choice, the author(s) make it impossible to consider the likelihood that the interventions studied may have greatly different effects depending on the policy or economic forces also affecting decisions. The study investigates the roles of several “moderator variables” but not of policy or economic factors (except for the comparison across geographical regions, which surely includes policy and economic influences but does not disaggregate them). Past research indicates that the effects of particular kinds of “behavioural” interventions can be strongly influenced by other factors in place when the interventions area introduced.

Moreover, the most effective interventions, in terms of TP x BP, are those that integrate interventions of different types. For instance, to quote Dietz et al. (2009), “the most effective documented weatherization programs... have combined financial incentives (grants or rebates covering most of the retrofit cost), convenience features (e.g., one-stop shopping), quality assurance (e.g., certification for contractors, inspection of work), and strong social marketing. The highest recorded plasticity is 85% over 27 months,” which was recorded in the Hood River (Oregon, USA) Conservation Project in the early 1980s. And weatherization has much higher TP x BP than almost all the behaviors for which studies met this paper’s methodological criteria.

Reply to Comment #3 (Reviewer #1): This comment makes two distinct yet interrelated points. First, the reviewer warns that examining behavioural interventions without considering the broader macro-level policies or regulations in which these behavioural interventions are implemented may pose some limitations in the estimation of effect sizes. Second, this limitation may be critical because the reviewer claims that the combination of multiple types of interventions is more effective. These are two important points that warrant discussion. We partially concur with the first point but recommend caution regarding the rationale used in the second point.

Regarding the first point, the reviewer rightly points out that we do not include in our analysis the evaluation of macro-level regulations and economic incentives. However, this does not necessarily pose a limitation. Our clear focus is on behavioural interventions. Whether these interventions interact, or not, with regulations or financial incentives, is worthy of investigation and future research should pursue this goal – it was simply not our goal here. The reviewer seems to consider this a more serious flaw because he/ she holds the assumption that interactive effects are expected to occur, and therefore, our effects appear to have been estimated in an artificial vacuum. We kindly would like to present some arguments contesting this rationale.

Firstly, it should be noted that the reviewer’s claims that integrative interventions that combine a variety of actions have the highest effectiveness are mainly based in a single study – the Hood River Conservation Project (see reference below Hirst 1988). This, in itself, poses uncertainties about generalization and replicability. However, our key question is: What was the main driver of the effects reported? The author states, without providing data to support such claim, that (page 317):

” The key factors leading to high participation include the offer of free measures, determination on the part of HRCP staff to enlist every eligible household, the use of many community-based marketing approaches, extensive word-of-mouth communication among Hood River residents (initially stimulated by the solicitation of households to participate in the special studies a few months before HRCP officially began), and the early-1985 personal contacts by staff among the remaining nonparticipants”.

Both the author and reviewer assume additive or multiplicative effects given the variety of actions taken, but this is just an assumption. Although additive or multiplicative effects may have occurred, there is no empirical evidence to support such claim. Let us illustrate this very critical point with the work from Lourenco (2015; see reference below) which rigorously tested main versus combined effects of both financial and non-financial interventions. The interventions tested were feedback, praise and monetary incentives, and these interventions were tested in isolation and in combination (between-subject conditions). This study was conducted in a setting unrelated to climate change (work productivity) but it clearly shows how the assumption of higher effectiveness for multiple interventions did not hold. There were individual main effects but no positive interaction effects. Similarly, for climate change mitigation, without a precise estimation of main effects versus interaction effects, we cannot precisely estimate the additional marginal contribution of multiple actions.

Moreover, a careful look at the Hood River project reveals that the conclusion about the high effectiveness of this study can be questioned. This study (i) was not a randomized controlled trial but a

quasi-experiment with matched control groups (i.e., neighbour communities); (ii) households self-selected to be part of the study – which from our analysis we know is associated with higher effect sizes; (iii) households that chose not to participate significantly differed from participating households. Participating households were poorer, had larger households and were less likely to own a house, which produces unknown impacts in intervention effect; (iv) the project’s main achievement was the uptake of free energy audits and measures (e.g., ceiling insulation) but tangible energy savings were low and much lower than expected.

Lastly, many of the behaviours included in our analysis lack intervention either from legislation or economic incentives. Most countries lack legislation, penalties or rewards for e.g., recycling or food waste. In other cases, e.g., consumption of animal products or air flights, there is typically no governmental interference to modulate demand. Possibly energy demand is the context where more has been done in terms of legislation and economic incentives, such as fluctuating peak-hour electricity prices, or government subsidies for electric vehicles or installation of solar panels. But even for such a case, our goal was to isolate and estimate the effect of behavioural interventions. Once such effects are robustly estimated, future research can build on our work and move to compare these effects with economic incentives or examine possible interactive effects. This is the clarity our paper aims to establish.

We consider this remark – and our response to it - to be crucial to establish the validity of our work. We added these considerations to our Discussion section as our final reflection (last paragraph, pages 11-12) but given its importance, we also present the revised text here:

“The available field experiments do not give rise to hopeful predictions about relying on the contribution of behavioural interventions to tackle climate change. It could be argued that low-cost small changes, scaled-up to large segments of the population, could produce significant positive changes. Whereas we acknowledge that a small impact is better than no impact, our results show that previous research may have overestimated how much behavioural interventions may help us achieve large and pressing reductions in carbon emissions. There is the possibility that behavioural effects could vary depending on the policy or economic forces concurrently in place or that specific combinations of behavioural strategies with legislation and/ or financial incentives produce better results. However, this is an open empirical question. We should not assume positive additive or multiplicative effects just because a variety of actions is taken in combination. Thus far, there is no clear causal support for such an assumption. Without a precise estimation of main effects versus interaction effects, we cannot precisely estimate the additional marginal contribution of multiple actions. Furthermore, there are several behaviours critical to mitigate climate change that are not targeted by regulation nor economic incentives. These are the cases, for instance, of food waste or recycling, usually with no legislation, penalties nor rewards to promote them. For the consumption of animal products or air flights, there is typically no governmental interference to regulate demand. This increases the reliance on behavioural interventions. In conclusion, our work provides a foundational ground for future research to build on and move to compare behavioural effects with economic incentives and regulations, or examine possible interactive effects. Whether economic incentives or regulations could be established as more effective interventions should also be subject to rigorous evaluation.”

Hirst E (1988) The Hood River Conservation Project: An evaluator’s dream. *Evaluation Rev* 12:310 – 325.

Lourenço, S. M. (2015). Monetary incentives, feedback, and recognition—Complements or substitutes? Evidence from a field experiment in a retail services company. *The Accounting Review*, 91(1), 279-297.

Comment #4 (Reviewer #1): *This paper says that Dietz et al (2009) used their estimates of behavioral plasticity to predict reductions in carbon emissions over 10 years and that the 2018 IPCC Special Report used those numbers to estimate emissions reductions from “behavior change interventions.”*

In fact, Dietz et al used their BP estimates to point to “Reasonably Achievable Emissions Reductions”, that is, reductions that could be achieved if the best documented and feasible known interventions were used. In other words, this was a best-case estimate, not a prediction.

If the IPCC report took those estimates as predictions, it misread the Dietz et al. paper. The BP estimates in Dietz et al. should not be taken as predictions. I see this paper as offering documentation that the studies meeting this paper’s methodological criteria do not by and large represent best practice.

The great variation in effect size reported in this paper indicates that much remains unknown about how to achieve higher levels of BP, including levels that Dietz et al. considered reasonably achievable. This is an area where more research is much needed, especially if it is focused on behaviors with high TP. I recognize that it is difficult to conduct randomized trials of interventions to affect high-TP household behaviors, especially because most of these behaviors are infrequent. It is especially difficult if the interventions most worthy of examination integrate financial and regulatory incentives with “behavioural” influences, as was the case in the Hood River project. This likely explains the relative absence of such studies from those reviewed here. It is also difficult to do rigorous evaluation when so many interacting variables may be in play. These difficulties do not invalidate the analyses done here, but they do affect the conclusions that can be drawn from them. This review appears to indicate that, measured in terms of degree of BP, the interventions studied, taken in isolation from other variables with which they are likely to interact, have very limited effects that are not long-lasting.

Reply to Comment #4 (Reviewer #1): We would kindly like to raise several reservations about these comments. We do acknowledge the reviewer’s excellent point about Dietz et al. results being more properly referred to as best-case estimates, rather than predictions, and thus toning down their role in the IPCC Report. We revised our text accordingly to reflect a more moderate tone in this description (pages 9-10 but given the importance of this point, we also present the revised text here, at the end of our response). Furthermore, we fully agree that there is much we don’t know yet about how to increase behavioural plasticity and promote behavioural change. This was in fact the fundamental motivation for our work.

However, we would like to offer diverging views on other remarks made. We are focused on producing rigorous and precise impact evaluation of behaviour change interventions. For this reason, we conducted a long procedure to identify, select and calculate behavioural effects from field experiments. We do not claim that our included papers represent the ‘best practices’ but we do claim that our papers represent the best available evidence from which estimates can be calculated about whether interventions work or not, and if so, to what extent.

Our work benefits from methodological transparency, as well as a strong recognition that experimental research designs, particularly when developed in the field, tend to produce low behavioural plasticity estimates. Yet, because these research designs allow us to establish ‘cleaner’ causal effects and are developed in ecologically valid contexts (e.g., measure tangible daily household actions), we claim these estimates are robust and more likely to represent what policies and public programs tend to be able to achieve.

The reviewer proposes that we reported low effects because we did not include the “best practice” interventions. However, it is not clear to us which are the grounds to question the validity of our approach – fully described and based on the highest methodological standards – while taking at face value that Dietz et al. provide higher estimates because they were able to identify the “best practices”. This is a particularly striking comment given that there was no transparency in how interventions were identified and selected in Dietz et al. We are not, in any way, suggesting that Dietz et al. were not able to do so. We are simply stating they provided no detailed procedure of how these ‘best practices’ were selected.

We consider Dietz et al. to be an extremely important paper, and view our own work as an extended update on their research published a decade ago (2009). Dietz et al. themselves acknowledged at the time that their data provided “a reasonable initial guide to what can be achieved” and “more precise estimates can be developed with better data” (p. 18454). There are several differences in the research goals between Dietz et al.’s and our paper – namely, Dietz et al. had a keener interest in the technical potential of interventions - but both papers examined behavioural plasticity (BP). It is simply

regarding this point that we offer divergent views. It is not our goal to scrutinize Dietz et al. work but simply to respond to the questions raised. For this reason, we will discuss what we consider to be some limitations in Dietz et al. analysis of behavioural plasticity (described in their page 18453 paragraphs 3-5).

Firstly, intervention selection was based on unknown criteria. The reader has not clear indication of which interventions were actually chosen for the estimates of BPxTP. Many of the references stated by Dietz et al. (page 18456) are books that we have reviewed, which provide mostly narrative and qualitative analyses, and generally do not provide quantification of effect sizes or aggregate estimates of effectiveness. Our approach is strictly an evidence-based intervention approach (Higgins & Green 2008; Gertler et al. 2016), which prioritizes transparency and the estimation of causal effects.

Secondly, Dietz et al. state they borrowed some interventions from health promotion, also without a clear indication of which were such interventions. This poses an additional limitation because environmental behaviour differs from health behaviour in an important dimension: personal accountability. Typically, there are no personal, direct consequences of behaving for or against the environment, whereas individuals bear the costs or benefits of their own health choices such as smoking or eating fast food. Thus, how the motivational triggers behind health interventions can be translated into environmental interventions remains uncertain.

Lastly, we resume the response to Comment #2 about single versus combined interventions. Both the reviewer and Dietz et al 2009 make the point about higher effectiveness of combined interventions based on a single study conducted in the late 80's (The Hood River Conservation Project). We reiterate that we cannot necessarily assume that this intervention was effective due to its multiple components – this may or may not have been the case. Without proper impact evaluation techniques comparing single with multiple stimuli interventions, we are not be able to ascertain this. This requires an evaluation of main versus interactive effects of the building blocks of interventions and it is an absolutely crucial step to improve intervention science and policy.

Our claim that previous work (i.e., Dietz et al.) may have overestimated the carbon reductions that may be achieved from household action is based on our estimates about behavioural plasticity, which are much more conservative than Dietz et al.'s, even in best case scenarios, given the range of our confidence intervals.

We present here our response to this very important comment, which is now included in our revised version of the paper (pages 9-10):

“Our work indicates alarming low levels of behavioural plasticity: most household behaviours relevant to climate change exhibit resistance when targeted by behavioural interventions. In particular, critical actions such as purchase of energy efficient appliances and private car use were barely affected by the behavioural interventions examined. We compare selected estimates with Dietz et al. (18), who also estimated the behavioural plasticity of several similar behaviours. For instance, Dietz et al. estimated that, under the most effective interventions, an 80% behavioural plasticity could be achieved for the purchase of energy efficient appliances. We report an average null effect size. For choices related to the private use of car, Dietz et al.'s estimates are more conservative (15% to 25%) but still substantially higher than ours (2.5%).

There are two key differences between our work and Dietz et al. (18) which could account for these discrepancies. Firstly, Dietz et al. included interventions using financial incentives. The discrepancies found may be due to a higher effectiveness of financial incentives. However, there is no clear way to establish this hypothesis because Dietz et al. did not specify which interventions were used, under which circumstances and for which specific behaviours. Secondly, Dietz et al. based their estimates on proxies and less stringent research designs. We consider this an important difference in our methodological approach, which allowed us to estimate the behavioural plasticity of tangible behaviour in natural experimental settings. It is well-established that interventions based on self-reported intentions and attitudes and non-experimental studies tend to produce higher effect sizes (15-17). Although the roots for these discrepancies cannot be fully uncovered, our results raise the possibility that past forecasts about how much can be achieved to reduce carbon emissions by household action could have been inflated. This is a critical issue because Dietz et al. used their results as “best-case scenarios” to estimate reductions in carbon emissions over 10 years from household action. Our estimates about

behavioural plasticity are much more conservative than Dietz et al.'s, even in best case scenarios, given the range of our confidence intervals. Future research should further address these inconsistencies because, even after a decade, Dietz et al. represent the benchmark for "best-case scenarios" estimates for the reduction in carbon emissions that could be expected from household action (1, 3) – and these estimates are based on very optimistic levels of behavioural plasticity.

Gertler, P. J., Martinez, S., Premand, P., Rawlings, L. B., & Vermeersch, C. M. (2016). *Impact Evaluation in Practice* (2nd Edition). Washington DC: The World Bank.
Higgins JPT, Green S. (2008). *Cochrane Handbook for Systematic Reviews of Interventions*. Version 5.1 <http://handbook-5-1.cochrane.org/>

Comment #5 (Reviewer #1): *What does this mean in terms of publishing the paper? Presuming that the statistical analyses hold up to scrutiny from reviewers who know these methods better than I do, it means that the analyses deserve publication (although I'm not sure whether a communications journal is the right venue) but that before any publication, the authors' claims for their practical implications need to be considerably revised and toned down to be defensible. In my view, the data do not indicate that "predictions about how much carbon emissions can be reduced by targeting demand-side behaviour changes" are necessarily inflated, because the work meeting the criteria for inclusion in this review represent a biased sample of efforts to change household demand behavior—especially considering that such efforts include regulatory and financial incentive-based interventions, as well as combinations of these with what this paper refers to as "behavioural" ones. The paper indicates that efforts to target demand via the intervention types reviewed, alone and not in combination with other intervention strategies, have small effects, so should not be expected, on their own, to have major effects in the near term on the behaviors examined in this review. One last point: the paper doesn't really speak in any direct way to "carbon emissions." Many of the outcome variables in the studies reviewed, such as water consumption and food waste disposal, have little or nothing to do with carbon emissions. The paper would be more on target (if perhaps less interesting to IPCC) if it identified the outcome variables in terms of direct household demands on environmental resources rather than on carbon emissions.*

Reply to Comment #5 (Reviewer #1): We deeply thank the reviewer for considering the paper merits publication - despite the uncertainties he/she raised, which we hoped to have clarified to a great extent.

The reviewer makes a new point here which we fully endorse. We make no claims about carbon emissions. This is a limitation of our work which we debated in the discussion section. This was the trade-off for including multiple interventions and contexts that vary highly in their reported outcomes – both statistically (e.g., t-test, regression coefficients, odds ratios) and in terms of units of measurement (e.g., liters, kilos, kWh). To be able to develop a comprehensive meta-analysis, we harmonised all dependent variables as standardised mean differences (SMD) or Cohen's d. What we gained in breadth, we lost in depth because SMD are unrelated to the original measurements. Thus, we cannot calculate exactly how many liters of water or kWh of energy were saved. Nonetheless, we consider that this limitation does not invalidate our contribution to estimate the effectiveness of different interventions to trigger mitigating behaviours. Whether the behaviours changed reduce more or less carbon emissions is a different matter altogether – extremely important but outside our scope. Our focus is on behavioural plasticity.

We also offer a clarification about water consumption and food waste. The reviewer makes the point that water consumption and food waste "have little or nothing to do with carbon emissions". We reinforce our answer, more fully developed in response to Comment #1, that the household actions that consume the largest amounts of water (showers and dishes/ clothes washing) are also actions that consume large amounts of energy (US Department of Energy 2006, 2016). Moreover, food waste is a vastly overlooked driver of climate change. We examined total amounts of food waste produced. The Food and Agriculture Organisation (FAO) quantified the food wastage in terms of carbon footprint and concluded that the contribution of food wastage emissions to global warming is almost equivalent (87%)

to global road transport emissions⁴. As famously stated, if food wastage were a country, it would be the third largest emitting country in the world⁵.

Furthermore, the reviewer also reiterates here his/ her premise that the small effect sizes our paper reported are due mostly to the types of intervention examined, plus the fact that we isolated main effects instead of examining integrative interventions. Regarding the former, as previously acknowledged, we did in fact exclude regulations and economic incentives. But at no point do we claim that regulations and economic incentives have no impact or are irrelevant. For the reasons stated in our introduction (page 2), we focused on behavioural interventions, which represent substantial and increasing efforts to implement changes regarding climate change. Our paper sets a robust foundational ground to examine the impact that economic incentives may have, in isolation or in combination with behavioural interventions, to promote change. But this is an evaluation that requires strict methodological procedures and should not be based on case studies and expert insights. Regarding the latter, about integrative interventions, please refer to our answer to previous comments, where we detail our reasons for disagreeing with this perspective. There is simply no causal evidence supporting that combined interventions are preferred or more effective than single-stimuli interventions.

Lastly, we would like to comment on the reviewer's description of our inclusion criteria. We would agree with adjective qualifiers for our final sample of included papers such 'strict' or 'meticulous' but the description as 'biased' simply implies some partiality or subjective favouritism. We completely refute this qualification of our inclusion criteria, which were defined in transparency to comprise the best evidence from experimental field studies, with both high internal and external validity.

US Department of Energy (2006). *Saving Water Saves Energy*. Technical Report <https://www.osti.gov/servlets/purl/927020>

US Department of Energy (2016). *Exploring the energy benefits of advanced water metering*. Technical Report <https://www.osti.gov/biblio/1345196-exploring-energy-benefits-advanced-water-metering>

Reviewer #2:

We will not present Reviewer #2 several positive comments and only focus on the aspects mentioned that required a response.

Comment #1 (Reviewer #2): Assuming the goal of this paper is to shorten the gap between empirical evidence and existing policy instruments (the so-called translational research), the paper can be a lot stronger if the policy (and/or theoretical) implications are provided. The Discussions section clearly outlines research challenges and gaps, however, discussion regarding what we should do next as researchers and policy-makers is missing in the paper. For instance, knowing that studies composed mostly of self-selected samples tend to greatly inflate the intervention results, how should we better approach study recruitment? Some of these issues are study limitations while some are not (meaning can be improved). Teasing these out is important to advance behavioral intervention research.

Reply to Comment #1 (Reviewer #2): We sincerely thank the reviewer for this comment and for the recognition of the translational nature of our work. We do aim to narrow the gap between scientific evidence and policy making by estimating the relative effectiveness of different behavioural approaches to mitigate climate change. We have revised our discussion section to elaborate on the reviewer's comment (please consult track changes pages 10-11).

We found a substantial difference between studies that required participant explicit consent to participate versus the studies that had no such requirement. This is likely because (i) participants that accept to participate have a higher interest in climate change mitigation and pro-environmental living,

⁴ <http://www.fao.org/3/a-bb144e.pdf>

⁵ https://wriorg.s3.amazonaws.com/s3fs-public/uploads/FLW_graphic2.jpg?_ga=2.15891443.437954611.1545999227-1793608830.1545999227

and because (ii) people are aware they are being monitored. This simple detail has proven to be determinant in changing behaviour, and there is evidence for this in the context of energy demand (Schwartz et al. 2013). This is an important question because scaling-up and implementing these interventions in a larger scale, possibly designed as policy-level action, necessarily targets a much wider variety of individuals with varying motivation to act pro-environmentally. Using the estimates from self-selected participants to inform policy makers creates an expectation that may not be matched when targeting larger populational segments. This question exposes an apparent trade-off between estimate reliability and ethical research conduct, namely the bedrock requirement of informed consent. We suggest several possible courses of action for future research.

Firstly, an increased use of public data or publicly observable behaviours, in collaboration with service organizations, utilities companies, and government institutions with access to large volumes of data that individuals may have accepted to share that are not sensitive or confidential in nature.

Secondly, waiving informed consent. Researchers should provide sufficient argumentation for waiving informed consent to allow the IRB to weigh the arguments carefully before granting such a waiver for any given project (37). Namely, our work showed a sevenfold difference between naïve versus self-selected participants, which should not be overlooked and could potentially be used to request waiving informed consent when no sensitive data is being collected.

Thirdly, when informed consent is absolutely required, researchers could make efforts to monitor the behaviour of non-participants. This will likely be an analysis performed in aggregate terms (e.g., buildings, regions), given that household-level data was not provided. Although this would provide an imperfect comparison, it could potentially still deliver important insights.

Lastly, when none of the above are possible, researchers should keep in mind that effects are likely overestimated compared to the general population, and apply a discount factor that aimed to correct for inflated effects. For instance, our results show a sevenfold difference between naïve versus self-selected participants. Thus, for instance, an effect size $d=-0.350$ based on self-selected participants could correspond to an effect size $d=-0.050$ in naïve subjects. This would be a rough estimate but it would contribute to caution what can really be achieved in larger populations that have heterogeneous interest and commitment to the environment.

Rebers, S., Aaronson, N. K., Van Leeuwen, F. E., & Schmidt, M. K. (2016). Exceptions to the rule of informed consent for research with an intervention. *BMC Medical Ethics*, 17(1), 9-20.

Schwartz, D., Fischhoff, B., Krishnamurti, T., & Sowell, F. (2013). The Hawthorne effect and energy awareness. *Proceedings of the National Academy of Sciences*, 110(38) 15242-15246.

Comment #2 (Reviewer #2): I think the author(s) can touch base a bit on the instruments used in the intervention to measure behavioral changes – how these contribute to or weaken the intervention effectiveness. While looking through the table of included papers (SI.2), it appears none of the interventions adopts technology that captures objective behavioral data, such as wearable trackers. I wonder if this type of study tends to be ruled out during the eligibility assessment process. If yes, then it's hard to say that “previous work has overestimated the impact of lifestyle interventions that promote sustainable behaviors”, and that it can be mentioned in the study limitations. In fact, it's also important to note that research employing objective behavioral tracking/monitoring technology could be less methodologically rigorous (relatively speaking), yet whether and to what extent its behavioral outcomes are sustaining compared to information and social influence strategies needs to be investigated. The paper can also benefit the audience by discussing the future directions of conducting behavioral interventions from an interdisciplinary angle.

Reply to Comment #2 (Reviewer #2): Thank you for the opportunity to clarify this comment, which we believe is a misunderstanding. A keystone inclusion criterion was the restriction to studies reporting objective, tangible behavioural changes. We believe the reviewer may have misread Table SI.2, because in fact the first study presented (Aittasalo et al. 2012) used pedometers – which would fall under the category of wearable trackers. Many studies did not use any technological means *per se* as wearable trackers, but reported nevertheless objective behavioural data from official gas bills, water meter readings, weight of recycling bins or kilometres in odometers. Our research goal was more closely

related to the content of the intervention and, for this reason, this was the main element we initially reported in Table SI.2.

Given the interest of the reviewer, we have now revised Table SI.2 to include the data collection means used (e.g., billing data, on-site metering, observed behaviour) to measure the behavioural changes reported in each study.

Please note an additional clarification: the reviewer seems to distinguish between “*objective behavioral tracking/monitoring technology*” and “*information and social influence strategies*”. Kindly note that these two things are orthogonal. Any tracking or monitoring technology is typically used as a means to collect data, whereas information or social comparison are interventions or experimental stimuli. For instance, several papers used social comparison messages and tracked their effects using real-time electricity metering from smart grids. We hope this clarifies the concerns raised by the reviewer.

Lastly, we appreciate the reviewer’s remark about interdisciplinary collaboration as a way to improve the discussion of our results and their repercussions. Conducting behavioural interventions from an interdisciplinary perspective is crucial particularly under two conditions. First, when implementing interventions that may combine different stimuli e.g., public service announcements, social comparison and government action. Complex, multi-layered interventions from the standpoint of their content and implementation may require the joint work of multiple experts. Second, using advanced technological solutions for data collection may also be more effectively implemented when combining behavioural scientists with technical expertise such as engineers and computer scientists. We now also include this rationale to our Discussion section (page 12).

Reviewer #3:

Comment #1 (Reviewer #3): This paper addresses a potentially important point. There is a lot of discussion about how we might attempt to ‘nudge’ individuals and households (or firms even) into following pro-environmental behaviors. However, my own take on the overall corpus of research in this area is that there is increasingly stronger evidence that it is ‘working’. You present a contrasting point of view, which is worthy of recognition, so long as the analysis holds up to scrutiny.

(...)

Regarding the subject matter, and phrasing of the results and contribution, I think it would be wise to contrast against the growing literature on ‘nudges’. This would seem necessary, since you need in my opinion to offer more direct qualification that your results challenge or contest our ‘known’ understandings, once we condition on all the things we ‘know’. I wonder for example if research on the use of in-home smart-meters or peer-benchmarked utility bills are reflected in your analysis or final conclusions. I provide a link to an industry report that points more towards the sort of areas in my

mind:

<https://www.mckinsey.com/~/media/mckinsey/industries/electric%20power%20and%20natural%20gas/our%20insights/giving%20us%20energy%20efficiency%20a%20jolt/sizing%20the%20potential%20of%20behavioral%20energy%20efficiency%20initiatives%20in%20the%20us%20residential%20market.ashx>

General searching around goal-setting and social-comparisons might be worth considering to uncover relevant academic literature.

Reply to Comment #1 (Reviewer #3): There are multiple ways in which this remark could be interpreted and, therefore, we will present several potential approaches in this response.

There are different interpretations of what ‘nudge’ entails. Some views consider nudge as a synonym for behavioural interventions – which is our core topic – and thus we assume this may have been the reviewer’s standpoint. All interventions that do not include regulations or economic incentives are commonly categorised as ‘nudge’. There are, nevertheless, more strict ways to conceptualise ‘nudge’. Nudges have been popularised by the acronym NUDGES developed by Thaler and Sustein in their seminal book (2004). This acronym summarises six principles of good choice architecture

(Chapter 5 pages 81-100) such as **D** standing for Defaults, or **S** standing for Structure complex choices (page 99-100).

Choice architecture is the designation used by Thaler and Sustain (2004) to accurately describe what they mean by ‘nudge’. It is the application of subtle changes based on insights from behavioural science to the design of interventions in order to “*indirectly influence the choices other people make*” (page 83). Please note that this is one of our categories of interventions, which we called *Facilitate* and described as encompassing choice architecture principles (our own pages 3-4). We explicitly describe *Facilitate* as “*choice architecture [that] nudge behaviour by removing external barriers, expediting access or facilitating pro-environmental behaviour by altering the structure of the environment in which people make sustainable choices. Examples are putting recycling bins closer to regular waste, reducing plate/ glass size or setting air conditioning by default to higher temperatures*” (our own pages 3-4). Therefore, we do agree with the reviewer as to whether nudges are working. In fact, we showed this is the category of interventions most likely to work.

However, when it comes to the examples given by the reviewer for nudges such as “*peer-benchmarked utility bills*” or “*goal-setting and social comparisons*”, some clarifications are required. Not all applications from behavioural science can be categorised as nudges. Nudges represent a specific category of interventions based on results showing that people are often driven by emotional reactions, habits, and limited cognitive capacity. Therefore, by Thaler and Sustain’s own definition, interventions promoting goal-setting are not a ‘nudge’. This type of interventions was labelled by us as *Engage* because, as we mention on page 4, these are interventions that “*try to change psychological processes such as promoting goal-setting, implementation intentions, commitment or engagement, and mindfulness towards sustainability*” (our own pages 3-4).

Let’s now take the case of “*peer-benchmarked utility bills*” or what the reviewer also described as “*social comparison*”. Again, this is not part of Thaler and Sustain (2004) conceptualisation of choice architecture or ‘nudges’. We categorised this type of interventions as *Pressure* that aimed to “*provide a comparative reference with respect to the sustainable behaviour of close others such as neighbours, colleagues/ friends or fellow citizens, based on principles of social influence*” (our own pages 3-4). Social comparison falls within this category.

Thus, although we disagree with the characterisation of some the examples provided by the reviewer as ‘nudge’, all the examples mentioned had been included in our analysis, simply under different categorisations – i.e., categories with different names. Nonetheless, all the examples provided were included in our analysis.

However, we acknowledge that these misunderstandings may have been created by the designations we used for our categories of interventions. In the revised version of the paper, we renamed the types of interventions to accommodate the reviewer’s comments, namely by replacing verbs with nouns that may better convey our message (please note changes in page 4, Table 1 page 5, and Figures 5 and 6 pages 8-9):

First Version	Revised Version
Inform	Information
Appeal	Appeals
Engage	Engagement
Pressure	Social comparison
Facilitate	Nudges

Hoping this discussion captured the reviewer’s conceptual remarks, then the discussion turns to what ‘working’ means. If by working, the reviewer means reporting statistical significant results, we also agree that it is working. Our estimates, both overall and for most subgroup analyses performed, are statistically significant. Effects seem to be occurring. Our main contribution is to understand to what extent are these effects substantial and perhaps with an overestimated expectation of success.

When I take a pill for a headache, my notion of the pill ‘working’ is whether it eliminates or greatly reduces my pain. If I take a pill that only very slightly reduces my headache, I probably will consider the pill has not worked. The standards to evaluate whether social and behavioural interventions ‘worked’ are much less stringent than clinical interventions, and it is often unclear what could account

as ‘working’. Thus, we followed Cohen’s own guidelines (1988). Cohen’s d below 0.2 tends to be interpreted as a very small effect, 0.3-0.4 a small effect, 0.5-0.6 a moderate effect, 0.7 and above a large effect. Thus, our conclusion from an average overall effect of $d=-0.093$ is that there is a very low impact of behavioural interventions to promote climate change mitigation. All our interpretations of effect magnitude are based on Cohen’s own recommendations.

It is often the case that conclusions from meta-analyses are less impressive than results from individual studies. Individual studies tend to become well-known if reporting results that are considered remarkable or noteworthy. Hunt Allcott’s work in collaboration with OPOWER is perhaps the most pivotal example (Allcott 2011; Allcott & Mullainathan 2010; Allcott & Rogers 2014). But there is a myriad of other studies, often equally robust and well-performed, that report much less impressive results. There are also many other unpublished studies that, due to their non-significant results, never see the light of day. A meta-analysis contrasts the perceptions created from a couple of famous studies with the aggregate and systematic evaluation of a body of research about what is ‘working’ and the extent to which interventions are ‘working’. A fundamental appreciation of the power of meta-analysis is its aggregate evidence of impact.

With respect to the McKinsey Report mentioned by the reviewer, it actually illustrates the type of evidence we aimed to transcend with our work. Firstly, in the authors’ own words “*results are self-reported and so have inherent inaccuracies; as such they represent only a rough estimate of the opportunity*” (page 3). Secondly, this Report is primarily focused on technical impact and which household actions would have the highest impact in energy savings. This was not our focus. Lastly, what the authors describe as behavioural ‘nudges’ (page 3) was limited to a brief reference to the social comparison reports provided by OPOWER. As we have clarified above, social comparison was not defined by Thaler and Sustein (2004) as ‘nudge’. Nonetheless, we did include such interventions in our analysis but under the category designated *Pressure (now designated Social Comparison in the revised version of the paper)* - which encompasses all interventions based on social influence and peer pressure.

Allcott, H. (2011). Social norms and energy conservation. *Journal of Public Economics*, 95(Special Issue: The Role of Firms in Tax Systems), 1082-1095.

Allcott, H., & Mullainathan, S. (2010). Behavioural Science and Energy Policy. *Science*, 327(6970), 1204–1205. <http://doi.org/10.1126/science.1180775>

Allcott, H., & Rogers, T. (2014). The short-run and long-run effects of behavioral interventions: Experimental evidence from energy conservation. *American Economic Review*, 104(10), 3003-37.

Cohen, J. (1988). *Statistical power analysis for the behavioural sciences*. Hillsdale, NJ: L. Lawrence Erlbaum Associates.

Schultz, P. W., Estrada, M., Schmitt, J., Sokoloski, R., & Silva-Send, N. (2015). Using in-home displays to provide smart meter feedback about household electricity consumption: A randomized control trial comparing kilowatts, cost, and social norms. *Energy*, 90, 351-358.

Comment #2 (Reviewer #3): As a note, in the introduction you imply that higher prices for water and energy are suitable. While this would effect change at the margins, there must be some concern that these effects would be most heavily felt among the lower income component of society and that they might be contradictory to economic growth, or rather development, aspirations.

Reply to Comment #2 (Reviewer #3): We do mention that prices for water consumption are typically very low and that there are no penalties for not recycling or producing too much food waste. As to whether higher prices are suitable, we make no such claim. From a purely technocratic perspective, higher prices could be enforced to lower consumption but we restrained from making such recommendations because, as the reviewer rightly point out, this would likely impose trade-offs between climate change mitigation versus economic growth and quality of life. This is an extremely important question but one we feel less qualified to address. Whether prices could be raised for energy and water consumption for the general population, paired with subsidies for more deprived groups, will likely be a political decision – notwithstanding the scientific evidence.

Comment #3 (Reviewer #3): *In short, in my attempts to replicate your main results (using R, not STATA) I have encountered some differences. The first, and probably most important is that the sample sizes for the ‘individuals’ sub-sample are not a perfect match. Even making this comparison was time-consuming, since you in fact did not include N as a variable in your .dta file. I therefore had to manually enter, check and re-check these from the table in the paper. Given your discussions around the importance of sample size I was surprised about this (and even a little apprehensive as to why you had removed it from the STATA .dta file.) Please be sure to provide ALL variables in a future version of the data.*

From my estimates N= 2,376,336 but from your paper you claim that N=2,364,924, giving a difference of 11,412, that I have so far been unable to reconcile through a number of attempts. For the households there are 725,904 cases (we arrive at the same numbers here), but then in row 1 of Table 1 the paper refers to a total N=2,984,159, the difference being explainable by using the value of household-N=619,235, yet without clarity over where the remaining 106,669 households went to (comparing again to the number in the abstract of 725,904). This leaves open two possibilities either (i) that I am making some trivial mistakes in counting the cases which I would presume to be due to some steps in the process you take needing to be made a little clearer, or (ii) that there are errors. What is required of me at this stage is to verify that (ii) is not the case.

I therefore, at this stage of review process present a direct request that you politely double check the numbers, but to also remove the possibility that I am taking different steps than you are, to provide the replication file itself (a .do file I assume) so that I can verify the steps taken.

Reply to Comment #3 (Reviewer #3): We genuinely thank the reviewer for this feedback. We will present some clarifications about why the meta-analysis protocol used did not require sample sizes for its calculation as well as clarify the differences found in sample sizes. The reviewer considers the fact that there were some discrepancies in the sample sizes reported in Table 1 “the first and probably most important” concern he/ she has. However, this can be easily clarified.

Regarding the information about sample sizes missing from our database, we did not intentionally remove this information. The reviewer rightly points out that sample sizes are crucial for meta-analysis – that is indeed the case. Therefore, we would not withhold this information from the readers. However, the STATA meta-analysis command we used [metaan] does not require the input of sample sizes in the database: “the command requires the studies’ effect estimates and standard errors as input” (page 396) <https://www.stata-journal.com/sjpdf.html?articlenum=st0201>. The reviewer’s concern is understandable because most meta-analysis require sample sizes as a key input. However, because we needed to convert all outcomes into a common measure (Cohen’s d or standardised mean differences) we used a meta-analysis protocol specific for this measure, which did not require sample size input.

Thus, how did our estimates account for sample size if this was not included in the database? By using a validated tool for outcome conversion. We used the Effect Size Calculator from the Campbell Collaboration. The Campbell Collaboration <https://www.campbellcollaboration.org/> is a leading organisation for evidence synthesis and meta-analysis production. This organisation developed a tool to support scholars and practitioners that run complex meta-analysis, which require outcome conversions and standardisations. This tool is the Effect Size Calculator <https://www.campbellcollaboration.org/research-resources/research-for-resources/effect-size-calculator.html>.

The procedure was as follows: for each individual paper in our sample, we extracted their reported outcomes – typically reported as t-test, regression coefficients+SE, means+SDs, percentages, etc. We inputted these estimates plus the sample size for each individual study in the calculator. The calculator then returns an estimate for Cohen’s d, its variance and the 95% CI. These are then the estimates transferred to STATA to perform the meta-analysis. The clarification presented here was added to our Methods section in page 13.

Therefore, sample size was not required in the database. Nonetheless, given that sample size is a crucial element for meta-analysis and shown to be critical for our results, we reported total sample sizes in our Table SI.2 detailing our included papers. However, for simplicity reasons, the sample size used initially was the ‘official’ sample size as reported by the authors of each paper. This total sample

size per paper may not have been sufficiently clear for replication purposes and in fact, it sometimes does not match the ‘real’ sample size after missing values or other exclusions have been accounted for.

Therefore, for clarity and transparency, we are now reporting the exact sample sizes per control group (variable N_CG) and per experimental group (variable N_EG). Please take note of this change in the STATA database to facilitate the reviewer’s verification analyses. In Table SI.2 in the supplementary material, we continue to present an aggregate sample size for each paper as a summary.

The reviewer makes an important remark about discrepancies in the total aggregate sample size reported by us in Table 1. This was our responsibility and we do apologise for the confusion. In the final stages before submitting the paper, we did find some discrepancies between our Tables presented in the paper and our working Excel files. We corrected the calculations in Excel but we missed some corrections in our Word document. For instance, the 106,669 difference for households refereed by the reviewer is related to a single paper (Ferraro & Price 2013) – this is their total sample (N=106,669). This paper was excluded by mistake from the final sum to reach the total aggregate numbers.

Although there were a couple of mistakes in reporting the total aggregate sample sizes, please note that this does not affect in any way the effect size estimates we presented. Given the procedure we used to calculate effect sizes based on the Campbell Calculator, sample sizes were correctly inputted there and estimates then transferred to STATA. This mistake occurred much later, when preparing the tables for visual presentation for the paper. Although this a mistake that should not have occurred, in the sense that it misrepresented the total aggregate sample size, it did not affect our results nor conclusions. Aggregate total sample sizes were not used in any calculation, these were reported just for the benefit of the reader.

We have now checked and double-checked all numbers, and confirmed their consistency throughout the paper. The final aggregate numbers we report now are based on the exact sample sizes we used in our analysis, and not the ‘official’ samples reported by authors. This adds to 724,792 households and 2,367,886 individuals. Please note all revisions in Table SI.2 and STATA database.

Comment #4 (Reviewer #3): I would further add that I attempted to further replicate all remaining analysis in the paper and obtained some different conclusions. Some of the differences are immaterial. For example, my own funnel plots place fewer observations outside of the confidence interval boundaries, but otherwise the core conclusion holds. However other differences are more material. For example, I cannot get close the main Cohen’s d estimates discussed from the final paragraph of page 3 onwards. Where you claim small effects, I am finding it difficult to obtain anything other than medium or large effects, unless I take steps which don’t follow the applied method, as I understand it from your provided description. I do not provide my analysis here for the simple reason that, as described above, I cannot yet even reconcile the descriptive statistics from the analysis and so subsequent analysis would of course differ. Maybe in solving that primary concern, the secondary concern over the estimates can also be alleviated.

Reply to Comment #3 (Reviewer #3): We would like to address this concern fully but without a clear reference to which R package was used and more statistical details, a response from us is more speculative than we would like. Furthermore, the reviewer mentions in rather vague terms that he/ she could not reach the same results, without specific examples that we could respond to.

Please note that we provided our database and detailed the methodological and statistical approaches taken (Campbell Effect Size Calculator and individualised STATA commands for various analyses) to allow replication and verification from reviewers. STATA is often considered the benchmark software for meta-analytic procedures, used in the highest impact publications worldwide (Lee et al. 2017; Medina-Gomes et al. 2017; Palmer et al. 2016). It is challenging to us to respond when not much detail is provided when using different procedures and software. We will try to speculate about what we consider to be the more plausible explanations for this difference.

STATA and R may have different parameterisation. Even if this was the case, it should not have produced massive differences, but only minor numerical inconsistencies. We are inclined to believe that the root of the problem is the incorrect use of sample size. More specifically, the estimates we presented in our database resulted from the joint input of each paper’s outcomes plus their sample

size. We used the procedure fully documented in our response to Comment #3 above. This was why sample size was no longer needed in the database. But the reviewer seems to be doubling on the use of sample size, using the estimates we provided (Cohen's d and 95% CI) and adding sample size to the calculations again. This may have produced different results.

Furthermore, the reviewer may have overlooked the adjustment for multiple estimates being provided for the same paper. Often, a single paper reports more than one experimental condition and/or more than one study. For instance, paper X compares a control group (N=100) with experimental condition 1 (e.g., information N=100) and experimental condition 2 (e.g., social comparison messages N=100). We would report a total N=300 for paper X. However, our database would include both the estimate for the effect of information (X1) and the estimate for the effect of social comparison (X2). Each of these estimates only corresponds to sample N=200. If the reviewer used the total N reported for all estimates, this naturally interferes with the results.

Thus, initially, for simplicity reasons, we reported in our Table SI.2, only the aggregate total sample size per paper. However, to support verification analyses and replication, we are now reporting the exact sample sizes per control group and per experimental group. Please refer to the STATA database where this information is now detailed per experimental arm (variables N_EG and N_CG).

In any case, we are fully receptive to replicate our results using R if the reviewer provides details of the analysis used, in particular the package used.

Lee, S. W. H., Chan, C. K. Y., Chua, S. S., & Chaiyakunapruk, N. (2017). Comparative effectiveness of telemedicine strategies on type 2 diabetes management: A systematic review and network meta-analysis. *Scientific Reports*, 7(1), 12680.

Medina-Gomez, C., Kemp, J. P., Dimou, N. L., Kreiner, E., Chesi, A., Zemel, B. S., & Evangelou, E. (2017). Bivariate genome-wide association meta-analysis of pediatric musculoskeletal traits reveals pleiotropic effects at the SREBF1/TOM1L2 locus. *Nature Communications*, 8(1), 121.

Palmer, S. C., Mavridis, D., Nicolucci, A., Johnson, D. W., Tonelli, M., Craig, J. C., & Natale, P. (2016). Comparison of clinical outcomes and adverse events associated with glucose-lowering drugs in patients with type 2 diabetes: a meta-analysis. *JAMA*, 316(3), 313-324.

Comment #5 (Reviewer #3): *In the paper you refer to the idea of short-term effects, yet do not in my mind clearly qualify what you mean by short term. After several re-reads I fell on the conclusion that by short term you mean while an intervention is in force, and that no long term effect refers to the idea that behavior reverses once the intervention is removed. But if an intervention is in force for a decade – could we still fairly deem this as short term? Is my inference right about how you define short-term? (I am fine if this is the case, but in my own primary discipline we often take a different view of what long-term and short-term mean).*

Reply to Comment #5 (Reviewer #3): This is an excellent point that we insufficiently addressed in the previous version of our paper. We have now included the duration of each intervention in our Table SI.2, in the supporting documentation and also added this data in the STATA database. The reviewer correctly understood our definition of short-term i.e., while in the intervention is in place. We characterised the intervention period as short-term given the mean and median duration reported in the included papers. Interventions ranged from a single day (one-shot intervention) to 730 days (2 years). However, the interventions in our sample had mean duration 64 days (SD=140 days) and median 7 days. The Figure below is now included in the paper as Fig 1 (page 3).

Most interventions are short-lived, with half of the interventions lasting up to one week, two thirds lasting up to a month (67.4%) and 83.4% lasting up to 3 months. We considered the duration of the overwhelming majority of interventions to be short-term by most disciplinary standards and interpretations. An important question could be whether duration could impact the intervention effect. We performed a meta-regression testing duration as a possible moderator and results showed no significant effect ($B=-.001$ $p=.435$) – we now report this result on page 4 as track change.

Nonetheless, to address the reviewer's remark, we have eliminated the use of 'short-term' and replaced it more clearly with 'while the intervention lasts' – please note changes in the Abstract, Results page 3-6, Discussion page 9).

Fig 1. Duration of the behavioural interventions reported in weeks

Comment #6 (Reviewer #3): *There is a paper identified in the dataset that is not included in the list of papers contained in the manuscript. Why? You have no available N for the paper by Hongbak et al (2016). Note that this would only increase the sample sizes further, and would therefore be unable to explain why my calculated N is greater than that reported in the paper.*

Reply to Comment #6 (Reviewer #3): Please note this paper is Kongsbak et al. (2016) and it was presented in both the Table SI.2 and reference list. We apologise for a spelling error in the STATA database which showed this paper as ‘Hongsbak’. The STATA database is corrected as the paper is now correctly labelled as Kongsbak. This was an oversight in spelling for which we apologise.

Comment #7 (Reviewer #3): *The formatting on Fig. 2 seems to need attention. First, the bottom part of 2017 is cut-off on the y-axis. Since this is a time-series, why not reverse the axis, as is a more typical convention when plotting time series data. And what do the circles on the plot represent? Intuitively in my mind there would be one circle per year reflecting the 10 year ‘cumulative’ effect but there are many more circles than this. Lastly, are these really cumulative plots? If you sum up the numbers in any given year they end up much larger than the numbers reported here. Can you please help clarify? My mean/median calculations are closer to your graph (though still not the same, with less volatility at the early part of the sample, and a larger sustained mean towards the end).*

Reply to Comment #7 (Reviewer #3): In order to perform this analysis, we used the documented *metacum* command (Sterne 2008) which restricts us to its features and output format. Different years are, by default, present vertically and not horizontally in cumulative meta-analysis. Please see an example below (from Haapakoski et al. 2015) that illustrates how our graph follows standard procedure and output format.

In cumulative meta-analysis, the pooled estimate of the treatment effect is updated each time the results of a new study are included. This makes it possible to track the accumulation of evidence on the effect of interventions over time (Lau et al. 1992). In response to the reviewer’s question, the circles in the plot represent the cumulative effect size at any given moment i.e., the average effect size resulting from the inclusion of studies up to a certain point. There is not an ‘average’ effect size per year, but each individual study is ordered and added by year of publication. Please note that cumulative meta-analysis is not additive i.e., it is not the sum of individual studies, but the updated average effect sizes over time (Lau et al. 1992).

We did not necessarily have to perform a cumulative meta-analysis to report average effect sizes. A cumulative meta-analysis returns the exact same final estimate as the standard meta-analysis, the only difference being that studies are ranked by year of publication. The chief contribution from cumulative meta-analysis is the visualisation of temporal patterns. We consider the visual impact of showing robust average estimates in the last decade an important contribution, in order to show that older studies tended to report, on average, higher estimates. This may be due to a variety of reasons – e.g., smaller sample sizes and self-selected participants – or perhaps unknown reasons related to the zeitgeist of the time. The main point is that assumptions and perceptions based on older studies may

provide an inflated expectation of effectiveness. A crucial example for this discussion is given by Reviewer #1's repeated claims of success of a single study conducted in 1988 (Hirst 1988). We are not suggesting this 1988 study was not effective but simply that there is a pattern for older studies reporting higher estimates that dissipated over time, with the last decade reporting consistently lower estimates.

Source: Haapakoski et al. (2015)

We hope to have clarified the concerns raised and we have now improved our visual representation of cumulative meta-analysis (please see below revised Figure 2 – now Figure 3), adding a dotted line for the final average estimate and by editing the years in the y-axis for clarity. Now we present only the years where papers were available and not a continuous year sequence from the 1970's to 2017, which may have misled the reader.

Lau J., E. M. Antman, J. Jimenez-Silva, et al. 1992. Cumulative meta-analysis of therapeutic trials for myocardial infarction. *New England Journal of Medicine* 327: 248–54.

Sterne, J. (2008). Cumulative meta-analysis. State Technical Bulletin (page 13-16) <http://cphs.huph.edu.vn/uploads/tainguyen/sachvabaocao/Meta-AnalysisinStata.pdf#page=200>

Haapakoski, R., Mathieu, J., Ebmeier, K. P., Alenius, H., & Kivimäki, M. (2015). Cumulative meta-analysis of interleukins 6 and 1 β , tumour necrosis factor α and C-reactive protein in patients with major depressive disorder. *Brain, Behavior, and Immunity*, 49, 206-215.

Hirst E (1988) The Hood River Conservation Project: An evaluator's dream. *Evaluation Rev* 12:310 – 325.

Reviewers' comments:

Reviewer #1 (Remarks to the Author):

This revised paper has some strengths that make it deserving of publication, but also some key weaknesses—not so much in the data analysis as in the fundamental conceptualization underlying the analysis and in the conclusions it draws.

Its focus is on the effects of “behavioral interventions” on the “probability of benefit (POB),” which has similarities to the behavioral plasticity (BP) concept in the Dietz et al (2009) study it cites, but also key differences (discussed below). POB measures the proportion of the target population that changes its behavior as a function of the intervention. The paper develops a useful characterization of types of intervention and examines the effects of interventions that combine types of behavioral interventions on a set of behavioral types. Its findings showing different POB by intervention type and behavior type are useful and interesting, and worthy of further exploration.

The paper seeks rigorous analysis by including only field-based experimental studies with actual behavior (not intentions) as the outcome variable, and it examines effects holding regulatory and financial incentives constant, thus avoiding analysis of interaction effects of behavioral interventions with other factors. This is a strength in one way: it gives a good summary for researchers who may presume that behavioral interventions acting alone can make a major contribution to limiting climate change.

However, it is also a weakness. The literature has long shown that the strongest potential for behavioral interventions to yield both behavioral change and reduced greenhouse gas (GHG) emissions lies not in behavioral interventions acting alone but in their inclusion in programs that integrate behavioral interventions with other interventions, such as financial incentives. This idea lay behind the effort of Dietz et al. (2009) to define BP by using the greatest behavior change that had been demonstrated with interventions that included behavioral change within the context of other influences present in the choice environment—not as the effects of behavioral change alone. This choice was influenced by long-standing evidence that the effects of financial incentives can be strongly affected by nonfinancial aspects of the incentive programs. For example, identical financial incentives for home weatherization offered through different utility companies have shown a tenfold or greater variation in the uptake of the incentives, depending on how the programs were implemented, including how they were marketed (see Stern et al., 1985). Such findings suggest that the strongest effects of behavioral interventions lie in their interactions with other intervention types.

Thus, it seems to me that this paper’s conclusion that behavioral interventions have only minimal usefulness in efforts to reduce GHG emissions is misleading if, as past research indicates, their greatest potential lies in combination with incentives or other intervention types. The paper says on p. 12: “The available field experiments do not give rise to hopeful predictions about relying on the contribution of behavioural interventions to tackle climate change.” That may well be true of behavioral interventions alone, but not when not combined with other intervention types. It also notes: “There is the possibility that behavioural effects could vary depending on the policy or economic forces concurrently in place or that specific combinations of behavioural strategies with legislation and/ or financial incentives produce better results. However, this is an open empirical question” with “no clear causal support”. If the only empirical support of any value must come from controlled field experiments, this may be true, but in my view, other types of field studies can add important value to our understanding of what works.

A recent review summarizing contributions of psychology to limiting climate change (Wolske & Stern, 2018), which focuses on implications of behavioral change for reducing GHG emissions (not only POB) identifies several tentative conclusions that involve combining behavioral interventions with other interventions, noting, for instance, that one function of financial incentives is to get

consumers to pay attention to an incentivized choice, after which behavioral interventions can have much larger effects than they can have on their own, in populations most of which are not already considering high-impact choices being incentivized. I would find this paper more valuable if it took a broader view of what kinds of behavioral evidence deserve consideration and as a result, was more circumspect about drawing conclusions about the ways in which behavioral interventions may matter. Doing this would make it a more important contribution, even though less expansive regarding its conclusion that behavioral interventions effectively don't matter. This view about the importance of behavioral interventions, implicit in many regulatory and financial incentive-based policy prescriptions, has proven seriously incomplete for decades.

Regarding a decision about publication, my recommendation is that this paper deserves publication only under one of two conditions: (1) that the author(s) be more circumspect about the paper's broad conclusions, as indicated above, or (2) that the paper appear alongside a commentary raising the issues raised in these review comments. I would prefer condition (1), as commentaries are less likely to be read than main papers. But whether to be more circumspect is ultimately the authors' choice.

References: Stern, P.C. et al. (1985). The effectiveness of incentives for residential energy conservation. *Evaluation Review*, 10, 147-176.

Wolske, K.S, and Stern, P.C. (2018). Contributions of psychology to limiting climate change: Opportunities through consumer behavior. Pp. 127-160 in *Psychology and Climate Change* (S. Clayton & C. Manning, eds.). Academic Press

Reviewer #2 (Remarks to the Author):

Thank you for the two primary clarifications on my prior comments. I particularly find the revised Table SI.2 helpful. The authors' remark on interdisciplinary collaboration also serves as a good argument to address the challenge of existing behavioral interventions (lack of an integrated framework or formula to follow in terms of study design to solve behavioral problems).

Regarding the authors' comments on making policy recommendations, I need to clarify that the policy-oriented recommendations I refer to serve more as "study implications". I believe the paper is in a good position to make these implications given the findings presented. More specifically, I believe the following points are important references for interdisciplinary researchers who are designing behavior interventions to address issues such as global warming and health problems.

These points are: (1) A need to examine the technical impact of behaviors under study. (2) Avoid using bundles of stimuli as this makes the key driver of effectiveness difficult to grasp. (3) Stronger investment in the proximal drivers of behavior change such as social comparison and choice architecture. More impactful interventions may imply interventions that depend on a minimum effort and willpower from individuals. (4) Traditional policy instruments have very limited effectiveness, such as the public service announcements, statistics, feedback or energy labels.

I think it is important to connect these points back to how we should better approach and design future behavior interventions (for global warming or other issues in general) as one of the main contributions of the paper. Comparing the extent to which behavioral estimates are more accurate is valuable information to have, yet, I think elaborations on implications toward behavior intervention study design is equally important. After reading the authors' rebuttal, I feel one message the authors would like to convey is that behavior interventions are not effective approach to address the global warming issue (not as good as what data from prior studies suggest). And if that's the case, what would the authors suggest us to take as the next step? Are we saying to decrease investment on behavior interventions in the public sector, or we should focus on using

and evaluating certain intervention strategies? I think making this component more explicit would make the paper stronger.

Reviewer #3 (Remarks to the Author):

COMMENTS FOR AUTHOR(S)

I still have difficulty replicating the numbers reported in Table 1 (and elsewhere), using the updated spreadsheet provided. For this reason, I remain unable to support publication. The authors offered a long response to my initial comments, some of which I applaud. However, the authors I think missed my main point – if I cannot match descriptives such as the sum of N, IRRESPECTIVE of how N may or may not have been used anywhere else, then there is simply a discrepancy that needs to be clarified. In the presence of any such discrepancy, and given the level of academic rigor expected, it is reasonable to request clarification from the authors. While the authors have admittedly taken some steps to eliminate my initial concerns, the problem still appears to remain. Moreover, it is not simply for the sum of N that the problem exists – there are other simple summaries by which our estimates differ. Here I offer more detail and examples with the hope that the authors can help clarify any differences in steps taken. Presumably some of the effort to explain to me the differences would need to be reflected inside the manuscript also.

For the clarity of the authors, I am not negative towards the work, but I am also not in a position to give it my support yet. I am inclined to think that the analysis done by the authors is not erroneous (all indications suggest they are ‘capable’ scholars) – yet taking the data provided and following the steps described, I simply cannot verify the numbers. My assumption is that the scholars are making some restrictions that I am not, but have not made it clear i.e. that our discrepancies are by differences in assumption(s) as opposed to problems (so to speak) with the analysis steps taken. I hope that the authors can appreciate however, that I cannot merely speculate over this, and have a level of ‘duty’ to understand the discrepancies. This would also imply a need to modify the text to ensure that various assumptions made in summarizing the data are less ambiguous.

Here I will offer more detail on the steps taken to try and verify numbers. I include some of the R code, so that the authors may then help advise on any mistakes I am making. The simplest option would be for the authors to provide their STATA code for verification. I see no reason why they would not do so in a confidential review process.

Let us begin with Table 1, and some of the elements of column 1 (variable ‘k’ the number of studies):
creates no differences in the data)

- Then I sum the number of observations by category, a few examples are given below to show these match:

```
> data = read.csv("data.csv")
> length(data[,1])
[1] 144
> length( which(data$Households==1) )
[1] 66
> length( which(data$Households==0) )
[1] 78
> length( which(data$Small_N0.100==1) )
[1] 82
> length( which(data$Medium_N101.500==1) )
```

```
[1] 45
> length( which(data$Large_N500.==1) )
[1] 17
```

- All Okay to this point. Now I want to try to replicate 'N'.

```
> data$N = data$N_EG+data$N_CG
> sum(data$N)
[1] 3529803
```

- A 'raw' total does not match the value of N = 3,092,678 reported in Table 1, and the difference is material (around 500,000 observations) .
- I then consider the possibility that the authors are instead summing up the number of observations by publication (i.e. removing instances of multiple treatments). This just seems like a reasonable alternative route to obtaining the sample size, but again the results differ:

```
for (i in 1:length(data[,1]))
{
data$paper[i] = paste(strsplit(as.character(data[i,1]),as.character(data[i,2]))[[1]][1], data[i,2], sep="")
}
sum(
aggregate(
(data$N_EG+data$N_CG), by=list(data$paper), FUN=function(x) max(x)
)[,2]
,na.rm=TRUE)
2945795
```

- There remains a discrepancy. The numbers which are calculated using the data in the spreadsheet are either larger, or smaller than that reported by the authors, but not the same. I cannot find any discussion that would tell me what to do to arrive at the exact number reported in the table.

Thus, to this point, my concern remains the same as my stated concern on the original submission. Using the information provided by the authors, even some of the simple summary statistics cannot be replicated.

I next turn attention to obtain an estimate of the third column ('d', the effect size). Intuitively one would expect the number here to be either a mean or median, so I first calculated those:

```
> mean( data$SMD )
[1] -0.286975
> median( data$SMD )
[1] -0.1901
```

- These numbers are both much larger than the value of -0.093 reported in Table 1 for the full sample (first row).
- So, let us take a glance at the distribution. I recovered an estimate of the kernel density, then located the value of 'd' with the highest probability. The code/plot below shows the distribution, with lines indicating the value reported in Table 1 (blue) and that implied by the maximum density estimate (red). The difference is not even visible given the scale of the x-axis:

```

> max.dense( data$SMD[which(data$Households==1)] )
[1] -0.148309
> plot(density(data$SMD))
> max.dense = function(x){
+ temp = density(x)
+ temp$x[ which(temp$y == max(temp$y) ) ]
+ }
> abline(v=-0.093, col="blue") # (from table provided by authors)
> abline(v=max.dense( data$SMD ), col="red") # (my calculation)
> max.dense( data$SMD )
[1] -0.09477717

```

- This is encouraging – but what happens when we try to apply this to the second and third rows i.e. for households and individuals respectively:

```

> plot(density( data$SMD[which(data$Households==1)] ))
> abline(v=-0.112, col="blue") # (from table provided by authors)
> abline(v=max.dense( data$SMD[which(data$Households==1)] ), col="red") # (my calculation)
> max.dense( data$SMD[which(data$Households==1)] )
[1] -0.148309

```

density.default(x = data\$SMD[which(data\$Households == 1)])

```
> plot(density( data$SMD[which(data$Households==0)] ))  
> abline(v=-0.118, col="blue") # (from table provided by authors)  
> abline(v=max.dense( data$SMD[which(data$Households==0)] ), col="red") # (my calculation)  
> max.dense( data$SMD[which(data$Households==0)] )  
[1] -0.08294122
```

density.default(x = data\$SMD[which(data\$Households == 0)])

- There continue to be discrepancies.
- I simply cannot be certain why, and as I mentioned above, while I can see your reported numbers are close to my calculations. Following reasonable steps, they are not identical. Software differences that may lead to different numerical estimates would not normally be sufficient to reconcile the differences I am observing.

Let us next turn to another component of the study – I will try to approximate the funnel plot in Fig. 2:

```
> library(metafor)
> ### fit fixed-effects model
> res <- rma(SMD, sd, data=data, measure="IRR", method="FE")
>
> funnel(res, main="Standard Error", level=c(.1,.05,.01),
+ ylim=c(1.5,0), xlim=c(-4,4),shade=c("white","gray60","gray70"));abline(h=1.25, col="red")
>
```

- I am not being very careful in the above application (as you note also, the STATA/R routines may arrive at different parameterizations owing to a number of reasons), but clearly the results here are similar.
- This is a positive signal to me that the authors are ‘capable’ scholars (for want of a better term) – increasing my confidence that the analysis is probably okay. Yet this still does not permit me to reconcile the descriptive statistics (i.e. those parts of the analysis that in principle cannot differ between different software).

Lastly, let us consider an attempt to replicate the trends in Fig. 2. I found this to be a very misleading graph (the plot seems to suggest that there were an equal number of estimates generated in any given

year given the equidistant spacing of the observations – for example it looks like there are several observations in 1976, when in fact there is only a single observation in the dataset), and the reported numbers prove difficult to reproduce.

The authors claim this to show “cumulative effect sizes over time”. The graph shows more observations than there are years – but how is this feasible? Do you date the authors to the month/day if so where is this information available? I instead (given the information in the provided data), summarized the cumulative effect size to the end of the most recent year. Given, from Table 1, you seem to report something close to the maximum density estimates of ‘d’, I use that as the way of calculating the cumulative effect here.

I present my numbers in time-series format, and only focus on the center of the distribution (I recognize you show tails that I do not report here). The basic pattern in the data shares some similarities, yet some non-trivial differences also. For example, your Fig 3. Suggests that the cumulative effect size had dropped to around -0.09, by around 2006. My calculations on the other hand imply this was not the case until around 2013. The conclusion is simple – we do not agree on the time-series properties implied by the data, and the description of

```
> plot_data = NULL
> for (t in unique(1978:2017)){
+ RANGE=1:t
+ temp = data$SMD[which(data$Year%in%RANGE)]
+ plot_data = rbind(plot_data, max.dense(temp) )
+ }
> plot( ts( plot_data, end=2017 ) )
```

I am willing to concede that the steps taken by the authors are preferable to my own – but I simply don't know with completeness what these steps are.

In summary, and with respect towards the authors, I am unable to verify the analysis using reasonable effort and taken reasonable steps on the basis of either norms in data analysis, or my most reasonable guess of the steps taken by the authors. Until I can do so, I am unable to recommend this study to proceed through to final publication. I require that the authors please more directly address the issue of replication. I reiterate – if you provide the code to replicate, I can do this most efficiently. This is the same polite request I made at the initial review. I am cautiously optimistic that the differences between the authors steps and my own, will in the end be reasonably modest.

Reviewer #1

This revised paper has some strengths that make it deserving of publication, but also some key weaknesses—not so much in the data analysis as in the fundamental conceptualization underlying the analysis and in the conclusions it draws.

Its focus is on the effects of “behavioral interventions” on the “probability of benefit (POB),” which has similarities to the behavioral plasticity (BP) concept in the Dietz et al (2009) study it cites, but also key differences (discussed below). POB measures the proportion of the target population that changes its behavior as a function of the intervention. The paper develops a useful characterization of types of intervention and examines the effects of interventions that combine types of behavioral interventions on a set of behavioral types. Its findings showing different POB by intervention type and behavior type are useful and interesting, and worthy of further exploration. The paper seeks rigorous analysis by including only field-based experimental studies with actual behavior (not intentions) as the outcome variable, and it examines effects holding regulatory and financial incentives constant, thus avoiding analysis of interaction effects of behavioral interventions with other factors. This is a strength in one way: it gives a good summary for researchers who may presume that behavioral interventions acting alone can make a major contribution to limiting climate change.

However, it is also a weakness. The literature has long shown that the strongest potential for behavioral interventions to yield both behavioral change and reduced greenhouse gas (GHG) emissions lies not in behavioral interventions acting alone but in their inclusion in programs that integrate behavioral interventions with other interventions, such as financial incentives. This idea lay behind the effort of Dietz et al. (2009) to define BP by using the greatest behavior change that had been demonstrated with interventions that included behavioral change within the context of other influences present in the choice environment—not as the effects of behavioral change alone. This choice was influenced by long-standing evidence that the effects of financial incentives can be strongly affected by nonfinancial aspects of the incentive programs. For example, identical financial incentives for home weatherization offered through different utility companies have shown a tenfold or greater variation in the uptake of the incentives, depending on how the programs were implemented, including how they were marketed (see Stern et al., 1985). Such findings suggest that the strongest effects of behavioral interventions lie in their interactions with other intervention types.

Thus, it seems to me that this paper’s conclusion that behavioral interventions have only minimal usefulness in efforts to reduce GHG emissions is misleading if, as past research indicates, their greatest potential lies in combination with incentives or other intervention types. The paper says on p. 12: “The available field experiments do not give rise to hopeful predictions about relying on the contribution of behavioural interventions to tackle climate change.” That may well be true of behavioral interventions alone, but not when not combined with other intervention types. It also notes: “There is the possibility that behavioural effects could vary depending on the policy or economic forces concurrently in place or that specific combinations of behavioural strategies with legislation and/ or financial incentives produce better results. However, this is an open empirical question” with “no clear causal support”. If the only empirical support of any value must come from controlled field experiments, this may be true, but in my view, other types of field studies can add important value to our understanding of what works.

A recent review summarizing contributions of psychology to limiting climate change (Wolske & Stern, 2018), which focuses on implications of behavioral change for reducing GHG emissions (not only POB) identifies several tentative conclusions that involve combining behavioral interventions with other interventions, noting, for instance, that one function of financial incentives is to get consumers to pay attention to an incentivized choice, after which behavioral interventions can have much larger effects than they can have on their own, in populations most of which are not already considering high-impact choices being incentivized. I would find this paper more valuable if it took a broader view of what kinds

of behavioral evidence deserve consideration and as a result, was more circumspect about drawing conclusions about the ways in which behavioral interventions may matter. Doing this would make it a more important contribution, even though less expansive regarding its conclusion that behavioral interventions effectively don't matter. This view about the importance of behavioral interventions, implicit in many regulatory and financial incentive-based policy prescriptions, has proven seriously incomplete for decades.

Regarding a decision about publication, my recommendation is that this paper deserves publication only under one of two conditions: (1) that the author(s) be more circumspect about the paper's broad conclusions, as indicated above, or (2) that the paper appear alongside a commentary raising the issues raised in these review comments. I would prefer condition (1), as commentaries are less likely to be read than main papers. But whether to be more circumspect is ultimately the authors' choice.

References

*Stern, P.C. et al. (1985). The effectiveness of incentives for residential energy conservation. *Evaluation Review*, 10, 147-176.*

*Wolske, K.S, and Stern, P.C. (2018). Contributions of psychology to limiting climate change: Opportunities through consumer behavior. Pp. 127-160 in *Psychology and Climate Change* (S. Clayton & C. Manning, eds.). Academic Press.*

*Reply to Reviewer #1: We are extremely grateful for Reviewer 1's comments. These were very useful to refine the conceptual scope of our work. We would also like to acknowledge that Reviewer 1 accurately summarises our paper: Our goal was to evaluate the impact of behavioural interventions in promoting climate change mitigation, by selecting research conducted with the highest methodological rigour (RCTs), which, by virtue of their design, hold all other things constant (ceteris paribus). The reviewer is also correct in saying that we present results that warn against depending *solely* on behavioural interventions to mitigate climate change. We agree that economic incentives and regulations may moderate the influence of behavioural interventions intended to mitigate climate change. In line with Reviewer 1's comments, the revised manuscript discusses this point in further detail. Below is the revised Discussion section, which can also be found in track changes on page 12:*

“The available field experiments do not give rise to hopeful predictions about relying on the individual contribution of behavioural interventions to tackle climate change. Our results show that it is unlikely that behavioural interventions, in isolation, will help us achieve large and pressing reductions in carbon emissions. There is, however, the possibility that behavioural effects could vary depending on the policy or economic forces concurrently in place, or that specific combinations of behavioural interventions with legislation and/ or financial incentives produce better results (32-34). Indeed, there is evidence strongly suggesting an interactive effect. For instance, financial incentive programs for home insulation that required prior energy audits were less effective in promoting conservation actions compared to programs that offered similar monetary incentives, but that did not involve requesting an audit (35-36). Another example provided by Stern and colleagues (36) reported that the same program to install energy-saving equipment in households produced very different uptake rates for energy audits (up to a 20% difference), depending on the source of the information letter received. These same authors reviewed multiple conservation programs in the U.S. combining both financial and non-financial interventions and concluded that, controlling for type and size of incentives, participation in conservation programs could vary by a factor of 10, depending on concomitant behavioural strategies, such as perceived social norms or message framing. This seminal evidence suggests

that financial incentives capture consumers' attention, after which behavioural interventions can have larger effects than if used in isolation.

An important point to emphasize is that our results do not suggest that behavioural interventions are less effective than financial incentives or regulations in mitigating climate change. Ultimately, our work shows that mitigating climate change via voluntary behavioural changes in households and individuals is a real challenge - not that behavioural interventions are less effective than alternative approaches. Our results do not prioritize regulatory and financial incentive-based policy prescriptions above behavioural interventions. These too should be subject to rigorous evaluation."

32. Stern, P. C. (2011). Contributions of psychology to limiting climate change. American psychologist, 66(4), 303-314.
33. Wolske, K.S. and Stern, P.C. (2018). Contributions of psychology to limiting climate change: Opportunities through consumer behavior. Pp. 127-160 in Psychology and Climate Change (S. Clayton & C. Manning, eds.). Academic Press.
34. Swim, J. K., Stern, P. C., Doherty, T. J., Clayton, S., Reser, J. P., Weber, E. U., & Howard, G. (2011). Psychology's contributions to understanding and addressing global climate change. American psychologist, 66(4), 241-250.
35. Stern, P. C. (1999). Information, incentives, and proenvironmental consumer behavior. Journal of consumer Policy, 22(4), 461-478.
36. Stern, P. C., Aronson, E., Darley, J. M., Hill, D. H., Hirst, E., Kempton, W., & Wilbanks, T. J. (1985). The effectiveness of incentives for residential energy conservation. Evaluation Review, 10(2), 147-176.
37. Stern, P. C. (1986). Blind spots in policy analysis: What economics doesn't say about energy use. Journal of Policy Analysis and management, 5(2), 200-227.

Reviewer #2

Thank you for the two primary clarifications on my prior comments. I particularly find the revised Table SI.2 helpful. The authors' remark on interdisciplinary collaboration also serves as a good argument to address the challenge of existing behavioral interventions (lack of an integrated framework or formula to follow in terms of study design to solve behavioral problems).

Regarding the authors' comments on making policy recommendations, I need to clarify that the policy-oriented recommendations I refer to serve more as "study implications". I believe the paper is in a good position to make these implications given the findings presented. More specifically, I believe the following points are important references for interdisciplinary researchers who are designing behavior interventions to address issues such as global warming and health problems. These points are: (1) A need to examine the technical impact of behaviors under study. (2) Avoid using bundles of stimuli as this makes the key driver of effectiveness difficult to grasp. (3) Stronger investment in the proximal drivers of behavior change such as social comparison and choice architecture. More impactful interventions may imply interventions that depend on a minimum effort and willpower from individuals. (4) Traditional policy instruments have very limited effectiveness, such as the public service announcements, statistics, feedback or energy labels.

I think it is important to connect these points back to how we should better approach and design future behavior interventions (for global warming or other issues in general) as one of the main contributions of the paper. Comparing the extent to which behavioral estimates are more accurate is valuable information to have, yet, I think elaborations on implications toward behavior intervention study design is equally important. After reading the authors' rebuttal, I feel one message the authors would like to convey is that behavior interventions are not effective approach to address the global warming issue (not as good as what data from prior studies suggest). And if that's the case, what would the authors suggest us to take as the next step? Are we saying to decrease investment on behavior interventions in the public sector, or we should focus on using and evaluating certain intervention strategies? I think making this component more explicit would make the paper stronger.

Reply to Reviewer #1: We would also like to thank Reviewer 2 for all insightful comments. These were all incorporated in our revised Discussion, which now makes explicit recommendations for (interdisciplinary) research designing behavioural interventions to address important social issues. The revised Discussion section is shown below (also in pages 11-13 with track changes), with the following structure:

- Technical Impact
- Stimuli Bundle
- Research Design
- Relative Effectiveness
- Participant Naiveté
- Summary/ Next Steps

“We propose several recommendations for future (interdisciplinary) research examining complex issues such as global warming. First, more research is needed to tackle behaviours with high technical potential, that is, behaviours that greatly contribute to reduce carbon emissions. Comparative estimates (19) showed that, for the average households in North America and Europe, the actions with the largest potential for carbon emissions reduction are eating fewer animal products, reducing driving and flying, and saving household energy, particularly from heating and cooling. Programs and initiatives targeting, for instance, recycling, turning off the lights after leaving a room, or unplugging battery chargers after use, also contribute to mitigate climate change, but have substantially lower marginal effects – even if the interventions could achieve 100% behavioural plasticity. Ideally, interventions worthier”

of implementation should both achieve high behavioural plasticity and target behaviours with high technical potential.

Second, future work should try to isolate the main effects of information, social comparison, environmental appeals, etc., as well as their combined effects with other behavioural stimuli, or incentives and regulations. The content of interventions needs to be tested more precisely because a large proportion of interventions to date implements bundles of stimuli from which the identification of the key driver of effectiveness is difficult to grasp. The implicit assumption behind combining stimuli is that the intervention is more likely to be effective, the larger the number of stimuli involved. But we should not automatically assume positive additive or multiplicative effects just because a variety of actions is taken in combination. It is crucial to estimate the additional marginal contribution of multiple actions to mitigate climate change. Conclusively establishing main versus interactive effects requires comparative analyses, specifically designed for this purpose (38-40).

Third, and directly related to the previous point, future research should implement research designs that allow for causal inferences. Whereas all research designs are valid tools for knowledge production, establishing causality requires specific research designs. Not only randomised controlled trials (RCTs) can be used to achieve this, but they are the gold-standard to test causal effects. Future research should focus on establishing to what extent a particular intervention - and that intervention alone - contributes to behaviour change. This is established by a valid comparison group which (1) has the same characteristics, on average, as the treatment group in the absence of the intervention; and (2) remains unaffected by the intervention. Randomised assignment of participants is the most recommended procedure to achieve a valid comparison group. However, many real-world circumstances prevent this randomisation, but researchers can turn to alternative impact evaluation methods (41) such as instrumental variables (i.e., variables outside the control of individuals that influence participation in the intervention but are otherwise irrelevant), regression discontinuity design, differences-in-differences or matching (for when intervention assignment rules are not clear). Simple before-and-after analyses, or the descriptive comparisons of programs with different characteristics provide less trustworthy estimates.

Fourth, researchers should continue to invest efforts in interventions targeting direct triggers of behaviour change, such as social comparison messages and choice architecture (“nudges”). Although such interventions have their own set of challenges (e.g., unknown collateral effects of downward social comparisons, or ethical concerns with consent and autonomy in nudge interventions), our results indicate these are the most effective interventions. The fact that choice architecture is the most effective behavioural intervention strongly suggests that people are, on average, poorly motivated to mitigate climate change. Choice architecture relies on people preferring the path of least resistance, such as having to walk less to the recycling bin or not giving a second look at the air conditioning temperature. This approach could be implemented in less explored areas. For instance, utilities companies could implement a default (opt-out) payment for renewable energy (e.g., 42-43). Moreover, the layout of cafeterias, restaurants and supermarkets could be rearranged to display animal products in less accessible areas. This is a strategy commonly used in health promotion to reduce consumption of sugary foods, but seldom used to fight climate change. Given the pressing need to change the consumption patterns of millions of individuals and households, interventions should target proximal drivers of behaviour change – over and above the longstanding goal of changing hearts and minds. This is recommended because, conversely, traditional policy instruments such as public service announcements, statistics, consumption feedback, and energy labels, were found to be the least effective ways to produce behavioural change. The assumption about the positive role of information seems to die hard, despite consistent evidence of its low impact. Whereas this is an unsurprising result to behavioural scientists (25) and some results have pointed in this direction (7), there is a considerable gap

between what is known in behavioural science and what is proposed and implemented by policy-makers. These strategies continue to dominate the proposed solutions in the latest IPCC Special Report with respect to demand-side behaviour changes, and the last randomised trials published in 2018 persist on the idea of testing information-based strategies to promote mitigation behaviours (1). Information-based strategies are employed under the assumption that, when provided with facts and figures, people will adjust their behaviour and monitor their actions that produce carbon emissions (44-45). Such assumptions are challenged by our results, estimating the probability that information can produce positive changes in 3.4%. This is a result of paramount significance considering the central role these strategies are expected to play in the fight against global warming (46-47). Nonetheless, there is still the need to evaluate the conditions under which information strategies may be boosted by financial incentives (35-37). Whereas information has little impact to motivate unmotivated individuals, information may be crucial to guide behaviour once people are motivated by incentives.

Fifth, future research should enrol, to the extent possible, naïve participants. Our results showed a sevenfold difference between studies that required participants' explicit consent versus studies that had no such requirement. One explanation for this effect is that people that accept to enrol in environmental studies are more interested in climate change mitigation than people that decline such invitations. Moreover, participants that provide consent are aware that their behaviour is monitored, thus leading to greater desire to conform to what is normative (48). Researchers may correct this bias by (i) using publicly available data; (ii) requesting waived informed consent due to compelling validity concerns (49); (iii) monitoring non-participants, at least at the aggregate level; and/ or (iv) correcting for the estimates collected with non-naïve participants. This an important question because scaling-up and implementing these interventions in a larger scale, possibly designed as policy-level action, necessarily targets a much wider variety of individuals with varying motivation to act pro-environmentally. Using the estimates from self-selected participants to inform policy makers creates an expectation that may not be matched when targeting larger populational segments.

In summary, our work speaks to the preferred options within the universe of behavioural interventions. Our recommendation is to increase the use of nudges and social comparison interventions, given that both types of interventions are independent of individuals' motivation to mitigate climate change. Furthermore, we recommend (i) a greater investment in RCTs or other research designs that allow to establish causality, (ii) targeting the most relevant behaviours to achieve carbon reductions, (iii) with naïve participants, and (iv) measuring tangible behaviour changes. Lastly, our results should not be used to argue for a disinvestment in behavioural interventions because this work did not compare behavioural interventions with alternative approaches. Alternative approaches should be subject to the same evaluation scrutiny.”

Reviewer #3

Comment #1: *I still have difficulty replicating the numbers reported in Table 1 (and elsewhere), using the updated spreadsheet provided. For this reason, I remain unable to support publication. The authors offered a long response to my initial comments, some of which I applaud. However, the authors I think missed my main point – if I cannot match descriptives such as the sum of N, IRRESPECTIVE of how N may or may not have been used anywhere else, then there is simply a discrepancy that needs to be clarified. In the presence of any such discrepancy, and given the level of academic rigor expected, it is reasonable to request clarification from the authors. While the authors have admittedly taken some steps to eliminate my initial concerns, the problem still appears to remain. Moreover, it is not simply for the sum of N that the problem exists – there are other simple summaries by which our estimates differ. Here I offer more detail and examples with the hope that the authors can help clarify any differences in steps taken. Presumably some of the effort to explain to me the differences would need to be reflected inside the manuscript also. For the clarity of the authors, I am not negative towards the work, but I am also not in a position to give it my support yet. I am inclined to think that the analysis done by the authors is not erroneous (all indications suggest they are ‘capable’ scholars) – yet taking the data provided and following the steps described, I simply cannot verify the numbers. My assumption is that the scholars are making some restrictions that I am not, but have not made it clear i.e. that our discrepancies are by differences in assumption(s) as opposed to problems (so to speak) with the analysis steps taken. I hope that the authors can appreciate however, that I cannot merely speculate over this, and have a level of ‘duty’ to understand the discrepancies. This would also imply a need to modify the text to ensure that various assumptions made in summarizing the data are less ambiguous. Here I will offer more detail on the steps taken to try and verify numbers. I include some of the R code, so that the authors may then help advise on any mistakes I am making. The simplest option would be for the authors to provide their STATA code for verification. I see no reason why they would not do so in a confidential review process. Let us begin with Table 1, and some of the elements of column 1 (variable ‘k’, the number of studies):*

- *I read the data into R (for simpler reading, I remove the top column and convert to a csv – which creates no differences in the data)*
- *Then I sum the number of observations by category, a few examples are given below to show these match:*

```
> data = read.csv("data.csv")
> length(data[,1])
[1] 144
> length( which(data$Households==1) )
[1] 66
> length( which(data$Households==0) )
[1] 78
> length( which(data$Small_N0.100==1) )
[1] 82
> length( which(data$Medium_N101.500==1) )
[1] 45
> length( which(data$Large_N500.==1) )
[1] 17
```

- *All Okay to this point. Now I want to try to replicate ‘N’.*

```
> data$N = data$N_EG+data$N_CG
> sum(data$N)
[1] 3529803
```

- A 'raw' total does not match the value of $N = 3,092,678$ reported in Table 1, and the difference is material (around 500,000 observations).
- I then consider the possibility that the authors are instead summing up the number of observations by publication (i.e. removing instances of multiple treatments). This just seems like a reasonable alternative route to obtaining the sample size, but again the results differ:

```
for (i in 1:length(data[,1]))
{
data$paper[i] = paste(strsplit(as.character(data[i,1]),as.character(data[i,2]))[[1]][1], data[i,2], sep="")
}
sum(
aggregate(
(data$N_EG+data$N_CG), by=list(data$paper), FUN=function(x) max(x)
)[,2]
,na.rm=TRUE)
2945795
```

- There remains a discrepancy. The numbers which are calculated using the data in the spreadsheet are either larger, or smaller than that reported by the authors, but not the same. I cannot find any discussion that would tell me what to do to arrive at the exact number reported in the table. Thus, to this point, my concern remains the same as my stated concern on the original submission. Using the information provided by the authors, even some of the simple summary statistics cannot be replicated. Thus, to this point, my concern remains the same as my stated concern on the original submission. Using the information provided by the authors, even some of the simple summary statistics cannot be replicated.

Reply to Comment #1: We very much appreciate the Reviewer’s careful engagement with our work. We are happy to clarify. As the Reviewer realised, adding up the total samples per experimental and control group returned a number higher than we reported. The second approach – number of observation per publication - is somewhat closer to our procedure. However, in this second approach the Reviewer restricted the sum to the estimates from each paper with the maximum number of participants, leaving some observations out. We now provide an excel document just dedicated to the explanation of the total sample size – please refer to document **Sample_Size_Procedure.xlsx**. We also provide a summary explanation here.

Many papers tested more than one intervention; however, studies only have one control group. Let’s take the example of Paper X, which randomised participants (N=300) to 3 groups: one control group (N=101), treatment 1 (N=97) and treatment 2 (N=102). You would see the following in the database:

	N_EG	N_CG
Paper Xtreat1	97	101
Paper Xtreat2	102	101

The total sample size for Paper X is 300 – and not 401 - because the same control group was used for comparison. Therefore, the calculation of the total sample size needs to be adjusted for a duplicate use of the same control group, as follows:

	N_EG	N_CG
Paper Xtreat1	97	101
Paper Xtreat2	102	101

There are also a couple of other papers which tested more than one behaviour. For instance, Paper X tested the same intervention in more than one behaviour e.g., energy and water consumption; or recycling of paper and recycling of glass. We provide a small example here:

	N_EG	N_CG
Paper X_water	205	195
Paper X_energy	205	195

The total sample for Paper X is 400 (N=205 experimental group + N=195 control group) because these are the same participants in both cases. This study simply tested the effect of the same intervention on both water and energy, which provide two different estimates. Therefore, we adjusted the total sample as follows:

	N_EG	N_CG
Paper X_water	205	195
Paper X_energy	205	195

These examples illustrate the root of the discrepancies found by Reviewer #3. The total sample may not be captured by an automated calculation, due to these complexities. There are, in particular, four papers (Friis et al 2017; Holland et al 2006; Kurz et al 2005; Schultz et al 2016) that required a more detailed explanation, and we present this information in the excel file uploaded as Supplementary File. There, we detail all steps taken.

Comment #2: *I next turn attention to obtain an estimate of the third column ('d', the effect size). Intuitively one would expect the number here to be either a mean or median, so I first calculated those:*

```
> mean( data$SMD )
```

```
[1] -0.286975
```

```
> median( data$SMD )
```

```
[1] -0.1901
```

- These numbers are both much larger than the value of -0.093 reported in Table 1 for the full sample (first row).

So, let us take a glance at the distribution. I recovered an estimate of the kernel density, then located the value of 'd' with the highest probability. The code/plot below shows the distribution, with lines indicating the value reported in Table 1 (blue) and that implied by the maximum density estimate (red). The difference is not even visible given the scale of the x-axis:

```
> max.dense( data$SMD[which(data$Households==1)] )
```

```
[1] -0.148309
```

```
> plot(density(data$SMD))
```

```
> max.dense = function(x){
```

```
+ temp = density(x)
```

```
+ temp$x[ which(temp$y == max(temp$y) ) ]
```

```
+ }
```

```
> abline(v=-0.093, col="blue") # (from table provided by authors)
```

```
> abline(v=max.dense( data$SMD ), col="red") # (my calculation)
```

```
> max.dense( data$SMD )
```

```
[1] -0.09477717
```

Reply to Comment #2: Please note that a meta-analysis is not a simple mean (nor median) of estimates from different studies. Due to differing sample sizes and populations, each study has a different level of sampling error. A problem in combining studies is the assignment of weights that reflect the relative "value" of the information provided in each study. If all the weights were the same, then meta-analysis would be equal to the simple mean intervention effect.

This weighting process may differ according to the meta-analysis model used. We conducted a **random-effects meta-analysis**, which assumes that different studies are estimating different, yet related, intervention effects. This is the recommended assumption given the heterogeneous and large number of papers included [*versus fixed-effect meta-analysis, which assumes that the true effect of intervention is the same value in every study and fixed across studies*]. As the weighting process, we used the simplest version known as the **DerSimonian and Laird method [DL]** (1986, 2007, 2015). This inverse-variance method is used by default in most random-effects meta-analysis software. All these details are described in our Methods section.

DerSimonian, R., & Laird, N. (1986). Meta-analysis in clinical trials. *Controlled clinical trials*, 7(3), 177-188.

DerSimonian, R., & Kacker, R. (2007). Random-effects model for meta-analysis of clinical trials: an update. *Contemporary clinical trials*, 28(2), 105-114.

DerSimonian, R., & Laird, N. (2015). Meta-analysis in clinical trials revisited. *Contemporary clinical trials*, 45, 139-145.

The command used in STATA was **metaan** <https://www.stata-journal.com/article.html?article=st0201> . This command was used with the DerSimonian and Laird method as follows: **metaan SMD se, dl label(Study)**

A very similar command in R is **metagen**, which was used to replicate most analyses, as follows: https://bookdown.org/MathiasHarrer/Doing_Meta_Analysis_in_R/random.html

```
library(meta)
library(metafor)
# Loading
```

```
library("readxl")
# xls files
data <- read_excel("Revised_database_for_R.xlsx")

# Random-effects meta-analysis with pre-calculated estimates (SMD) and using DerSimonian and Laird method
```

```
m.hksj<-metagen(data$SMD,
  data$se,
  data=data,
  studlab=paste(data$Study),
  comb.fixed = FALSE,
  comb.random = TRUE,
  method.tau = "DL",
  hakn = TRUE,
  prediction=TRUE,
  sm="SMD")
```

We present both STATA and R estimates below. Only the confidence intervals present minor variations, narrower in R.

Overall Estimate	SMD	LB CI	UB CI	I ²
STATA	-0.093	-0.160	-0.055	64.6%
R	-0.093	-0.123	-0.063	64.6%

Comment #3: *This is encouraging – but what happens when we try to apply this to the second and third rows i.e. for households and individuals respectively:*

```
> plot(density( data$SMD[which(data$Households==1)] ))
> abline(v=-0.112, col="blue") # (from table provided by authors)
> abline(v=max.dense( data$SMD[which(data$Households==1)] ), col="red") # (my calculation)
> max.dense( data$SMD[which(data$Households==1)] )
[1] -0.148309
```

```

> plot(density( data$SMD[which(data$Households==0)] ))
> abline(v=-0.118, col="blue") # (from table provided by authors)
> abline(v=max.dense( data$SMD[which(data$Households==0)] ), col="red") # (my calculation)
> max.dense( data$SMD[which(data$Households==0)] )
[1] -0.08294122

```

- There continue to be discrepancies.
- I simply cannot be certain why, and as I mentioned above, while I can see your reported numbers are close to my calculations. Following reasonable steps, they are not identical. Software differences that may lead to different numerical estimates would not normally be sufficient to reconcile the differences I am observing.

Reply to Comment #3: Please note that the case here is exactly the same as for the calculation of the overall effect above. This requires the use of meta-analytic analyses as follows:

Example for Households vs. Individuals

```

x.sub1 <- subset(data,data$Households==1)
x.sub2 <- subset(data,data$Households==0)

```

```

m.sub1<-metagen(x.sub1$SMD,
  x.sub1$se,
  data=x.sub1,
  studlab=paste(x.sub1$Study),
  comb.fixed = FALSE,
  comb.random = TRUE,
  method.tau = "DL",
  hakn = TRUE,
  prediction=TRUE,
  sm="SMD")

```

Households	SMD	LB CI	UB CI	I ²
STATA	-0.112	-0.221	-0.057	73.1%
R	-0.112	-0.162	-0.062	73.1%

```

m.sub2<-metagen(x.sub2$SMD,
  x.sub2$se,
  data=x.sub2,
  studlab=paste(x.sub2$Study),
  comb.fixed = FALSE,
  comb.random = TRUE,
  method.tau = "DL",
  hakn = TRUE,
  prediction=FALSE,
  sm="SMD")

```

Individuals	SMD	LB CI	UB CI	I ²
STATA	-0.118	-0.221	-0.060	51.9%
R	-0.118	-0.166	-0.071	52.0%

Comment #4: *Let us next turn to another component of the study – I will try to approximate the funnel plot in Fig. 2:*

```

> library(metafor)
> ### fit fixed-effects model
> res <- rma(SMD, sd, data=data, measure="IRR", method="FE")
>
> funnel(res, main="Standard Error", level=c(.1,.05,.01),
+ ylim=c(1.5,0), xlim=c(-4,4),shade=c("white","gray60","gray70"));abline(h=1.25, col="red")
>

```

- I am not being very careful in the above application (as you note also, the STATA/R routines may arrive at different parameterizations owing to a number of reasons), but clearly the results here are similar.
 - This is a positive signal to me that the authors are ‘capable’ scholars (for want of a better term) – increasing my confidence that the analysis is probably okay. Yet this still does not permit me to reconcile the descriptive statistics (i.e. those parts of the analysis that in principle cannot differ between different software).

Reply to Comment #4: We provide the command in R for this analysis. Please note the Reviewer used a fixed-effects model (instead of random-effects) and the estimate Incidence Rate Ratio (instead of SMD). This was the graph we presented in the paper (page 6) using STATA (confunnel command <https://www.stata-journal.com/sjpdf.html?articlenum=gr0033>)

Below is the graph produced in R using the command for small-study bias (funnel):

Small-study bias

```
funnel(m.hksj, xlab="Effect estimate", col="black",cex=1.5,hlines="red",bg='black',ylim=(c(1.1, 0)),
  contour = c(.95,.975,.99), col.contour=c("darkgray","gray","lightgray"))+
  legend(1.4, 0, c("Studies", "< 0.01", "p<0.025", "p < 0.05", "p>0.05"),bty = "o",
  pch=c(16,NA,NA,NA,NA),
  fill=c("white", "lightgray", "gray", "darkgray", "white"),bg='white',border = "black")
```

Comment #5: *Lastly, let us consider an attempt to replicate the trends in Fig. 2. I found this to be a very misleading graph (the plot seems to suggest that there were an equal number of estimates generated in any given year given the equidistant spacing of the observations – for example it looks like there are several observations in 1976, when in fact there is only a single observation in the dataset), and the reported numbers prove difficult to reproduce.*

The authors claim this to show “cumulative effect sizes over time”. The graph shows more observations than there are years – but how is this feasible? Do you date the authors to the month/day if so where is this information available? I instead (given the information in the provided data), summarized the cumulative effect size to the end of the most recent year. Given, from Table 1, you seem to report something close to the maximum density estimates of ‘d’, I use that as the way of calculating the cumulative effect here.

I present my numbers in time-series format, and only focus on the center of the distribution (I recognize you show tails that I do not report here). The basic pattern in the data shares some similarities, yet some non-trivial differences also. For example, your Fig 3. Suggests that the cumulative effect size had dropped to around -0.09, by around 2006. My calculations on the other hand imply this was not the case until around 2013. The conclusion is simple – we do not agree on the time-series properties implied by the data, and the description of

```
> plot_data = NULL
> for (t in unique(1978:2017)){
+ RANGE=1:t
+ temp = data$SMD[which(data$Year%in%RANGE)]
+ plot_data = rbind(plot_data, max.dense(temp) )
+ }
> plot( ts( plot_data, end=2017 ) )
```

I am willing to concede that the steps taken by the authors are preferable to my own – but I simply don’t know with completeness what these steps are.

In summary, and with respect towards the authors, I am unable to verify the analysis using reasonable effort and taken reasonable steps on the basis of either norms in data analysis, or my most reasonable guess of the steps taken by the authors. Until I can do so, I am unable to recommend this study to proceed through to final publication. I require that the authors please more directly address the issue of replication. I reiterate – if you provide the code to replicate, I can do this most efficiently. This is the same polite request I made at the initial review. I am cautiously optimistic that the differences between the authors steps and my own, will in the end be reasonably modest.

Reply to Comment #5: We present below the equivalent R command that replicates our graph. Please note that here too is necessary to apply meta-analytic analyses. The R command used above (**metagen**) does not have the features necessary to produce the cumulative graph. We used an alternative command (**rma**) to replicate this graph.

#Fit a simple random-effects meta-analytic model

```
random = rma(yi=data$SMD, vi=data$se, method="DL", test="t",slab=paste(data$Study),tau2="TRUE")
tmp <- cumul(random, order=order(data$Year), transf=TRUE)
forest(tmp, xlim=c(-4,2), at=(c(-2, -1, -0.5, -.1,0)),
       digits=c(2,2), cex=0.5, yaxt="n" )
```

We agree with the reviewer that our cumulative graph needs improvement. This cumulative graph (both in STATA and R) is visually hard to interpret due to the high number of studies included. Thus, we adjusted the visual presentation of this graph to a simpler layout, more reader-friendly for interpretation. But we created some imprecisions in its visual presentation in the previous versions, which we aim to correct now. We now only present selected years that mark changes in the cumulative estimate, in terms of its interpretation according to Cohen’s *d* guidelines. We also present the raw estimates in the Table below, so that the Reviewer is able to verify the estimates. We are happy to work on it further if the Reviewer considers this necessary.

Cumulative Meta-analysis Estimates (from STATA)

We present the cumulative estimates here as shown in STATA (**command metacum**: metacum SMD se, effect(r) id(Year)). With each new entrance, the overall estimate is update, but also considering the relative weight of each paper. These cumulative estimates fluctuate up and down, but we marked the point after which the estimate doesn't go higher than a certain threshold. We followed Cohen's *d* interpretation:

0.7 and above – Large effect size; 0.5-0.6 Medium effect size; 0.3-0.4 Medium-small effect size; 0.1-0.2 Small effect size; Below 0.1 – Very small effect size

	Study	Cumulative d	LB CI	UB CI
LARGE TO MEDIUM EFFECTS SIZE	Seligman & Darley 1976	-0.845	-1.605	-0.085
	Becker 1978 treat1	-0.707	-1.094	-0.319
	Becker 1978 treat2	-0.657	-1.015	-0.299
	Winett et al 1979 treat2	-0.648	-0.969	-0.326
	Winett et al 1979 treat1	-0.741	-1.043	-0.440
	Katzev et al 1980	-0.695	-0.979	-0.411
	McCaul & Kopp 1982	-0.674	-0.896	-0.452
	Anderson & Claxton 1982 treat1b	-0.556	-0.842	-0.270
	Anderson & Claxton 1982 treat2b	-0.463	-0.754	-0.173
	Anderson & Claxton 1982 treat1d	-0.375	-0.659	-0.090
	Pardini & Katzev 1983	-0.396	-0.672	-0.120
	Burn & Oskamp 1986 treat2	-0.440	-0.712	-0.168
	Burn & Oskamp 1986 treat1	-0.492	-0.765	-0.220
	Katzev & Pardini 1987	-0.523	-0.788	-0.258
	Spaccarelli et al 1989	-0.496	-0.726	-0.266
	Thompson & Stoutmeyer 1991 treat2	-0.464	-0.679	-0.250
	Burn 1991 treat1	-0.520	-0.743	-0.298
	Burn 1991 treat2	-0.529	-0.744	-0.314
	MEDIUM-SMALL EFFECT SIZES	Hutton & Ahtola 1991	-0.479	-0.667
Thompson & Stoutmeyer 1991 treat1		-0.472	-0.652	-0.291
Dickerson et al 1992 treat2		-0.466	-0.641	-0.291
Dickerson et al 1992 treat1		-0.463	-0.634	-0.293
Werner et al 1995		-0.477	-0.645	-0.309
Cobern et al 1995		-0.455	-0.617	-0.294
Bryce et al 1997		-0.433	-0.582	-0.284
Tertololen et al 1998 treat3		-0.406	-0.550	-0.262
Schultz 1998 treat1		-0.388	-0.523	-0.252
Schultz 1998 treat2		-0.368	-0.497	-0.239
Tertololen et al 1998 treat1		-0.363	-0.488	-0.238
Tertololen et al 1998 treat2		-0.351	-0.471	-0.230
Schultz 1998 treat4		-0.343	-0.458	-0.228
Schultz 1998 treat3		-0.334	-0.444	-0.223
Mutrie et al 2002 cycle		-0.324	-0.433	-0.216
Rowland et al 2003 car		-0.303	-0.407	-0.200
Garvill et al 2003		-0.302	-0.403	-0.201
Matsukawa 2004		-0.294	-0.391	-0.196
Sargeant et al 2004c		-0.295	-0.393	-0.198
Sargeant et al 2004b		-0.296	-0.391	-0.201
Sargeant et al 2004a		-0.291	-0.385	-0.197
Fujii & Taniguchi 2005	-0.282	-0.373	-0.192	
Kurz et al 2005 treat3e	-0.284	-0.373	-0.194	
Kurz et al 2005 treat1e	-0.284	-0.372	-0.195	

After this point the estimate will not return to medium-small effect sizes (d=>0.3)	Kurz et al 2005 treat3w	-0.282	-0.369	-0.194	
	Kurz et al 2005 treat2w	-0.286	-0.373	-0.199	
	Kurz et al 2005 treat1w	-0.285	-0.370	-0.199	
	Kurz et al 2005 treat2e	-0.283	-0.367	-0.198	
	Holland et al 2006 treat2b	-0.302	-0.390	-0.214	
	Holland et al 2006 treat2a	-0.310	-0.398	-0.221	
	Holland et al 2006 treat1b	-0.319	-0.407	-0.230	
	Holland et al 2006 treat1a	-0.317	-0.405	-0.229	
	Bamberg 2006	-0.311	-0.397	-0.225	
	Goldstein et al 2008 St1	-0.307	-0.391	-0.224	
	Nolan et al 2008 treat2	-0.297	-0.380	-0.214	
	Schultz et al 2008 St3	-0.294	-0.375	-0.212	
	Nolan et al 2008 treat4	-0.283	-0.364	-0.203	
	Nolan et al 2008 treat1	-0.319	-0.408	-0.231	
	Goldstein et al 2008 St2	-0.316	-0.403	-0.229	
	Schultz et al 2008 St2	-0.316	-0.402	-0.230	
	Nolan et al 2008 treat3	-0.321	-0.406	-0.235	
	Eriksson et al 2008	-0.319	-0.403	-0.234	
	Wen et al 2008 parents_car	-0.310	-0.391	-0.229	
	Hemmingsson et al 2009 car	-0.306	-0.386	-0.226	
	Mair & Bergin-Seers 2010 treat2	-0.305	-0.384	-0.225	
	Mair & Bergin-Seers 2010 treat1	-0.303	-0.382	-0.224	
	Mair & Bergin-Seers 2010 treat3	-0.299	-0.377	-0.220	
	Sussman & Gifford 2011 treat2	-0.301	-0.379	-0.224	
	SMALL EFFECT SIZE	Goldstein et al 2011 St1 treat2	-0.298	-0.374	-0.223
Goldstein et al 2011 St1 treat1		-0.289	-0.364	-0.214	
Carrico & Riemer 2011 treat2		-0.293	-0.367	-0.218	
Alcott 2011 treat2		-0.274	-0.341	-0.208	
Sussman & Gifford 2011 treat1		-0.267	-0.333	-0.201	
Carrico & Riemer 2011 treat1		-0.265	-0.330	-0.200	
Alcott 2011 treat1		-0.225	-0.275	-0.175	
Bapuji et al 2012		-0.229	-0.279	-0.179	
Aittasalo et al 2012		-0.231	-0.281	-0.181	
Baca-Motes et al 2013 treat2		-0.225	-0.274	-0.176	
Baca-Motes et al 2013 treat1		-0.217	-0.265	-0.169	
Houde et al 2013		-0.211	-0.258	-0.164	
Handgraaf et al 2013 treat1		-0.213	-0.260	-0.166	
Ferraro & Price 2013 treat2		-0.187	-0.228	-0.146	
Fielding et al 2013 treat3		-0.185	-0.226	-0.144	
Kallbekken et al 2013 dryers treat1		-0.184	-0.224	-0.143	
Costa & Kahn 2013		-0.164	-0.201	-0.127	
Schwartz et al 2013		-0.156	-0.192	-0.121	
Handgraaf et al 2013 treat2		-0.159	-0.195	-0.123	
Fielding et al 2013 treat2		-0.159	-0.195	-0.124	
Ferraro & Price 2013 treat3		-0.143	-0.176	-0.110	
Schleich et al 2013		-0.139	-0.171	-0.107	
Kallbekken & Saelen 2013 treat2		-0.141	-0.173	-0.109	
Wansink & van Ittersumm 2013		-0.143	-0.175	-0.111	
Ferraro & Price 2013 treat1		-0.129	-0.159	-0.099	
Baca-Motes et al 2013 treat3		-0.127	-0.157	-0.098	
Kallbekken et al 2013 dryers treat2		-0.127	-0.156	-0.098	
		Goodman et al 2013 car	-0.094	-0.116	-0.071

Very Small Effect Size	Fielding et al 2013 treat1	-0.094	-0.116	-0.071
	Kallbekken & Saelen 2013 treat1	-0.094	-0.117	-0.071
	Reese et al 2014	-0.095	-0.118	-0.072
	Bohner & Schluter 2014 St1 treat1	-0.096	-0.119	-0.073
	Tornblad et al 2014	-0.095	-0.118	-0.073
	Delmas & Lessem 2014 treat1	-0.096	-0.119	-0.074
	Delmas & Lessem 2014 treat2	-0.096	-0.119	-0.073
	Ascensio et al 2014 treat2	-0.097	-0.120	-0.074
	Ascensio et al 2014 treat1	-0.096	-0.119	-0.073
	Bohner & Schluter 2014 St1 treat2	-0.096	-0.118	-0.073
	Jeong et al 2014	-0.092	-0.115	-0.070
	Goodhew et al 2015 St1 treat1	-0.092	-0.115	-0.070
	Seyranian et al 2015 treat2	-0.093	-0.115	-0.071
	Datta et al 2015 treat2	-0.090	-0.112	-0.068
	Schultz et al 2015 treat1	-0.090	-0.112	-0.068
	Schultz et al 2015 treat2	-0.092	-0.113	-0.070
	Goodhew et al 2015 St1 treat2	-0.091	-0.113	-0.070
	Terrier & Marfaing 2015 treat1	-0.093	-0.115	-0.071
	Dolan & Metcalfe 2015 St1	-0.097	-0.119	-0.075
	Datta et al 2015 treat1	-0.095	-0.116	-0.073
	Seyranian et al 2015 treat1	-0.095	-0.116	-0.074
	Goodhew et al 2015 St2	-0.096	-0.117	-0.074
	Terrier & Marfaing 2015 treat2	-0.098	-0.120	-0.077
	Datta et al 2015 treat3	-0.096	-0.117	-0.075
	Seyranian et al 2015 treat3	-0.096	-0.117	-0.075
	Thondhlana & Kua 2016	-0.097	-0.118	-0.076
	Lynham et al 2016	-0.097	-0.118	-0.076
	Schultz et al 2016 treat1	-0.097	-0.118	-0.076
	Carroll et al 2016	-0.097	-0.118	-0.076
	Kongback et al 2016	-0.097	-0.118	-0.076
	Schultz et al 2016 treat2	-0.098	-0.119	-0.077
	Friis et al 2017 treat1	-0.099	-0.120	-0.078
	Qi & Roe 2017 treat1	-0.099	-0.120	-0.078
	Allcott & Sweneey 2017	-0.094	-0.114	-0.074
	Clayton & Nesnidol 2017	-0.094	-0.114	-0.074
	Anderson et al 2017	-0.093	-0.113	-0.073
	Hsieh et al 2017 treat1	-0.093	-0.113	-0.073
	Hsieh et al 2017 treat2	-0.093	-0.113	-0.073
	Sparkman & Walton 2017 St 5 treat1	-0.093	-0.113	-0.073
	Friis et al 2017 treat2	-0.093	-0.113	-0.073
	Bamberg & Rees 2017 cycle	-0.093	-0.112	-0.073
	Qi & Roe 2017 treat2	-0.093	-0.112	-0.073
	Chen et al 2017 treat2	-0.092	-0.112	-0.073
	Christina et al 2017	-0.091	-0.111	-0.072
	Sparkman & Walton 2017 St 5 treat2	-0.092	-0.111	-0.073
Sparkman & Walton 2017 St 4 treat2	-0.093	-0.113	-0.074	
Chen et al 2017 treat1	-0.093	-0.112	-0.073	
Sparkman & Walton 2017 St 4 treat1	-0.092	-0.112	-0.073	
Kendel et al 2017	-0.092	-0.112	-0.073	
Friis et al 2017 treat3	-0.093	-0.113	-0.074	

Reviewers' comments:

Reviewer #1 (Remarks to the Author):

This paper will get significant attention and could have useful impact on research on greenhouse gas mitigation and on the thinking of behavioral scientists, and even policy makers and private actors working in this domain. However, its impact could be either positive or negative, very much depending on how it is framed. In response to reviewers' comments, the author(s) have added text in the discussion section to indicate that the weak effects observed in this review could indicate not so much that behavioural interventions have little effect, but rather that such interventions alone have little effect. But this point is not maintained throughout, and is easily lost. For example, the paragraph on p. 10 beginning "An important point to emphasize..." is a bit of a non sequitur. The same is true for the penultimate paragraph of the discussion, which also compares behavioural with other types of intervention rather than addressing the issue of possible positive interactions between different types of approaches.

The most important issue regarding the effectiveness of behavioural interventions is not whether they (in isolation) have an effect when isolated from other intervention types or whether they have greater or lesser effects than financial or regulatory interventions, also in isolation. The most important issue is whether, and under what conditions, behavioural interventions of particular types can make a substantial difference—either alone or when integrated with other types of interventions to increase adoption of actions with significant potential to reduce greenhouse gas emissions. It is, as mentioned in an added paragraph in the Discussion section, a question of interaction effects. This paper's findings that main effects (i.e., behavioural interventions holding everything else constant) are weak should point researchers to focus more on interaction effects. This is important because behavioural interventions, like financial and regulatory ones, typically fall short of the expectations derived from untested assumptions about behavioral plasticity. Interactions are where the greatest potential lies.

To direct researchers and practical intervenors to the most likely effective uses of behavioural interventions, the paper needs to make the point about interaction effects more salient throughout. The most important place to do this is in the title of the eventually published article. Titles frame the entire article and can strongly influence the perceptions of researchers, policy makers, and private-sector intervenors who will hear about the article from secondary sources. The current title suggests that no matter how they are employed and in whatever contexts, behavioural interventions have little impact. That is not the proper conclusion to draw from the data. I suggest using a modified title for the article. The title (before the colon) now reads:

“Low impact of behavioural interventions to promote climate change mitigation”

But the findings support a different title before the colon, perhaps something like this:

“Behavioural interventions acting alone have low impact on household actions to promote climate change mitigation”

The title, the abstract, and elsewhere, the article should make clearer that the review focuses on individual and household consumption as the targets of interventions and that behavioural interventions are examined holding all other factors constant (i.e., looking only at main effects). With that clarification, the paper will probably still have an important influence on the thinking of researchers, policy makers, and private-sector actors interested in promoting mitigation of climate change, but it will not lead them to draw the almost surely mistaken conclusion that behavioural interventions have no place in the armamentarium for achieving emissions reduction targets. It is probably in conjunction with other types of intervention that behavioural interventions can have the greatest impact. For example, as has been argued in research going back to the 1980s, the effect of prime psychological effect of financial incentives particularly for high-impact behaviors such as investments in energy-efficient equipment, is to get consumers to pay attention to an action. After that, information, marketing, nudges, etc., can turn attention into action. The IPCC report may have missed such important subtleties about interaction effects; if so, this paper can supply an important corrective, and will be highly cited and influential as a result.

Here are some suggestions for edits after the title to make what I see as the main point clearer.

Abstract: Second sentence: “Results show that these interventions, holding other factors constant,*...*” [italics indicate proposed additions]. The last two sentences might be revised to end with “...or their effectiveness in combination with other strategies. These possibilities should be subject to rigorous evaluation.”

Introduction: Last paragraph, research question #1: “Are behavioural interventions on their own...” or “Are behavioural interventions effective, holding other conditions constant...”. The same change should be made in the boldface reiteration of question #1 in the Results section.

Discussion: The comparison of the behavioral plasticity estimates between this study and Dietz et al. (2009) is an important one. But there are some errors in the comparison. It is true that Dietz et al. based their estimates on less stringent research designs and also that those estimates were based on best cases (though not scenarios, but rather effects of actual programs). Contrary to this paper’s claim, though, the Dietz et al. estimates were based on observed behavior, not self-reported intentions or attitudes.

At p. 10, the first sentence of the first whole paragraph goes beyond the data and will likely lead to mischaracterizations, even though the next sentence adds the critical clarifier, “in isolation”. I suggest editing the first sentence as follows: “Thus, the available field experiments do not give rise to hopeful predictions about relying on behavioural interventions alone to tackle climate change.”

Some other thoughts:

- The category “Energy consumption” probably should be “energy consumption in the home”, as transportation also entails energy consumption by individuals and households.
- Can POB be reported for food waste, animal products consumption, and recycling?
- The recommendations for examining the interactive effects of multiple behavioural and other intervention types seem to be at cross-purposes with the call for randomized controlled trials that allow for drawing conclusions about both main effects and interactions. Although both these recommendations are sensible, the need to develop scientifically supported, actionable findings quickly to inform interventions does not realistically allow for the time and effort needed to separate main effects from interaction with the highest scientific confidence. What is needed is the identification of highly promising approaches to achieving high behavioral plasticity for actions with high technical potential. This is addressed to some degree in the paragraph headed “Third,...”
- I respectfully disagree with the inference that the relative strength of choice architecture manipulations means that people are not strongly motivated to mitigate climate change. An alternative explanation that I find more credible is that most people are unwilling to invest the time and effort needed to identify effective actions for limiting climate change, and so are amenable to influence by cognitive short-cuts. Regarding informational interventions, this observation about time and effort suggests a conclusion that has been drawn elsewhere (e.g., refs. 33 and 35): that informational interventions in which information comes from trustworthy sources and is easily available at the point of decision are much more effective than other interventions.

Reviewer #2 (Remarks to the Author):

The authors did elaborate further on the recommendations for future study, which I think are quite helpful. The only concern I have left toward the paper is its high-level contribution – the generalizability of these future recommendations. There are two primary aspects:

- First, I think the intervention effectiveness and the sustainability of behavioral intervention (i.e., behavioral sustainability) are two different things. It’s a key concept of behavior intervention, yet not addressed in the paper. For instance, the paper suggests targeting behaviors that have high

technical potential such as eating fewer animal products and reduce driving and flying. However, will this stay true when taking into account the behavioral sustainability issue? If not, then the argument/recommendation becomes not strong enough. Also, what would the authors advise future researchers considering this big intervention challenge (i.e., sustainability)? Clarifying these in the recommendations for future research can make the paper stronger (e.g., first and fourth recommendations).

- Second, it's important that the authors pointed out the necessity to tease out the multiple effect(s) in an intervention (e.g., second recommendation). This is also suggested by prior behavior intervention research. Having said that, I am wondering if the authors look into which *combinations* of intervention components are more effective when being used together in a single intervention? As the paper also implicitly mentions, information-based strategy could be helpful during a certain timing/context. If this is the case, what about an intervention that starts with information-based strategy, and then shifts to social comparison? This is similar to the concept of "adaptive intervention", which becomes popular in recent years in the field of behavior intervention research. I wonder if more elaboration can be made regarding "adaptive behavior intervention" (or even tailored/personalized intervention) [1] in the Discussion section (recommendations for future research). I think it's important to further clarify this, as financial incentives and social comparison are shown to have short-lived effect based on some prior work - incorporating multiple intervention components/strategies in an intervention could then become the solution.

References

[1] Miller, C. K. (2019). Adaptive Intervention Designs to Promote Behavioral Change in Adults: What Is the Evidence?. *Current diabetes reports*, 19(2), 7.

Response to Reviewers NCOMMS-18-33120C

We sincerely appreciate all comments received, and we thank both reviewers for their engagement in our work. We feel the revised manuscript is richer and more adequately situated in the relevant literature than its predecessor. We do hope you like what we have done and consider the paper worthy of publication in Nature Communication.

Reviewer 1

Comment 1: This paper will get significant attention and could have useful impact on research on greenhouse gas mitigation and on the thinking of behavioral scientists, and even policy makers and private actors working in this domain. However, its impact could be either positive or negative, very much depending on how it is framed. In response to reviewers' comments, the author(s) have added text in the discussion section to indicate that the weak effects observed in this review could indicate not so much that behavioural interventions have little effect, but rather that such interventions alone have little effect. But this point is not maintained throughout, and is easily lost. For example, the paragraph on p. 10 beginning "An important point to emphasize..." is a bit of a non sequitur. The same is true for the penultimate paragraph of the discussion, which also compares behavioural with other types of intervention rather than addressing the issue of possible positive interactions between different types of approaches.

The most important issue regarding the effectiveness of behavioural interventions is not whether they (in isolation) have an effect when isolated from other intervention types or whether they have greater or lesser effects than financial or regulatory interventions, also in isolation. The most important issue is whether, and under what conditions, behavioural interventions of particular types can make a substantial difference—either alone or when integrated with other types of interventions to increase adoption of actions with significant potential to reduce greenhouse gas emissions. It is, as mentioned in an added paragraph in the Discussion section, a question of interaction effects. This paper's findings that main effects (i.e., behavioural interventions holding everything else constant) are weak should point researchers to focus more on interaction effects. This is important because behavioural interventions, like financial and regulatory ones, typically fall short of the expectations derived from untested assumptions about behavioral plasticity. Interactions are where the greatest potential lies.

Reply to Comment 1: We would like to thank the Reviewer for these important recommendations. We have removed the sentence "An important point to emphasize..." from the Discussion, and replaced it with the arguments presented by the Reviewer (page 10, last paragraph):

"Our findings, in combination with this seminal evidence, suggest that researchers should focus more on interaction effects. An important issue to be further addressed is whether, and under what conditions, specific behavioural interventions can make a substantial difference when integrated with other types of interventions to increase adoption of actions with a high potential to reduce carbon emissions. This is an important issue because behavioural interventions, like financial incentives and regulation, typically fall short of expectations derived from untested assumptions about behavioural plasticity. Interactions may be where the greatest potential lies."

We have also added in the penultimate paragraph of the discussion the need to examine interaction effects (page 13): *"Our results should not be used to argue for a disinvestment in behavioural interventions because this work did not compare behavioural interventions with alternative approaches, nor did it examine their interactive effect with other strategies"*.

Moreover, we have incorporated all additional suggestions made by the Reviewer to improve the consistency of the message across the manuscript, including changes to the title, abstract, introduction, and further changes to the discussion. All changes done according to the recommendations of the Reviewer are detailed in the comments below.

*Comment 2: To direct researchers and practical intervenors to the most likely effective uses of behavioural interventions, the paper needs to make the point about interaction effects more salient throughout. The most important place to do this is in the title of the eventually published article. Titles frame the entire article and can strongly influence the perceptions of researchers, policy makers, and private-sector intervenors who will hear about the article from secondary sources. The current title suggests that no matter how they are employed and in whatever contexts, behavioural interventions have little impact. That is not the proper conclusion to draw from the data. I suggest using a modified title for the article. The title (before the colon) now reads: “Low impact of behavioural interventions to promote climate change mitigation”
But the findings support a different title before the colon, perhaps something like this:
“Behavioural interventions acting alone have low impact on household actions to promote climate change mitigation”*

Reply to Comment 2: Changed as requested (Title with track changes).

*Comment 3: The title, the abstract, and elsewhere, the article should make clearer that the review focuses on individual and household consumption as the targets of interventions and that behavioural interventions are examined holding all other factors constant (i.e., looking only at main effects). With that clarification, the paper will probably still have an important influence on the thinking of researchers, policy makers, and private-sector actors interested in promoting mitigation of climate change, but it will not lead them to draw the almost surely mistaken conclusion that behavioural interventions have no place in the armamentarium for achieving emissions reduction targets. It is probably in conjunction with other types of intervention that behavioural interventions can have the greatest impact. For example, as has been argued in research going back to the 1980s, the effect of prime psychological effect of financial incentives particularly for high-impact behaviors such as investments in energy-efficient equipment, is to get consumers to pay attention to an action. After that, information, marketing, nudges, etc., can turn attention into action. The IPCC report may have missed such important subtleties about interaction effects; if so, this paper can supply an important corrective, and will be highly cited and influential as a result. Here are some suggestions for edits after the title to make what I see as the main point clearer.
Abstract: Second sentence: “Results show that these interventions, holding other factors constant,...” [italics indicate proposed additions]. The last two sentences might be revised to end with “...or their effectiveness in combination with other strategies. These possibilities should be subject to rigorous evaluation.”*

Reply to Comment 3: Changed as requested (Abstract with track changes).

Comment 4: Introduction: Last paragraph, research question #1: “Are behavioural interventions on their own...” or “Are behavioural interventions effective, holding other conditions constant...”. The same change should be made in the boldface reiteration of question #1 in the Results section.

Reply to Comment 4: Changed as requested (pages 3 and 4 with track changes). Moreover, we added a new mention in the Introduction (page 2, 2nd paragraph, 2nd sentence) as a way to increase the salience that we are looking at the impact of behavioural interventions holding all other factors constant.

Comment 5: Discussion: The comparison of the behavioral plasticity estimates between this study and Dietz et al. (2009) is an important one. But there are some errors in the comparison. It is true that Dietz et al. based their estimates on less stringent research designs and also that those estimates

were based on best cases (though not scenarios, but rather effects of actual programs). Contrary to this paper's claim, though, the Dietz et al. estimates were based on observed behavior, not self-reported intentions or attitudes.

Reply to Comment 5: As recommended, we removed this part of the sentence (page 10 with track changes).

Comment 6: *At p. 10, the first sentence of the first whole paragraph goes beyond the data and will likely lead to mischaracterizations, even though the next sentence adds the critical clarifier, "in isolation". I suggest editing the first sentence as follows: "Thus, the available field experiments do not give rise to hopeful predictions about relying on behavioural interventions alone to tackle climate change."*

Reply to Comment 6: Changed as requested (page 10 with track changes).

Comment 7: *Some other thoughts: The category "Energy consumption" probably should be "energy consumption in the home", as transportation also entails energy consumption by individuals and households.*

Reply to Comment 7: Changed as requested (page 2 with track changes).

Comment 8: *Can POB be reported for food waste, animal products consumption, and recycling?*

Reply to Comment 8: Added as requested (page 7 with track changes)

Comment 9: *The recommendations for examining the interactive effects of multiple behavioural and other intervention types seem to be at cross-purposes with the call for randomized controlled trials that allow for drawing conclusions about both main effects and interactions. Although both these recommendations are sensible, the need to develop scientifically supported, actionable findings quickly to inform interventions does not realistically allow for the time and effort needed to separate main effects from interaction with the highest scientific confidence. What is needed is the identification of highly promising approaches to achieving high behavioral plasticity for actions with high technical potential. This is addressed to some degree in the paragraph headed "Third,..."*

Reply to Comment 9: To clarify this point, we distinguish between cases where randomisation is possible (versus when it is not), and how the analysis of main versus interaction effects may be conducted in each case. Please note the added text using track changes in the second and third recommendation sections (page 11).

In the second recommendation section (where we discuss the need for research designs that establish causality and alternative solutions for when this is not possible), we added the following segment:

"These more pragmatic, quasi-experimental methods may be faster, cheaper and logistically more feasible than randomized controlled trials, and could provide valuable information (39-40). Results should, nevertheless, be presented with due caution."

In the third recommendation section (where the need to disentangle main versus interaction effects is discussed), we added the following segment:

“As this work attests, ample research has been conducted in the field randomly assigning individuals or households to different treatments. When such randomisation is possible, factorial designs are an adequate tool to test main versus interaction effects (41). Testing these main versus interaction effects requires only minor changes to the study design and potentially an increase in sample size, mostly for discrete dependent variables (42). However, when individual or household-level randomisation is not possible, cluster randomisation or quasi-experimental designs can still be conducted using a factorial design (43). In more restrictive research contexts, researchers may use counterbalanced designs (44-45), in which different treatments are implemented stepwise. These treatments may include single stimulus and bundles of stimuli, introduced and/or removed sequentially (46). Although the estimation of these effects may be more challenging and less precise than with a true experimental design, these quasi-experimental alternatives may still shed some light about the differential effects of single versus combined stimuli (47)”.

39. Kontopantelis, E., Doran, T., Springate, D. A., Buchan, I., & Reeves, D. (2015). Regression based quasi-experimental approach when randomisation is not an option: interrupted time series analysis. *BMJ*, 350, h2750.
40. Calonico, Sebastian, Matias D. Cattaneo, Max H. Farrell, and Rocio Titiunik. (2018). "Regression discontinuity designs using covariates." *Review of Economics and Statistics*.
41. Montgomery, A. A., Peters, T. J., & Little, P. (2003). Design, analysis and presentation of factorial randomised controlled trials. *BMC medical research methodology*, 3(1), 26.
42. Faul, F., Erdfelder, E., Lang, A. G., & Buchner, A. (2007). G* Power 3: A flexible statistical power analysis program for the social, behavioral, and biomedical sciences. *Behavior research methods*, 39(2), 175-191.
43. Bol, L., Hacker, D. J., Walck, C. C., & Nunnery, J. A. (2012). The effects of individual or group guidelines on the calibration accuracy and achievement of high school biology students. *Contemporary Educational Psychology*, 37(4), 280-287.
44. Siegel, J. T., Alvaro, E. M., Crano, W. D., Lac, A., Ting, S., & Jones, S. P. (2008). A quasi-experimental investigation of message appeal variations on organ donor registration rates. *Health Psychology*, 27(2), 170.
45. Sarkies, M. N., Skinner, E. H., Bowles, K. A., Morris, M. E., Williams, C., O'Brien, L., & White, J. (2019). A novel counterbalanced implementation study design: methodological description and application to implementation research. *Implementation Science*, 14(1), 45-45.
46. Hemming, K., Haines, T. P., Chilton, P. J., Girling, A. J., & Lilford, R. J. (2015). The stepped wedge cluster randomised trial: rationale, design, analysis, and reporting. *BMJ*, 350, h391.
47. Oswald, W. D., Gunzelmann, T., Rupprecht, R., & Hagen, B. Differential effects of single versus combined cognitive and physical training with older adults: the SimA study in a 5-year perspective. *European Journal of Ageing* 3(4), 179-192 (2006).

***Comment 10:** I respectfully disagree with the inference that the relative strength of choice architecture manipulations means that people are not strongly motivated to mitigate climate change. An alternative explanation that I find more credible is that most people are unwilling to invest the time and effort needed to identify effective actions for limiting climate change, and so are amenable to influence by cognitive short-cuts. Regarding informational interventions, this observation about time and effort suggests a conclusion that has been drawn elsewhere (e.g., refs. 33 and 35): that informational interventions in which information comes from trustworthy sources and is easily available at the point of decision are much more effective than other interventions.*

***Reply to Comment 10:** We removed the sentence related to the motivation to mitigate climate change. Furthermore, we have expanded our discussion on how information strategies may yield greater impact when combined with financial incentives, delivered by trustworthy sources and easily available at the point of decision (end of fourth recommendation, page 12).*

Reviewer 2

Comment 1: The authors did elaborate further on the recommendations for future study, which I think are quite helpful. The only concern I have left toward the paper is its high-level contribution – the generalizability of these future recommendations. There are two primary aspects:

First, I think the intervention effectiveness and the sustainability of behavioral intervention (i.e., behavioral sustainability) are two different things. It's a key concept of behavior intervention, yet not addressed in the paper. For instance, the paper suggests targeting behaviors that have high technical potential such as eating fewer animal products and reduce driving and flying. However, will this stay true when taking into account the behavioral sustainability issue? If not, then the argument/recommendation becomes not strong enough. Also, what would the authors advise future researchers considering this big intervention challenge (i.e., sustainability)? Clarifying these in the recommendations for future research can make the paper stronger (e.g., first and fourth recommendations).

Reply to Comment 1: This is an excellent point; we incorporated it in the revised manuscript. The reviewer correctly points out that intervention effectiveness and behavioural sustainability are different things. The former relates to how much behavioural change an intervention can achieve, while the latter relates to how durable and sustained these changes are, both during and after the implementation period. We have incorporated this important discussion in a new fifth recommendation (please refer to comment below), as well as added an additional mention to this point in our final summary paragraph (page 13).

*Comment 2: Second, it's important that the authors pointed out the necessity to tease out the multiple effect(s) in an intervention (e.g., second recommendation). This is also suggested by prior behavior intervention research. Having said that, I am wondering if the authors look into which *combinations* of intervention components are more effective when being used together in a single intervention? As the paper also implicitly mentions, information-based strategy could be helpful during a certain timing/context. If this is the case, what about an intervention that starts with information-based strategy, and then shifts to social comparison? This is similar to the concept of "adaptive intervention", which becomes popular in recent years in the field of behavior intervention research. I wonder if more elaboration can be made regarding "adaptive behavior intervention" (or even tailored/personalized intervention) [1] in the Discussion section (recommendations for future research). I think it's important to further clarify this, as financial incentives and social comparison are shown to have short-lived effect based on some prior work - incorporating multiple intervention components/strategies in an intervention could then become the solution.*

[1] Miller, C. K. (2019). Adaptive Intervention Designs to Promote Behavioral Change in Adults: What Is the Evidence? Current diabetes reports, 19(2), 7.

Reply to Reviewer: These are also excellent points that we now elaborate further. In response to the Reviewer's question, we did not look into which combinations of stimuli were more effective because the included papers reported no clear, repeated patterns in the combination of stimuli that would allow us to conduct a rigorous analysis. This reinforces our third recommendation on stimuli bundles, further underscoring the importance of properly measuring and reporting single stimulus versus stimuli combination. Moreover, regarding the point about adaptive interventions, we have partially incorporated this comment in the paper. The perspective of tailoring or personalising interventions is less relevant to our goal, which is to support large-scale, evidence-based policy action. The principles of personalised medicine rely on individualised care, tailored to patient characteristics that are typically examined in small-scale feasibility studies. In contrast, most policy-making relies on identifying the effects that are likely to be the best course of action for the largest number of people. However, the notion that interventions may need to be dynamic and adaptive to increase effectiveness, is a very important one. This part of the argument has been incorporated in our response.

We have merged the discussion of these interrelated points from Comment 1 and 2 in our new fifth recommendation. Please find this additional text below (also in track changes, page 12).

“Fifth, more evidence is needed to identify which interventions can stand the test of time. Regarding which interventions produce long-lasting behavioural changes after the intervention is concluded, the available evidence was limited, and the included papers with follow-up data reported an average null effect. Our current recommendations for more investment in interventions with the highest behavioural plasticity potential (social comparison and nudges) are based on effect sizes while the intervention is in place. Future research may adjust these recommendations if evidence establishes that interventions with a more modest short-term impact maintain their effect in follow-up periods, whereas more impactful interventions may see their effect fade sharply after the interventions is concluded. Habit is one of the persistence pathways proposed to explain sustained effects (54). Yet, holding all factors constant, the intervention duration did not have an effect. This is puzzling, to some extent, because most consequential household actions are frequent and recurrent in stable settings (e.g., turning on the air conditioning at home, driving to work, selecting meal size and type). Thus, the expectation could be that, the longer the intervention, the more likely it is to create a positive habit in such recurring contexts. Our work included very heterogeneous research contexts and behaviours, possibly with different timescales required for change to happen, which may have masked the effect of intervention duration. Other persistence pathways include changing how people think (e.g., beliefs), changing future costs of mitigation behaviours (e.g., energy-efficient appliances), and external reinforcement (either social or financial). There is no conclusive answer on how to succeed in these pathways, or which one is more effective. Nonetheless, some isolated examples show that evidence for changing future costs is not compelling (55) but social comparison (external reinforcement), and nudges (automatic habit reshape) have reported striking persistent effects (56-57) even with a one-shot intervention (5). These examples suggest social comparison and nudges may be good options to transform transient behaviour changes into sustained behavioural patterns. Nonetheless, future research should establish more precisely when and why intervention effects persist.

Given that most behavioural interventions seem to be short-lived, incorporating multiple, sequential intervention components could be a promising solution. Relevant to this debate are the concepts of complex and adaptive interventions (58-59)—interventions composed of various interconnecting parts that emphasize the timing and sequence for different treatment(s)—as a way to adjust treatment response to circumvent saturation effects during implementation, and to promote sustained effects after implementation is concluded (58). Applied to climate change research, an adaptive complex intervention could start with a social comparison message, after which it could shift to (or add) an information-based strategy, and conclude with a change to (or an extra added) environmental appeal to save the planet. However, a challenge for these sequential designs is what theoretical basis justifies the order selected and tipping point for adjustment (59). In line with our previous arguments, a sequence of interventions should probably start with motivating, eye-catching strategies (e.g., social comparison, financial incentives), to subsequently move to (or add-on) more information-based interventions, once attention for climate change has been grabbed. The analysis of temporal dynamics in such cumulative designs could contribute both to providing evidence on whether interventions should be adaptive over time, along with contributing to our understanding of how stimuli bundle works (3rd recommendation).”

5. Ferraro, P. J., Miranda, J. J., & Price, M. K. (2011). The persistence of treatment effects with norm-based policy instruments: evidence from a randomized environmental policy experiment. *American Economic Review*, 101(3), 318-22.

54. Frey, E., & Rogers, T. (2014). Persistence: How treatment effects persist after interventions stop. *Policy Insights from the Behavioral and Brain Sciences*, 1(1), 172-179.

55. Lucas, D., Fuchs, A. & Gertler, P. (2014). Cash for Coolers: Evaluating a large-scale appliance replacement program in Mexico. *American Economic Journal: Economic Policy*, 6(4), 207-238.

56. Allcott, H., & Rogers, T. (2014). The short-run and long-run effects of behavioral interventions: Experimental evidence from energy conservation. *American Economic Review*, 104(10), 3003-37.

57. Kurz, V. (2018). Nudging to reduce meat consumption: Immediate and persistent effects of an intervention at a university restaurant. *Journal Environmental Economics & Management*, 90, 317-341.

58. Craig, P., Dieppe, P., Macintyre, S., Michie, S., Nazareth, I., & Petticrew, M. (2008). Developing and evaluating complex interventions: the new Medical Research Council guidance. *BMJ*, 337, a1655.

59. Miller, C. K. (2019). Adaptive Intervention Designs to Promote Behavioral Change in Adults: What Is the Evidence? *Current diabetes reports*, 19(2), 7.

Reviewers' comments:

Reviewer #1 (Remarks to the Author):

In my view, the revised paper has been significantly improved by the changes the author(s) made in response to my review (I was Reviewer #1). However, reading the revised version has alerted me to some issues with the paper that were not obvious to me before. I elaborate below, and then offer some ideas about ways to address these issues.

The most fundamental issue, I think, lies in the framing of the paper: Asking the extent to which behavioral interventions acting alone can change the target behaviors. There is a long-standing literature indicating that the best ways to use behavioral interventions to change household actions to mitigate climate change is not alone, but in combination with financial and other kinds of interventions. This is the view expressed by Dietz et al., 2009 and in many prominent papers in the literature going back at least to the 1980s. Thus, the findings in this paper should be no surprise to readers familiar with that literature. What is new in this paper is the careful methodological analysis quantifying the small effects of behavioral interventions on their own, even when they are statistically significant. This may well be news to many researchers, particularly psychologists, who continue to argue that statistically significant effects of certain behavioral interventions on certain climate-relevant household behaviors mean that behavioral interventions focused on households can on their own can make a significant contribution to limiting climate change. This view may be implicit in the sections of the recent IPCC report focused on behavioral research, so the findings in this paper may in fact be significant in getting behavioral researchers to pay more attention to the need to integrate behavioral with other types of intervention to have real-world impact.

A second issue is that the paper does not distinguish among the contexts in which behaviors occur, assuming implicitly that for each type of behavior there is a single best estimate of the amount of change that can be achieved by a particular type of intervention. The discussion of interaction effects, though appropriate as far as it goes, presumes that there is also a single best estimate of each interaction effect that could be made by analyzing appropriately collected data. A contrasting view, which is present in the Dietz et al. paper and is in my view supported by the available evidence from multiple methods, is that behavioral changes are mediated by the presence or absence of various barriers to change, which may be physical, institutional, economic, cognitive, attitudinal, etc. In this view, the greatest behavioral plasticity is achieved by intervention packages that adequately address the specific barriers to change that apply to target populations and contexts. I think this is why Dietz et al. defined behavioral plasticity as the greatest level of behavior change that had been demonstrated in real-world applications in the USA, using any combination of intervention types, rather than via behavioral interventions alone. Admittedly, the barriers view is not readily tested by controlled experimentation. In my view, though, the presence of different barriers depending on the behavior and the context poses a fundamental limitation to the approach of treating controlled experimentation as the gold-standard methodology. What can be learned from this method can be valuable, but its limitations for drawing general conclusions need to be recognized because behavioral change is sensitive to context-specific barriers.

Another issue that emerges more clearly to me in the revision is the paper's failure to distinguish frequently repeated behaviors, whose effects must be aggregated through multiple repetitions (e.g., travel to work, adjusting room temperatures, changing diet), from behaviors whose effects are almost fully achieved by a single action (e.g., upgrading the energy efficiency of building shells, adopting more energy-efficient home appliances and vehicles). The paper recognizes that nearly all the experimental studies meeting its selection criteria focus on frequently repeated behaviors (the only four studies that do not do so focus on appliance choice), but it does not fully consider the implications of that fact. One implication is that the establishment of habit (the issue of whether the behaviors "stand the test of time"), which has an important place in this paper, is important only for frequently repeated behaviors. Another concern arises from the fact that the

most consequential behaviors (at least as analyzed on a time scale of years to decades, as is done in the Dietz et al., paper) are not, as the paper asserts, frequently occurring, but are infrequent, one-time actions that have lasting effects without changing daily practices. Of the behaviors studied by Dietz et al., home weatherization and acquisition of fuel-efficient vehicles, both one-time actions, have a far greater Potential Emissions Reduction (i.e., what would be achieved if these behaviors were universally adopted) than changes in appliance purchases. Other one-time actions that have become more widespread since the publication of that paper, such as adoption of photovoltaic (PV) energy systems and electric vehicles that can run on PV electricity, arguably also have much greater potential importance than appliance purchases. Thus, the fact that this paper does not address high-impact one-time actions involving acquiring lower-carbon household technologies or upgrading existing ones seriously compromises the conclusions that readers may draw from the paper.

As Stern and Gardner noted as far back as their review in 1980, psychologists have tended to focus on frequently repeated behaviors, neglecting one-time behaviors that have major practical potential. This review suggests that this gap in the literature is still important. It may be understandable because it is difficult to do research on these behaviors that meets the methodological criteria of this review. However, a focus on behaviors with high Potential Emissions Reduction is critical for behavioral research to make its greatest contribution. Because the experimental literature has largely ignored the highest-potential actions, a review of that literature, such as this one, necessarily also ignores those actions, so its conclusions need to be qualified carefully before making general claims about the potential impact of behavioral interventions. (The 1980 reference is to Stern, P.C., and Gardner, G.T. Psychological research and energy policy. *American Psychologist*, 36(4), 329.)

The above issues are important for considering the implications and potential importance of the findings reported here. To me, these findings indicate that behavioral researchers who focus their attention on frequently repeated behaviors or who seek universal generalizations about the effects of particular intervention strategies on particular types of behavior are barking up the wrong tree. This was the implicit position taken in reviews such as that of Wolske and Stern (2018) and other past work that has drawn on multiple research methods to offer practical advice to those seeking to reduce the climate footprint of actions taken in and by households.

I see two ways the journal editors and the paper's author(s) might usefully proceed from here. One is for the authors to revise the paper again to take these issues explicitly into account. This would lead to the paper making weaker claims about the practical importance of the findings, but possibly stronger claims about the inadequacy of analyses, such as may be implicit in recent IPCC efforts, to the extent that they presume that behavioral interventions targeted at households can alone yield meaningful reduction on carbon emissions globally or in high-income countries. Thus, I think the data analyzed in this paper imply criticism of certain approaches to research on behavioral interventions, but do not imply that almost all such interventions will have little or no effect. Also, as noted above, the paper offers almost no evidence on the effects of behavioral interventions on one-time household actions, which offer the greatest potential impact on household climate footprints.

The authors may not want to take this direction. The alternative is to publish the paper more or less as is, but to commission a commentary that would raise points such as the above. I would prefer the first option, because more people are likely to read a review paper than a commentary on it and also because the story is best told all in one place. But the paper as is does not tell the whole story.

Reviewer #2 (Remarks to the Author):

I think the authors have addressed my comments well in this revision - particularly the issue regarding behavioral sustainability. Two minor points in need of clarification for the fifth recommendation:

1. Wondering if the authors can clarify their speculation regarding the statement that "the intervention duration did not have an effect". Wondering if there is any prior work that could help support this claim (i.e., "Thus, the expectation could be that, the longer the intervention, the more likely it is to create a positive habit in such recurring contexts.")?
2. It will be helpful if the authors can further clarify whether they think the "one-shot intervention" is one of the reasons that makes "social comparison and nudges" have persistent effects. In other words, would the effect become not persistent for prolonged, repeated or sequential interventions? The way it is written now could make the readers think these two strategies are relatively more powerful, which may not be the message authors would like to convey.

Response to Reviewers

Reviewer #1

Introduction to the response to Reviewer 1

The concerns expressed by the Reviewer in all rounds of comments are valid and noteworthy. It is legitimate to consider whether showing the small average effect sizes of behavioural interventions could be used to argue for a disinvestment in this type of approach. Following recommendations from Reviewer 1 in the past rounds, we introduced changes to the Title, Abstract, Introduction, Results and Discussion to caution against an oversimplified interpretation of our results. Our goal is not to diminish past research in any way, and particularly it is not to discredit Dietz et al. 2009, which we consider to be a seminal piece of research. We fully agree with Dietz et al. 2009 that household actions could help reduce carbon emissions. Our question is to what extent have behavioural interventions been able to produce changes in household actions. If behaviour change cannot be achieved, this is a moot point. We are exclusively focused on behavioural plasticity. Dietz et al. 2009 say that, with specific combinations of interventions and under particular circumstances, behavioural interventions could achieve significant behaviour change. We say that the bulk of research testing behavioural interventions in a rigorous way, over the past 40 years, is very far from reporting substantive and lasting behaviour changes. We are focused on establishing average main effects from causal field evidence, whereas the Reviewer is focused on emphasising the best combination effects, mostly recurring to other research designs. These views are not mutually exclusive.

In my view, the revised paper has been significantly improved by the changes the author(s) made in response to my review (I was Reviewer #1). However, reading the revised version has alerted me to some issues with the paper that were not obvious to me before. I elaborate below, and then offer some ideas about ways to address these issues. The most fundamental issue, I think, lies in the framing of the paper: Asking the extent to which behavioral interventions acting alone can change the target behaviors. There is a long-standing literature indicating that the best ways to use behavioral interventions to change household actions to mitigate climate change is not alone, but in combination with financial and other kinds of interventions. This is the view expressed by Dietz et al., 2009 and in many prominent papers in the literature going back at least to the 1980s. Thus, the findings in this paper should be no surprise to readers familiar with that literature.

Please note our discussion and clarifications about this concern in previous rounds of comments (Round 1: page 4 paragraphs 3, 5-7; page 9 paragraph 1; Round 2 page 2; Round 3 pages 1-4). The Reviewer seems to claim that only best-case estimates, as presented in Dietz et al. 2009, should be taken as an indication of how effective behavioural interventions are. We respectfully disagree. Average effect sizes represent expected impacts averaging out all covariates, and these average effects are a fundamental benchmark for intervention and policy making. We agree with the Reviewer that there may be particular contextual facilitators and combinations of interventions that are more effective than others. But this does not preclude the relevance and importance of average effect sizes, and our results should be reckoned with. Moreover, our findings should be interpreted as a meaningful contribution because the

distinction between behavioural interventions alone versus combinations of interventions was not systematically tested in this seminal research dating back from the 1980s.

What is new in this paper is the careful methodological analysis quantifying the small effects of behavioral interventions on their own, even when they are statistically significant. This may well be news to many researchers, particularly psychologists, who continue to argue that statistically significant effects of certain behavioral interventions on certain climate-relevant household behaviors mean that behavioral interventions focused on households can on their own can make a significant contribution to limiting climate change. This view may be implicit in the sections of the recent IPCC report focused on behavioral research, so the findings in this paper may in fact be significant in getting behavioral researchers to pay more attention to the need to integrate behavioral with other types of intervention to have real-world impact.

We appreciate and agree with the Reviewer that we conducted a careful methodological analysis quantifying average effect sizes of behavioural interventions holding all other factors constant. We hope our work will drive future research to pay more attention to how the impact of behavioural interventions can be improved, which may include more investment in the strategies shown more effective (e.g., nudges, social comparison), as well as more attention to testing which bundles of stimuli should co-occur in interventions to have a real-world impact. But, again, this does not preclude the relevance and importance of average effect sizes, as reference standards that future research should keep in mind as a benchmark for improvement.

A second issue is that the paper does not distinguish among the contexts in which behaviors occur, assuming implicitly that for each type of behavior there is a single best estimate of the amount of change that can be achieved by a particular type of intervention. The discussion of interaction effects, though appropriate as far as it goes, presumes that there is also a single best estimate of each interaction effect that could be made by analyzing appropriately collected data.

Please note our discussion and clarifications about this concern in previous rounds of comments (Round 1: page 6 paragraphs 4-6; similar point in response to Reviewer 3 page 16 paragraph 2). We do not claim to identify the single best estimates, but the *average* estimates of the amount of change that has been achieved to date. This means we estimated the average impacts that can be expected when using a particular type of intervention (e.g., information, appeals, nudges etc.). The discussion about interaction effects also presumes an average – not the best - expected impact from some combination of stimuli.

A contrasting view, which is present in the Dietz et al. paper and is in my view supported by the available evidence from multiple methods, is that behavioral changes are mediated by the presence or absence of various barriers to change, which may be physical, institutional, economic, cognitive, attitudinal, etc. In this view, the greatest behavioral plasticity is achieved by intervention packages that adequately address the specific barriers to change that apply to target populations and contexts. I think this is why Dietz et al. defined behavioral plasticity as the greatest level of behavior change that had been demonstrated in real-world applications in the USA, using any combination of

intervention types, rather than via behavioral interventions alone. Admittedly, the barriers view is not readily tested by controlled experimentation. In my view, though, the presence of different barriers depending on the behavior and the context poses a fundamental limitation to the approach of treating controlled experimentation as the gold-standard methodology. What can be learned from this method can be valuable, but its limitations for drawing general conclusions need to be recognized because behavioral change is sensitive to context-specific barriers.

The Reviewer appears to hold the view that changing mitigation behaviours in households is too complex and multi-layered to render an experimental analysis meaningful. We respectfully disagree. We understand the Reviewer's point and we do not dismiss the potential relevance of context. However, the Reviewer seems to place a greater emphasis on contextual variables than on the effectiveness of the intervention per se. It is unclear which grounds exist for this argument. Some studies show that behaviour may change depending on context, but this does not mean the context is more relevant than the main effect of the intervention, and the plasticity of the behaviour under analysis.

We established expected impacts per intervention and per mitigation behaviour, averaging out all contextual variables. This is a characteristic intrinsic to meta-analysis. As previously discussed (Round 1 page 4; Round 2 page 2; Round 3 pages 1-4), there is a difference between testing main effects versus examining moderation and mediation effects. We have made clear in multiple sections of the paper (Abstract and several parts of the Discussion) that the main effects of behavioural interventions may be moderated by contextual factors, and that packages of interventions may be more effective. We reinforce that this does not reduce the significance and value of average main effects.

Also, as previously discussed (Round 1 pages 12-13), the core strength of meta-analysis is its aggregate evidence of impact. Estimating average main effects from field experiments and large sample sizes is an appropriate method to draw general conclusions because it holds all other factors constant. This approach pays less attention to contextual specificities, but it allows for more generalizable conclusions, compared to conclusions based on specific packages of interventions, under specific circumstances.

Another issue that emerges more clearly to me in the revision is the paper's failure to distinguish frequently repeated behaviors, whose effects must be aggregated through multiple repetitions (e.g., travel to work, adjusting room temperatures, changing diet), from behaviors whose effects are almost fully achieved by a single action (e.g., upgrading the energy efficiency of building shells, adopting more energy-efficient home appliances and vehicles). The paper recognizes that nearly all the experimental studies meeting its selection criteria focus on frequently repeated behaviors (the only four studies that do not do so focus on appliance choice), but it does not fully consider the implications of that fact. One implication is that the establishment of habit (the issue of whether the behaviors "stand the test of time"), which has an important place in this paper, is important only for frequently repeated behaviors.

We very much thank the Reviewer for this important comment, and we fully agree that we should distinguish between frequent versus single-action mitigation behaviours. This may be an important distinction between our paper and Dietz et al. 2009, and we have included this additional clarification in the Discussion (please refer to track changes pages 9-10).

“Thirdly, Dietz et al. based a significant part of their analysis on several high-impact one-time actions (e.g., home weatherisation, purchase of energy-efficient home appliances and fuel-efficient vehicles), whereas our work mostly examined frequently occurring behaviours (e.g., daily energy and water

saving, recycling, food waste) because the available experimental field evidence mostly targeted frequently occurring behaviours. The possibility exists that single action behaviours may be more effectively influenced by (behavioural) interventions compared to recurring behaviours, which may be more resistant to change if automatized in habitual routines (32)''.

32. Webb, T. L., Sheeran, P., & Luszczynska, A. (2009). Planning to break unwanted habits: Habit strength moderates implementation intention effects on behaviour change. *British Journal of Social Psychology*, 48(3), 507-523.

Another concern arises from the fact that the most consequential behaviors (at least as analyzed on a time scale of years to decades, as is done in the Dietz et al., paper) are not, as the paper asserts, frequently occurring, but are infrequent, one-time actions that have lasting effects without changing daily practices. Of the behaviors studied by Dietz et al., home weatherization and acquisition of fuel-efficient vehicles, both one-time actions, have a far greater Potential Emissions Reduction (i.e., what would be achieved if these behaviors were universally adopted) than changes in appliance purchases. Other one-time actions that have become more widespread since the publication of that paper, such as adoption of photovoltaic (PV) energy systems and electric vehicles that can run on PV electricity, arguably also have much greater potential importance than appliance purchases. Thus, the fact that this paper does not address high-impact one-time actions involving acquiring lower-carbon household technologies or upgrading existing ones seriously compromises the conclusions that readers may draw from the paper. As Stern and Gardner noted as far back as their review in 1980, psychologists have tended to focus on frequently repeated behaviors, neglecting one-time behaviors that have major practical potential. This review suggests that this gap in the literature is still important. It may be understandable because it is difficult to do research on these behaviors that meets the methodological criteria of this review. However, a focus on behaviors with high Potential Emissions Reduction is critical for behavioral research to make its greatest contribution. Because the experimental literature has largely ignored the highest-potential actions, a review of that literature, such as this one, necessarily also ignores those actions, so its conclusions need to be qualified carefully before making general claims about the potential impact of behavioral interventions. (The 1980 reference is to Stern, P.C., and Gardner, G.T. Psychological research and energy policy. *American Psychologist*, 36(4), 329.)

We have revised the statement made in the Discussion page 12 2nd paragraph from “*most consequential household actions*” to “*most household actions examined in our work*”. Although the debate about technical impact is beyond our research goals, we would like to comment that other authors (our reference 19) support eating fewer animal products (a daily occurring behaviour) would reduce an individual’s footprint by 22%, whereas a fuel-efficient vehicle would reduce an individual’s footprint by 9%. Thus, it is not consensually accepted that one-time actions are the most consequential household behaviours to mitigate climate change.

Please note most remaining comments made by the Reviewer (e.g., about Potential Emissions Reduction, photovoltaic (PV) energy systems, electric vehicles, “... one-time behaviors that have major practical potential”) are related to technical potential, and less to the behavioural plasticity that can be achieved by behavioural interventions – our core focus. Nonetheless, due to the Reviewer’s recommendations in past rounds and the current comment that “*a focus on behaviors with high Potential Emissions Reduction is critical for behavioral research to make its greatest contribution*”, we remind that our first recommendation for future research is exactly this (page 11): “*...more research is needed*

to tackle behaviours with high technical potential, that is, behaviours that greatly contribute to reduce carbon emissions”.

We would like to address several strong, yet unsubstantiated claims made by the Reviewer. We consider the Reviewer’s claim that not addressing high-impact one-time actions “*seriously compromises the conclusions that readers may draw from the paper*” to be flawed. If there had been more field experimental evidence testing behavioural interventions on high-impact one-time actions, this evidence would have been included in our paper. Given that we included all available causal evidence, it is unclear on which grounds is the Reviewer making this claim. We also remind that the impact of behavioural interventions to promote the purchase of energy-efficient appliances (a one-time action) is very small and only marginally significant ($d=-0.036$ 95% CI $-0.129-0.058$). Therefore, the Reviewer is either grounding this claim on unverified assumptions or in evidence of lower methodological quality.

Moreover, we have cautioned the reader since the initial round of comments (Round 1 page 2 last paragraph; incorporated in the Discussion page 13) that our analysis did not include some potentially important household actions due to lack of causal field evidence. This is a research caveat, not a serious challenge to our conclusions. The Reviewer also states that “*it is difficult to do research on these behaviors that meets the methodological criteria of this review*”. We also consider this to be a flawed assumption. All field experiments implementing interventions in naturalistic settings are difficult, time-consuming projects, with many logistic hurdles. But targeting this type of decisions and behaviours is not uniquely more difficult than targeting daily energy consumption behaviours or food purchase and waste.

The above issues are important for considering the implications and potential importance of the findings reported here. To me, these findings indicate that behavioral researchers who focus their attention on frequently repeated behaviors or who seek universal generalizations about the effects of particular intervention strategies on particular types of behavior are barking up the wrong tree. This was the implicit position taken in reviews such as that of Wolske and Stern (2018) and other past work that has drawn on multiple research methods to offer practical advice to those seeking to reduce the climate footprint of actions taken in and by households.

I see two ways the journal editors and the paper’s author(s) might usefully proceed from here. One is for the authors to revise the paper again to take these issues explicitly into account. This would lead to the paper making weaker claims about the practical importance of the findings, but possibly stronger claims about the inadequacy of analyses, such as may be implicit in recent IPCC efforts, to the extent that they presume that behavioral interventions targeted at households can alone yield meaningful reduction on carbon emissions globally or in high-income countries. Thus, I think the data analyzed in this paper imply criticism of certain approaches to research on behavioral interventions, but do not imply that almost all such interventions will have little or no effect. Also, as noted above, the paper offers almost no evidence on the effects of behavioral interventions on one-time household actions, which offer the greatest potential impact on household climate footprints.

We would respectfully say we are not barking up the wrong tree, we are barking up a different tree. Throughout all previous rounds of comments, we have revised the paper to make weaker and more cautious claims to incorporate the Reviewer’s concerns. Our results do not imply that almost all behavioural interventions have little or no effect. Our results show that average effects are small, but that some interventions (i.e., social comparison, nudges) are more effective than others. We are also

categorical in acknowledging the possibility that packages of interventions may be more effective, but that this interaction approach needs to be more rigorously tested. Moreover, the comment about climate footprints is related to technical potential, and not to behavioural plasticity – our core focus.

We propose three additional efforts to address the Reviewer’s comments and concerns:

- First, we proposed a change to the title to describe our work in more neutral terms and to not prime the reader in any way: Meta-analysis of randomised controlled trials with 3M observations testing behavioural interventions to promote household action on climate change
- Second, we revised the Abstract to clarify in plain terms what our results mean to behavioural interventions: Our results do not imply behavioural interventions are less effective than alternative strategies such as financial incentives or regulations, nor exclude the possibility that behavioural interventions could have stronger effects when used in combination with alternative strategies. These possibilities should also be subject to rigorous evaluation.
- Third, to avoid what could be interpreted as a criticism to IPCC efforts, we have removed any direct mention to the IPCC Report in the Introduction 1st paragraph.

The authors may not want to take this direction. The alternative is to publish the paper more or less as is, but to commission a commentary that would raise points such as the above. I would prefer the first option, because more people are likely to read a review paper than a commentary on it and also because the story is best told all in one place. But the paper as is does not tell the whole story.

We value all constructive criticism to our work. The possibility to publish our paper in its current form and to commission a commentary was formerly proposed by the Reviewer two rounds ago (Round 2 page 2). We would accept such a suggestion.

Reviewer #2

I think the authors have addressed my comments well in this revision - particularly the issue regarding behavioral sustainability. Two minor points in need of clarification for the fifth recommendation: 1. Wondering if the authors can clarify their speculation regarding the statement that "the intervention duration did not have an effect". Wondering if there is any prior work that could help support this claim (i.e., "Thus, the expectation could be that, the longer the intervention, the more likely it is to create a positive habit in such recurring contexts.")?

We thank the Reviewer for the possibility to clarify this important issue. We examined the effect of intervention duration empirically, using meta-regression procedures described in page 3 (last paragraph and Figure 1) and page 4 (second last paragraph). The motivation to examine the impact of intervention duration is grounded in a robust literature about habit formation, showing that action repetition over time, particularly under the same environmental cues, is critical to establish new habits. Much of this work is based on animal behaviour and operant conditioning (Thraillkill & Bouton 2015) but empirical data from humans also exists, mainly in education and health settings (Lally et al. 2010). In this revised version of the manuscript, we have elaborated more on how action repetition over time may be needed to produce lasting behaviour changes (Discussion fifth recommendation page 12).

Habit is one of the persistence pathways proposed to explain sustained effects (54). Research on habit formation has shown that action repetition, particularly under the same environmental cues, is critical to establish new habits (59-60). Therefore, the expectation could be that, the longer the intervention, the more likely it is to create a positive habit in such recurring contexts. Yet, holding all factors constant, the intervention duration did not have an effect. This is puzzling, to some extent, because most household actions examined in our work are frequent and recurrent in stable settings (e.g., turning on the air conditioning at home, driving to work, selecting meal size and type). Our work included very heterogeneous research contexts and behaviours, possibly with different timescales required for change to happen, which may have masked the effect of intervention duration.

59. Lally, P., Van Jaarsveld, C. H., Potts, H. W., & Wardle, J. (2010). How are habits formed: Modelling habit formation in the real world. *European journal of social psychology*, 40(6), 998-1009.

60. Thraillkill, E. A., & Bouton, M. E. (2015). Contextual control of instrumental actions and habits. *Journal of Experimental Psychology: Animal Learning and Cognition*, 41(1), 69.

2. It will be helpful if the authors can further clarify whether they think the "one-shot intervention" is one of the reasons that makes "social comparison and nudges" have persistent effects. In other words, would the effect become not persistent for prolonged, repeated or sequential interventions? The way it is written now could make the readers think these two strategies are relatively more powerful, which may not be the message authors would like to convey.

Social comparison and nudges are not necessarily one-shot interventions. Our message is that the evidence for short-term effects is clear: social comparison and nudges are the most promising strategies. The question remains whether these would also be the most effective strategies in the long-run. Given that we could not provide a definitive answer on this, due to the limited papers that reported follow-up effects, we provided some examples as a point of discussion. Many of these isolated examples showing persistent effects were found in interventions using social comparison and nudges, some based on

interventions that lasted years (our Ref 56), but also in a specific case where participants received only a one-shot social comparison message (our Ref 5). These are just examples suggesting that social comparison and nudges could be the best strategies both in the short and long term. Please note some clarifications (with track changes) that have been added to this section in Discussion fifth recommendation page 12:

“Regarding whether behavioural interventions have persistent effects once the intervention is concluded, the available data showed an overall average null effect. We could not provide a definitive answer on persistent effects per specific type of intervention due to the small number of papers that reported follow-up effects. [...]

[...] Nonetheless, isolated examples show that evidence for changing future costs is not compelling (58) but some interventions using social comparison (external reinforcement), and nudges (automatic habit reshape) have reported striking persistent effects (59-60). Some of these persistent effects are based on interventions that lasted years (59), but there is also a specific case where participants received only a one-shot intervention (5).”

REVIEWERS' COMMENTS:

Reviewer #1 (Remarks to the Author):

I have summarized the major claims of this paper in reviews of previous versions. This revised paper, as before, offers a rigorous analysis of a set of data of broad interest concerning the potential of behavioral interventions, acting alone, to influence household actions to limit climate change. The latest revision does a better job of addressing some key complexities in drawing conclusions from the analysis about the potential of behavioral interventions in household actions in order to limit climate change. It will be a stimulus to important discussions among researchers and especially among those concerned with making practical progress in limiting climate change. I therefore believe that this version deserves publication.

However, I remain quite concerned that many readers will draw facile but inappropriate conclusions about the practical import of the paper, such as a conclusion that "behavioral interventions can have very little, if any, practical value for limiting household consumers' contributions to climate change." This paper could easily be read to support that conclusion, but it does not.

I therefore recommend to the editors that the paper be published along with one or more brief commentary pieces to begin the discussion that the paper should initiate. The authors of commentary(ies) should be asked to speak to the question of what the practical import of the paper is, and is not, for the use of "behavioral" interventions to limit climate change. This paper's author(s) should be invited to write a response to the commentary(ies), to be published in the same issue as the paper and the commentary(ies).

I therefore recommend to the editors that the paper be published along with one or more brief commentary pieces to begin the discussion that the paper should initiate. The authors of such commentaries should be asked to speak to the question of what the practical import of the paper is (and is not) for the use of "behavioral" interventions to limit climate change. This paper's author(s) should be invited to write a response to the commentary(ies), to be published in the same issue as the paper and the commentary(ies), so as to stimulate further discussion.

Paul C. Stern

Reviewer #2 (Remarks to the Author):

The clarifications made for this round of review are clear and helpful - I think they strengthen the paper's main arguments. I also think the revisions made to address reviewer 1's comments are helpful and important (the revised title & abstract).

Some additional comments: (1) Wondering if the references 59 & 60 listed right after "have reported striking persistent effects" are mistaken? I think these don't refer to papers by Lally et al. (2010) and Thrailkill et al. (2015). (2) I think it will be helpful for the authors to provide more details regarding what "striking persistent effects" mean?, like how effective are these nudge examples. And maybe instead of saying " *Some* of these persistent effects are based on interventions that lasted years", it will help to specify the number.

I also encourage the authors to include the major points addressed so far toward prior reviewers' comments within the study limitations paragraph, as I think it will help readers better understand the main rationale of this paper. This is similar to the new revisions being made in the Abstract: "(1) The intervention with the highest average effect size is choice architecture ("nudges") but this

strategy has been tested in a limited number of behaviors. (2) Our results do not imply behavioral interventions are less effective than alternative strategies such as financial incentives or regulations, nor exclude the possibility that behavioral interventions could have stronger effects when used in combination with alternative strategies." Re-phrasing these in some form in the limitation paragraph would be helpful.

Response to Reviewers

Reviewer #1

I have summarized the major claims of this paper in reviews of previous versions. This revised paper, as before, offers a rigorous analysis of a set of data of broad interest concerning the potential of behavioral interventions, acting alone, to influence household actions to limit climate change. The latest revision does a better job of addressing some key complexities in drawing conclusions from the analysis about the potential of behavioral interventions in household actions in order to limit climate change. It will be a stimulus to important discussions among researchers and especially among those concerned with making practical progress in limiting climate change. I therefore believe that this version deserves publication.

However, I remain quite concerned that many readers will draw facile but inappropriate conclusions about the practical import of the paper, such as a conclusion that “behavioral interventions can have very little, if any, practical value for limiting household consumers' contributions to climate change.” This paper could easily be read to support that conclusion, but it does not.

I therefore recommend to the editors that the paper be published along with one or more brief commentary pieces to begin the discussion that the paper should initiate. The authors of commentary(ies) should be asked to speak to the question of what the practical import of the paper is, and is not, for the use of “behavioral” interventions to limit climate change. This paper’s author(s) should be invited to write a response to the commentary(ies), to be published in the same issue as the paper and the commentary(ies).

I therefore recommend to the editors that the paper be published along with one or more brief commentary pieces to begin the discussion that the paper should initiate. The authors of such commentaries should be asked to speak to the question of what the practical import of the paper is (and is not) for the use of “behavioral” interventions to limit climate change. This paper’s author(s) should be invited to write a response to the commentary(ies), to be published in the same issue as the paper and the commentary(ies), so as to stimulate further discussion.

Paul C. Stern

Dear Paul,

We are grateful for your insightful comments and recommendations. We feel that the resulting manuscript is richer and more precise than the first version on which you commented. We look forward to reading your commentary, as well as the commentaries from other experts interested in this line of research. We are honoured to participate in this important debate.

Reviewer #2

*The clarifications made for this round of review are clear and helpful - I think they strengthen the paper's main arguments. I also think the revisions made to address reviewer 1's comments are helpful and important (the revised title & abstract). Some additional comments: (1) Wondering if the references 59 & 60 listed right after "have reported striking persistent effects" are mistaken? I think these don't refer to papers by Lally et al. (2010) and Thraillkill et al. (2015). (2) I think it will be helpful for the authors to provide more details regarding what "striking persistent effects" mean?, like how effective are these nudge examples. And maybe instead of saying " *Some* of these persistent effects are based on interventions that lasted years", it will help to specify the number.*

I also encourage the authors to include the major points addressed so far toward prior reviewers' comments within the study limitations paragraph, as I think it will help readers better understand the main rationale of this paper. This is similar to the new revisions being made in the Abstract: "(1) The intervention with the highest average effect size is choice architecture ("nudges") but this strategy has been tested in a limited number of behaviors. (2) Our results do not imply behavioral interventions are less effective than alternative strategies such as financial incentives or regulations, nor exclude the possibility that behavioral interventions could have stronger effects when used in combination with alternative strategies." Re-phrasing these in some form in the limitation paragraph would be helpful.

Dear Reviewer 2,

We would like to extend our gratitude and appreciation for your suggestions which greatly improved our manuscript.

(1) Please note the references for Thraillkill et al. (2015) and Lally et al. (2010) have been corrected, and are now, respectively, references (55) and (56).

(2) We have clarified this passage in the text (page 13): "...some interventions using social comparison (external reinforcement), and nudges (automatic habit reshape) have reported persistent effects. Ferraro and colleagues (5) report that the impact from a single social comparison treatment could be detected more than two years after this one-shot intervention (1.3% less water consumption in the treatment group). Allcott and Rogers (59) report that, after a two-year intervention using social comparisons to reduce energy consumption, effects persisted when the intervention was discontinued, decaying at 10-20 percent per year. -Regarding nudges, Kurz (60) showed that a three-month layout change in a restaurant produced a persistent reduction in meat consumption by 4% in the following three months after the intervention ended."

Regarding your comments about study limitations, please note that most reviewers' comments in the multiple rounds of review have been incorporated in the Discussion section.

As recommended, the sentence in the Abstract about nudges has been re-phrased and added to the limitations paragraph (page 14): "Another limitation worth mentioning is the small number of contexts in which nudges have been tested, mostly targeting reduced meat consumption and food waste. Although we recommend this type of intervention due to the higher effect sizes reported, this strategy has seldom been tested in behaviours with high technical impact, such as energy consumption or private car use – where the greatest gains for carbon reductions can be achieved."

Lastly, please note that the sentence in the Abstract "Our results do not imply behavioral interventions are less effective than alternative strategies..." is conveyed several times throughout the paper i.e., (i) in the Introduction page 3; (ii) the Discussion page 11 2nd paragraph, (iii) page 12 3rd paragraph, (iv) page 13 1st paragraph, and (v) page 14 3rd paragraph. It is not specifically mentioned in the limitations paragraph, but it is widely cautioned across the Discussion.